

This manuscript is a non-peer reviewed preprint submitted to EarthArXiv

# Environmental controls on the brGDGT and brGMGT distributions across the Seine River basin (NW France): Implications for bacterial tetraethers as a proxy for riverine runoff

Zhe-Xuan Zhang[1,2], Edith Parlanti[2], Christelle Anquetil[1], Jérôme Morelle[3], Anniet M. Laverman[4], Alexandre Thibault[5], Elisa Bou[6], Arnaud Huguet[1]*

1. Sorbonne Université, CNRS, EPHE, PSL, UMR METIS, Paris, 75005, France
2. Univ. Bordeaux, CNRS, Bordeaux INP, EPOC, UMR 5805, F-33600 Pessac, France
3. Department of Biology and CESAM – Centre for Environmental and Marine Studies, University of Aveiro, Campus de Santiago, Aveiro, 3810-193, Portugal
4. Univ. Rennes 1, CNRS, ECOBIO-UMR 6553, Rennes, 35000, France
5. Antea Group, Innovation Hub, 803 boulevard Duhamel du Monceau, Olivet, 45160, France
6. Université de Toulouse, CNRS, Toulouse INP, Université Toulouse 3 - Paul Sabatier (UPS), Laboratoire Ecologie Fonctionnelle et Environnement, Route de Narbonne 118, Toulouse, 31062, France

*Correspondence to*: Arnaud Huguet (arnaud.huguet@sorbonne-universite.fr)

**Abstract.** Branched glycerol dialkyl glycerol tetraethers (brGDGTs) are bacterial lipids that have been largely used as environmental proxies in continental paleorecords. Another group of related lipids, branched glycerol monoalkyl glycerol tetraethers (brGMGTs), has recently been proposed as a potential paleotemperature proxy. Nevertheless, the sources and environmental dependencies of both brGDGTs and brGMGTs along the river-sea continuum are still poorly understood, complicating their application as paleoenvironmental proxies in some aquatic settings. In this study, the sources of brGDGTs and brGMGTs and the potential factors controlling their distributions are explored across the Seine River basin (NW France), which encompasses the freshwater to seawater continuum. BrGDGTs and brGMGTs were analyzed in soils, Suspended Particulate Matter (SPM), and sediments ($n$=237) collected along the land-sea continuum of the Seine basin. Both types of compounds (i.e. brGDGTs and brGMGTs) are shown to be produced *in situ*, in freshwater and saltwater, based on their high concentrations and distinct distributions in aquatic settings (SPM and sediments) *vs*. soils. Redundancy analysis further shows that both salinity and nitrogen dominantly control the brGDGT distributions. Furthermore, the relative abundance of 6-methyl *vs*. 5-methyl brGDGTs (IR$_{6Me}$ ratio), Total Nitrogen (TN), $\delta^{15}N$ and chlorophyll *a* concentration co-vary in a specific geographical zone with low salinity, suggesting that 6-methyl brGDGTs are preferentially produced under low-salinity and high-productivity conditions. In contrast with brGDGTs, brGMGT distribution appears to be primarily regulated by salinity, with a distinct influence on the individual homologues. Salinity is positively correlated with homologues H1020a and H1020b, and negatively correlated with compounds H1020c and H1034b in SPM. This suggests that bacteria living in freshwater preferentially produce compounds H1020c and H1034b, whereas bacteria primarily growing in saltwater appear to be predominantly responsible for the production of homologues H1020a and H1020b. Based on the abundance ratio of the freshwater-derived compounds (H1020c and H1034b) *vs*. saltwater-derived homologues (H1020a and H1020b), a novel proxy, Riverine IndeX (RIX) is proposed to trace riverine organic matter inputs, with high values (>0.5) indicating higher riverine

contribution. We successfully applied RIX to the Godavari River basin (India) and a paleorecord across the upper Paleocene and lower Eocene from the Arctic Coring Expedition at Lomonosov Ridge, showing its potential applicability in both modern samples and in paleorecords.

## 1 Introduction


Branched glycerol dialkyl glycerol tetraethers (brGDGTs) are membrane lipids produced by bacteria, some of them belonging to the phylum *Acidobacteria* (Sinninghe Damsté et al., 2011; Chen et al., 2022; Halamka et al., 2023). These compounds were observed to occur ubiquitously in a wide range of terrestrial and aquatic environments (Schouten et al., 2013; Raberg et al., 2022). The distribution of brGDGTs (number of cyclopentane moieties and methyl groups; cf. structures in Fig. S1) has been

linked with pH and Mean Annual Air Temperature (MAAT) in soils (Weijers et al., 2007; De Jonge et al., 2014; Véquaud et al., 2022), peats (Naafs et al., 2017; Véquaud et al., 2022) and lake sediments (Tierney et al., 2010; Russell et al., 2018; Martínez-Sosa et al., 2021). The brGDGT-based proxies (i.e. MBT'$_{5ME}$ and CBT') have been largely applied to reconstruct paleoclimate from sedimentary archives (Coffinet et al., 2018; Harning et al., 2020; Wang et al., 2020).

In aquatic settings, brGDGTs were initially suggested to be transported by erosion to the sediments (Hopmans et al., 2004). Based on this assumption, the Branched and Isoprenoid Tetraethers (BIT) index was defined as the abundance ratio of the major brGDGTs to crenarchaeol (isoprenoid GDGT mainly produced by marine *Nitrososphaerota*). The BIT index ranges between 0 and 1, with high BIT values (around 1) reflecting a higher contribution of terrestrial organic matter compared to marine organic matter (Hopmans et al., 2004).


Over the last years, the BIT index has been broadly used to quantify the relative contribution of terrestrial organic matter in aquatic systems (Xu et al., 2020; Yedema et al., 2023) and to evaluate the reliability of the TEX$_{86}$ palaeothermometer (Cramwinckel et al., 2018). However, several studies have shown that brGDGTs can also be produced *in situ* in aquatic settings, including rivers (e.g. De Jonge et al., 2015; Freymond et al., 2017; Kim et al., 2015; Zell et al., 2014, 2013), lakes (Tierney

and Russell, 2009), and marine settings (Dearing Crampton-Flood et al., 2019; Zeng et al., 2023). This adds complexity to the identification of brGDGT sources in aquatic ecosystems and to the application of brGDGTs as (paleo)environmental proxies, including the BIT index.

The BIT values have all the more to be carefully interpreted, especially considering the potential influence of the selective

degradation of branched *vs.* isoprenoid GDGTs (Smith et al., 2012). Thus, complementary molecular proxies for quantifying the input of terrestrial organic matter to aquatic settings are still needed. These proxies may cross-validate other available terrestrial proxies, such as the $\delta^{13}C$ of organic carbon (Lamb et al., 2006), heterocyst glycolipids (Kang et al., 2023), and long-

chain diols (Lattaud et al., 2017). Recently, a machine-learning approach (BIGMaC model) was proposed to infer the origin of environmental samples (e.g. soil, peat, marine and lake settings) based on their GDGT distribution (Martínez-Sosa et al., 2023). While such an approach shows potential for differentiating distinct sources of GDGTs, its application to aquatic systems has not yet been extensively explored.

The improvement of chromatographic methods allowed the separation and quantification of 5-, 6- and 7-methyl brGDGTs (methyl groups at the fifth, sixth, and seventh positions; Fig. S1) that previously co-eluted (De Jonge et al., 2013, 2014; Ding et al., 2016). This led to the development of new brGDGT-based proxies based on these specific brGDGT isomers (De Jonge et al., 2014). Compounds eluting later than 7-methyl brGDGTs are tentatively designated 1050d and 1036d, as their exact chemical structures are currently unknown (Wang et al., 2021). The fractional abundance of the individual brGDGT isomers was shown to be influenced by distinct environmental factors. For example, the relative abundance of 5-methyl brGDGTs was correlated with temperature, whereas that of 6-methyl brGDGTs was correlated with pH (De Jonge et al., 2014). In addition, recent studies in lakes observed an influence of salinity on the relative abundance of 6-methyl, 7-methyl brGDGTs and their late-eluting compounds (Wang et al., 2021; Kou et al., 2022). This suggests that salinity could also control the distribution of these compounds in other systems like river-sea continuums but this assumption has not yet been studied.

Compared with brGDGTs, the branched glycerol monoalkyl glycerol tetraethers (brGMGTs, also referred as H-brGDGTs) are a much less studied group of lipids. Recent studies have revealed their presence in diverse environments, including peatlands (Naafs et al., 2018; Tang et al., 2021), marine settings (Liu et al., 2012; Xie et al., 2014), rivers (Kirkels et al., 2022a), soils (Baxter et al., 2021; Kirkels et al., 2022a) and lakes (Baxter et al., 2019, 2021). BrGMGTs are labelled as H1020, H1034, and H1048 respectively (cf. in Fig. S1), with isomers suggested by a suffix letter (a-c) following the order in which they elute according to Baxter et al. (2019). These compounds are structurally similar to brGDGTs, but possess an additional covalent carbon–carbon bond between the alkyl chains, leading to "H-shaped" structure. The bridge of brGMGTs was considered to be a primary adaptation to heat stress (Naafs et al., 2018; Baxter et al., 2019). Their presumed membrane stability under high temperature conditions was inferred from the behaviour of isoprenoid glycerol monoalkyl glycerol tetraethers (isoGMGTs), which were identified in a hyperthermophilic methanogen (Morii et al., 1998) and deep-sea hydrothermal vents (Schouten et al., 2008). Although a rigorous chemical characterization of brGMGTs is lacking and the source organisms of brGMGTs are unknown, correlations between the relative abundances of brGMGTs and MAAT were observed in peat soils (Naafs et al., 2018) and lakes (Baxter et al., 2019), showing their potential as temperature proxies. In addition to temperature, anoxic conditions may also trigger brGMGT production in the anoxic zone of peats (Naafs et al., 2018; Tang et al., 2021), anoxic part of the water column and/or sediments in lakes (Baxter et al., 2021), in regularly inundated soils (Kirkels et al., 2022a), as well as in the oxygen minimum zone in the marine environments (Xie et al., 2014). Furthermore, shifts in microbial community composition in response to other unknown environmental factors seem to control the relative abundances of brGMGTs in peats and lignites (Elling et al., 2023). Henceforth, in order to use the brGMGTs as environmental proxies in sedimentary records,

it is still necessary to understand which factors control their distributions in riverine and marine water columns and sediments. This remains to date poorly understood (Sluijs et al., 2020; Bijl et al., 2021).

Based on previous studies of brGDGTs and brGMGTs in terrestrial and marine settings (Dearing Crampton-Flood et al., 2019; Wang et al., 2021; Kirkels et al., 2022a, 2022b; Kou et al., 2022), we hypothesize (1) that both brGDGTs and brGMGTs can be produced *in situ* in aquatic systems and (2) that brGDGT and brGMGT distribution are influenced by surrounding environmental factors and vary spatially along the land-sea continuum. These compounds have a potential to be used as proxies of riverine organic matter inputs along estuaries. These hypotheses were tested by examining and comparing the distribution

of brGDGTs and brGMGTs in soils, suspended particulate matter (SPM) and sediments ($n = 237$) collected all along the Seine River basin (NW France), covering its riverine and estuarine parts. The aim of the present study was (1) to investigate the sources of brGDGTs and brGMGTs along the Seine land-sea continuum, (2) to determine the predominant environmental controls affecting the distribution of these molecules and (3) to assess the potential of brGMGTs as a riverine runoff proxy.

## 2 Material and methods

### 2.1 Study area

The Seine River basin (Seine River and its estuary; Fig. 1a) is more than 760km long, draining through the greater Paris region (over 12 million inhabitants) to the English Channel (Flipo et al., 2021). The Seine Estuary is a macrotidal estuary according to its large tidal range, small depth and morphology. The maximum flows are generally observed in winter (over 700 m$^3$/s; Fig. 1b), whereas the minimum flows are observed in summer (below 250 m$^3$/s; Fig. 1b). The tide influences the estuary up to the

city of Poses (site 5, KP 202 in Fig. 1a; KP represents kilometric point and is defined as the distance in kilometers from the city of Paris), where a dam constitutes the boundary between the river and the estuary. Based on spatiotemporal variations of salinity, the estuary can be divided into two major parts. The upstream estuary corresponds to the freshwater tidal sector (KP 202 to KP 298, from site 5 to site 12; Fig. 1a and Table 1) and the downstream estuary is influenced by a salinity gradient (starting at KP 298, from site 12 to the coastal area; Fig. 1a and Table 1) (Romero et al., 2016; Druine et al., 2018).



**Table 1.** Location of the sampling sites along the Seine Basin, with the type of samples collected

| Site | Name | Longitude (°) | Latitude (°) | KP | Zone | Date | Type |
|------|------|---------------|--------------|----|------|------|------|

| | | | | | | | |
|---|---|---|---|---|---|---|---|
| 1 | Marnay sur Seine | 3.56 | 48.51 | -200 | River | 2020-11 | Sub-surface SPM ($n=1$) |
| 2 | Bougival | 2.13 | 48.87 | 40 | River | 2020-11 | Sub-surface SPM ($n=1$) |
| 3 | Triel sur Seine | 2.00 | 48.98 | 80 | River | 2020-11 | Sub-surface SPM ($n=1$) |
| 4 | Les Andelys | 1.40 | 49.24 | 175 | River | 2019-6; 2019-7; 2020-9 | Sub-surface and bottom SPM ($n=6$) |
| 5 | Poses | 1.24 | 49.31 | 202 | Upstream estuary | 2016-4; 2020-11 | Sub-surface SPM ($n=2$) |
| 6 | Oissel | 1.10 | 49.34 | 229.4 | Upstream estuary | 2019-6; 2019-7; 2020-9 | Sub-surface and bottom SPM ($n=18$) |
| 7 | Rouen | 1.03 | 49.43 | 243 | Upstream estuary | 2016-4 | Sub-surface SPM ($n=1$); Sediments ($n=10$) |
| 8 | Petit Couronne | 1.01 | 49.38 | 251.3 | Upstream estuary | 2020-9; 2021-2; 2021-3 | Sub-surface SPM ($n=3$) |
| 9 | Haulot Sur Seine | 0.98 | 49.36 | 255.6 | Upstream estuary | 2019-6 | Sub-surface SPM ($n=1$) |
| 10 | Val des Leux | 0.92 | 49.40 | 265.55 | Upstream estuary | 2019-6; 2019-7; 2020-9 | Sub-surface and bottom SPM ($n=18$) |
| 11 | Duclair | 0.87 | 49.48 | 278 | Upstream estuary | 2020-9; 2021-2; 2021-3 | Sub-surface SPM ($n=3$) |
| 12 | Heurtauville | 0.82 | 49.45 | 297.65 | Downstream estuary | 2019-6 | Sub-surface SPM ($n=1$) |
| 13 | Caudebec | 0.75 | 49.52 | 310.5 | Downstream estuary | 2015-4; 2015-9; 2016-4; 2019-6; 2019-7; 2020-9; 2021-2; 2021-3 | Sub-surface and bottom SPM ($n=24$) |
| 14 | Vatteville-La-Rue | 0.67 | 49.47 | 318 | Downstream estuary | 2019-6 | Sub-surface SPM ($n=1$) |

| | | | | | | | |
|---|---|---|---|---|---|---|---|
| 15 | Tancarville | 0.47 | 49.47 | 337 | Downstream estuary | 2015-1; 2015-4; 2015-9; 2019-6; 2019-7; 2020-9; 2021-2; 2021-3 | Sub-surface and bottom SPM ($n$=24); Sediments ($n$=20) |
| 16 | Berville-Sur-Mer | 0.37 | 49.44 | 346 | Downstream estuary | 2019-6 | Sub-surface SPM ($n$=1) |
| 17 | Fatouville | 0.32 | 49.44 | 350 | Downstream estuary | 2015-4; 2015-7; 2015-9; 2016-4 | Sub-surface SPM ($n$=4); Sediments ($n$=28) |
| 18 | Honfleur | 0.23 | 49.43 | 355.8 | Downstream estuary | 2015-4; 2015-9; 2019-6; 2020-9; 2021-2; 2021-3 | Sub-surface SPM ($n$=6) |
| 19 | La Carosse | 0.03 | 49.48 | 370 | Downstream estuary | 2015-7; 2016-4; 2016-4 | Sub-surface SPM ($n$=2); Sediments ($n$=10) |
| A | n.a. | 3.72 | 48.56 | n.a. | Soil (around the river) | 2021-9 | Soil ($n$=1) |
| B | n.a. | 3.23 3.26 | 48.43 48.42 | n.a. | Soil (around the river) | 2021-9 | Soil ($n$=5) |
| C | n.a. | 3.11 3.13 | 48.83 48.85 | n.a. | Soil (around the river) | 2021-10 | Soil ($n$=3) |
| D | n.a. | 0.38 0.38 0.38 | 49.47 49.46 49.45 | n.a. | Soil (around the downstream estuary) | 2021-3; 2021-9 | Soil ($n$=8) |
| E | n.a. | 0.41 0.41 | 49.44 49.45 | n.a. | Soil (around the downstream estuary) | 2018-2; 2018-6; 2018-8; 2018-10; 2020-9; 2021-3 | Soil ($n$=34) |

## 2.2 Sampling

From June 2019 to March 2021, water samples ($n$=102) were collected across the Seine River (Fig. 1a). Sub-surface water (ca.
1m depth) samples were collected in high-flow (over 250 m³/s) and low-flow (below 250 m³/s) periods from the three zones
(river, upstream estuary and downstream estuary) of the Seine River basin (Table 1). At 5 sites (sites 4, 6, 10, 13, and 15, Fig.
1a and Table 1), both sub-surface and bottom water (2.2-16 m depth) samples were retrieved using a pump into precleaned
20L FLPE Nalgene carboys. Estuarine water samples (sites 6, 10, 13, and 15; Fig. 1a and Table 1) were collected at three tide
periods (high tide, low tide and mid tide). For these sites, 0.25-43 L of water were immediately filtered using pre-combusted
Whatman GF/F 0.7 µm glass fiber filters. After filtration, filters were freeze-dried, scratched and stored frozen at -20°C prior
to analysis.

Additional SPM samples ($n$=16; Table 1) used in this study for brGDGT and brGMGT analysis were collected from the
upstream and downstream estuary (site 5, 7, 13, 15, 17, 18, and 19; Fig. 1a and Table 1) in 2015 and 2016, as detailed by
Thibault et al. (2019). Sediments ($n$=68) from 7 cores (10-cm depth) were collected in the river channel at the same sites as
these SPM samples in 2015 and 2016 using a UWITEC corer as described by Thibault et al. (2019) (Table 1). These sediments
were further sliced (1-cm thickness) and freeze-dried. For each core, ten samples were analyzed for brGDGTs and brGMGTs,
except for the one collected at site 17 in April 2016, where no lipids were detected between 4-5 and 5-6 cm depth. Surficial
soils ($n$=9) were collected in the lateral area of the upstream section of the Seine River in 2021 (site A, B, and C, Fig. 1a and
Table 1) and freeze-dried. Additional wetland soils and mudflat sediments ($n$=42) were collected in the downstream estuary
in 2018, 2020, and 2021 (site D and E, Fig. 1a and Table 1), representing allochthonous material transported into the estuary
by tidal effect. These samples were collected at low tide using a plexiglass® core (4.5 cm depth), and back to the laboratory,
homogenized, freeze-dried, and ground using a ball mill (model MM400, Retsch®).

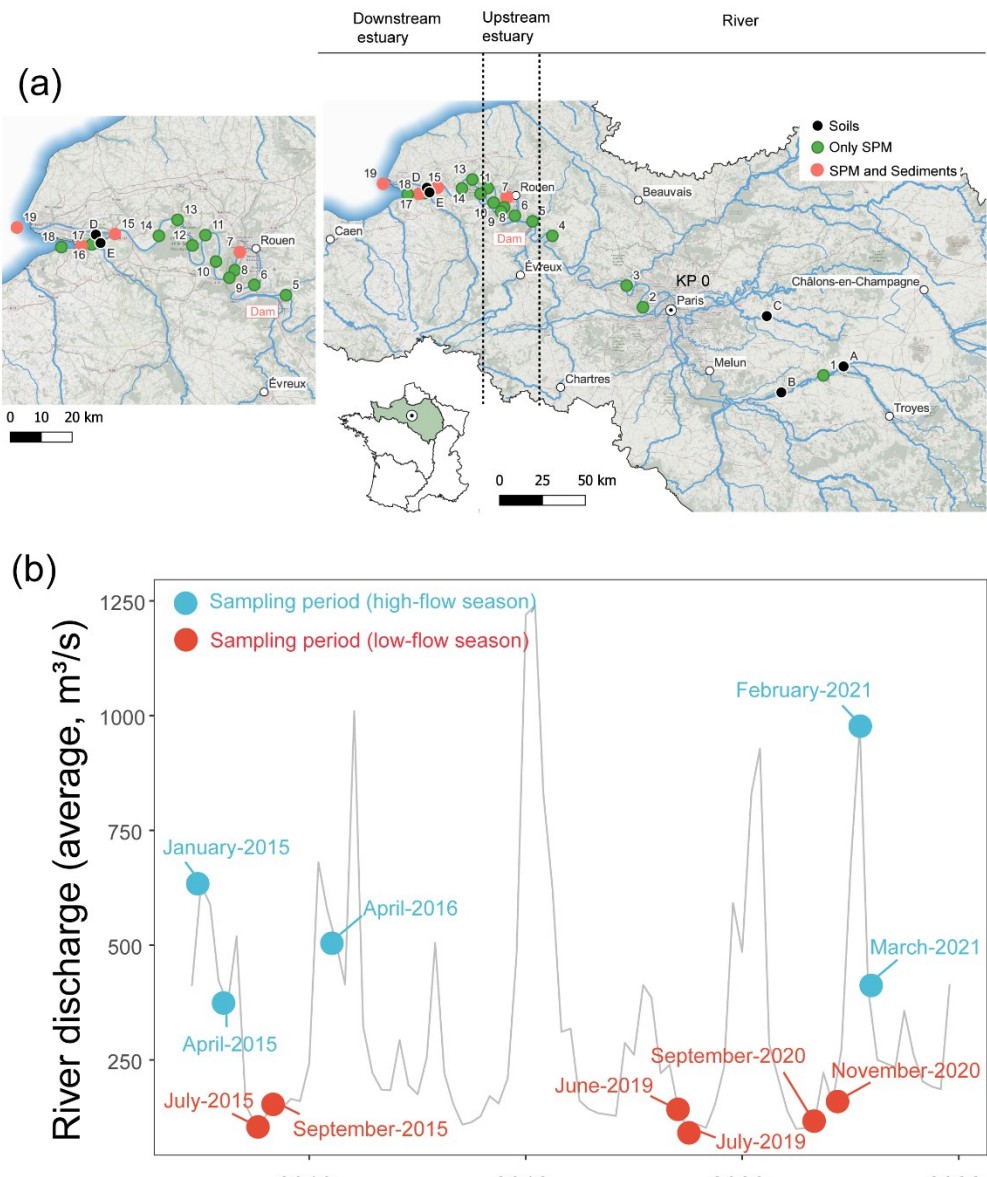

Figure 1: (a) Geographical locations of sampling sites in the Seine River Basin (KP: kilometric point, the distance in kilometers from the city of Paris (KP 0)). The sampling sites from upstream estuary and downstream estuary are shown in the zoom-in figure. Sub-surface SPM was collected for all sites from site 1 to site 18, while both sub-surface and bottom SPM were collected at sites 4, 6, 10, 13, and 15. (b) Mean monthly water discharge for the Seine River at the Paris Austerlitz station from 2015 to 2021 (data from https://www.hydro.eaufrance.fr/). Bullets represent the sampling period in high-flow (>250 m³/s - blue) and low-flow (<250 m³/s - red) seasons.

## 2.3 Elemental and isotopic analyses

Elemental and isotopic analyses of the soils (surficial soils and mudflat sediments, $n$=51) and SPM ($n$=102) collected from 2018 to 2021 were performed following the method described in Thibault et al. (2019). The samples were split, and one aliquot was decarbonated. Briefly, 40 mg of SPM and 1 g of soils/sediments samples were firstly decarbonated by adding 10 mL of 3 M HCl for 2 h with magnetic stirring at room temperature. Subsequently, these samples were rinsed using ultrapure water and centrifuged until reaching neutral pH. The obtained decarbonated samples were stored at −20 °C and freeze dried. Both decarbonated and non-decarbonated samples (~6 mg for SPM and ~20 mg for soils) were enclosed in a tin capsule. Total Organic Carbon content (TOC) and stable carbon isotopic composition ($\delta^{13}C$) were measured in decarbonated samples using an elemental analyzer coupled with an isotope ratio mass spectrometer (Thermo Fisher Scientific Delta V Advantage) at the ALYSES platform (Sorbonne University / IRD, Bondy, France). Total Nitrogen (TN) and nitrogen isotope ($\delta^{15}N$) were measured in non-decarbonated samples as acidification could impact the N contents (Ryba and Burgess, 2002). The isotopic composition ($\delta^{13}C$ or $\delta^{15}N$) was expressed as relative difference between isotopic ratios in samples and in standards (Vienna Pee Dee Belemnite for carbon and atmospheric $N_2$ for nitrogen). Additional elemental and isotopic data based on SPM and sediments collected in 2015 and 2016 ($n$=84) were obtained from Thibault et al. (2019).

## 2.4 Lipid extraction and analyses

The lipids from surficial soils and mudflat sediments (4–20g, $n$=51) and from SPM samples (~150mg, $n$=102) were extracted ultrasonically (3×) with 20 to 40 mL of dichloromethane (DCM): methanol (MeOH) (5/1, v/v) per extraction. Lipids from the SPM and sediments samples ($n$=84) collected in 2015 and 2016 were previously extracted by Thibault (2018) following the same method. The total lipid extracts were then separated into fractions of increasing polarity on an activated silica gel column (ca. 10 mg), using (i) 30 mL of heptane, (ii) 30 mL of heptane:DCM (1/4, v/v), and (iii) 30 mL of DCM/MeOH (1/1, v/v) as eluents. An aliquot (30%) of the third (polar) fraction containing GDGTs and GMGTs was dried, re-dissolved in heptane, and passed through a 0.2µm polytetrafluoroethylene (PTFE) filter (Ultrafree-MC; Merck). $C_{46}$ Glycerol Trialkyl Glycerol Tetraether (GTGT) was used as an internal standard (Huguet et al., 2006). 5 µl of this standard (0.01025 mg/mL) was typically added to 45 µl of sample.

GDGTs and GMGTs were analyzed using a Shimadzu LCMS 2020 high pressure liquid chromatography coupled with mass spectrometry with an atmospheric pressure chemical ionization source (HPLC-APCI-MS) in selected ion monitoring mode, modified from Hopmans et al. (2016) and Huguet et al. (2019). Tetraether lipids were separated with two silica columns in tandem (BEH HILIC columns, 2.1 × 150 mm, 1.7 µm; Waters) thermostated at 30℃. Injection volume was 30 µL. The flow rate was set at 0.2 mL/min. GDGTs and GMGTs were eluted isocratically for 25 min with 82% A/18% B (A= hexane, B=hexane/isopropanol 9/1, v/v), followed by a linear gradient to 65% A/35% B in 25 min, then a linear gradient to 100% B in 30 min, and back to 82% A/18% B in 4 min, maintained for 50 min. Identification of the different brGMGT isomers was

achieved by comparison of peak retention time with that of known brGMGTs in Baxter et al. (2019) and Kirkels et al. (2022a). Semi-quantification of GDGTs and brGMGTs was performed by comparing the integrated signal of the respective compound

with the signal of a $C_{46}$ synthesized internal standard (Huguet et al., 2006) assuming their response factors to be identical.

The detection limit was set at a signal-to-noise ratio (SNR) of 3. Peaks with lower SNR (<3) are not distinguishable from the background noise and are considered below the limit of quantification.

## 2.5 Calculation of GDGT proxies

The $IR_{6Me}$ index represents the proportion of 6-methyl brGDGTs *vs.* 5-methyl brGDGTs and was calculated according to De Jonge et al. (2015; Eq. 1) with Roman numbers referring to the structures in annex (Fig. S1):

$$IR_{6Me} = \frac{II_{a_6} + II_{b_6} + II_{c_6} + III_{a_6} + III_{b_6} + III_{c_6}}{II_{a_5} + II_{b_5} + II_{c_5} + II_{a_6} + II_{b_6} + II_{c_6} + III_{a_5} + III_{b_5} + III_{c_5} + III_{a_6} + III_{b_6} + III_{c_6}} \qquad (1)$$

The $IR_{7Me}$ index represents the proportion of 7-methyl brGDGTs and late-eluting isomers following Wang et al. (2021; Eq. 2):

$$IR_{7Me} = \frac{III_{a_7} + 1050d + II_{a_7} + 1036d}{III_{a_5} + III_{a_6} + III_{a_7} + 1050d + II_{a_5} + II_{a_6} + II_{a_7} + 1036d} \qquad (2)$$

The $IR_{6+7Me}$ index represents the average value between $IR_{6Me}$ and $IR_{7Me}$ according to Wang et al. (2021; Eq. 3):

$$IR_{6+7Me} = \frac{IR_{6Me} + IR_{7Me}}{2} \qquad (3)$$

The ACE index was calculated as follows (Turich and Freeman, 2011; Eq. 4):

$$ACE = \frac{Archaeol}{Archaeol + GDGT-0} \times 100 \qquad (4)$$

The BIT index including the 6-methyl brGDGTs was calculated following De Jonge et al. (2015; Eq. 5):

$$BIT = \frac{I_a + II_{a_5} + II_{a_6} + III_{a_5} + III_{a_6}}{I_a + II_{a_5} + II_{a_6} + III_{a_5} + III_{a_5} + crenarchaeol} \qquad (5)$$


Based on replicate injections of 3 different samples, the averaged standard deviations were 0.004 for $IR_{6Me}$, 0.005 for $IR_{7Me}$, 0.003 for $IR_{6+7Me}$, 8.54 for ACE, and 0.032 for BIT.

## 2.6 Water quality measurements

Water turbidity was measured by a CTD Probe Sea-bird®. Water temperature, dissolved oxygen, salinity, and pH were
measured using an automated YSI 6000 multi-parameter probe (YSI inc., Yellow springs, OH, USA). Chlorophyll *a* (Chl *a*)

concentrations were measured on water samples after filtration on Whatman GF/F 0.7 μm glass fiber filters, which were stored frozen (-20° C) before analysis. Chl *a* was extracted from filters with incubation in 10 ml of 90% acetone for 12 hours in the dark at 4°C. After two centrifugations (1700 g, 5 min), Chl *a* concentrations were measured using a Turner Designs Fluorometer according to the method of Strickland and Parsons (1972) as described in the reference protocol of SNO SOMLIT (Service d'observation du Milieu Littoral). Water quality measurements were performed at the Laboratoire Ecologie Fonctionnelle et Environnement (Université de Toulouse) as well as at UMR BOREA (Université de Caen Normandie).

### 2.7 Statistical analyses

All statistical analyses were performed using the R software (version 4.2.1). The non-parametric statistical tests were used due to the non-normal distribution of the dataset (tested by Shapiro–Wilk normality test; $p$-values $< 0.05$). Specifically, the Spearman's correlation was used to investigate potential correlations among different features (environmental parameters, fractional abundances of brGDGTs and brGMGTs, and proxies derived from these compounds), and the unpaired two-samples Wilcoxon test (also known as Mann-Whitney test or Wilcoxon rank sum test) was used for two independent group comparisons. Significance level is indicated by asterisks: *$p$-value $< 0.05$; **$p$-value $< 0.01$; ***$p$-value $< 0.001$; ****$p$-value $< 0.0001$; ns (not significant), $p$-value $> 0.05$.

A Principal Component Analysis (PCA) was performed on the fractional abundances of brGDGTs and brGMGTs, using the R packages factoextra and FactoMineR. The different groups of samples were highlighted by adding 95% concentration ellipses. The proportion of variance in brGDGT and brGMGT compositions that can be explained by different groups was evaluated by permutational multivariate analysis of variance using distance matrices (adonis) in the adonis2 function of the R package vegan, using the Bray-Curtis distances and 999 permutations.

A Redundancy analysis (RDA) was performed using the R package vegan to investigate the relationship between environmental parameters and brGDGT or brGMGT distributions in SPM. Angles between brGDGTs or brGMGTs and environmental factors were used to identify the potential relationships. Right angles (90°) reflect a lack of linear correlations, whereas small or straight angles (close to 0° or 180°, respectively) imply positive or negative linear correlations. The compounds that are close to each other were assumed to be strongly linked, representing similar distribution patterns and comparable responses to the environmental conditions. To evaluate the relative importance of each explanatory variable (environmental parameters) on brGDGT or brGMGT distributions, a hierarchical partitioning method implemented in the R package rdacca.hp was used. Briefly, this approach suggests that shared variance can be decomposed into equal components based on the number of involved predictors (environmental factors), allowing for the estimation of the relative importance of each predictor by adding its partial $R^2$ to the sum of all allocated average shared $R^2$. While most selection procedures, such as forward selection, use predictor ordering to assess variable importance, hierarchical partitioning calculates individual

importance (the sum of unique and total average shared effects) from all subset models, generating an unordered assessment of variable importance (Lai et al., 2022).

270

Spatio-temporal variations of environmental factors and proxies derived from brGDGTs and brGMGTs were assessed after applying a locally estimated scatterplot smoothing (LOESS) method. This method identifies nonlinear data patterns and buffers the effect of aberrant data and outliers. LOESS was implemented by the geom_smooth function of the R package ggplot2.

## 2.8 Machine learning

The BigMac model, developed by Martínez-Sosa et al. (2023) based on brGDGT and isoGDGT distribution, was applied. Subsequently, using the same algorithm (random forest), we developed our own model based on either brGDGTs or brGMGTs.

For independent models, our lipid dataset was split into a training set (75%) and a test set (25%). We then used a supervised machine-learning algorithm (random forest) to train models. This algorithm was applied to classify the downstream estuary 280 and soil samples based on brGDGTs or brGMGTs as input, implemented using the scikit-learn library (https://github.com/scikit-learn/) (Pedregosa et al., 2011) in Python (version 3.10.12). Hyperparameter tuning was conducted using a randomized search approach implemented through the RandomizedSearchCV function in scikit-learn.

SHapley Additive exPlanations (SHAP) is a game-theoretical method used to interpret machine learning models (Lundberg et 285 al., 2020). SHAP analysis was applied to identify which compounds were important for the classifications, implemented by the SHAP library in Python. A higher SHAP value indicates a more substantial contribution of the feature (brGDGTs or brGMGTs) to the predicted outcome (downstream estuary or soils).

# 3 Results

## 3.1 Distribution of bulk parameters from land to sea

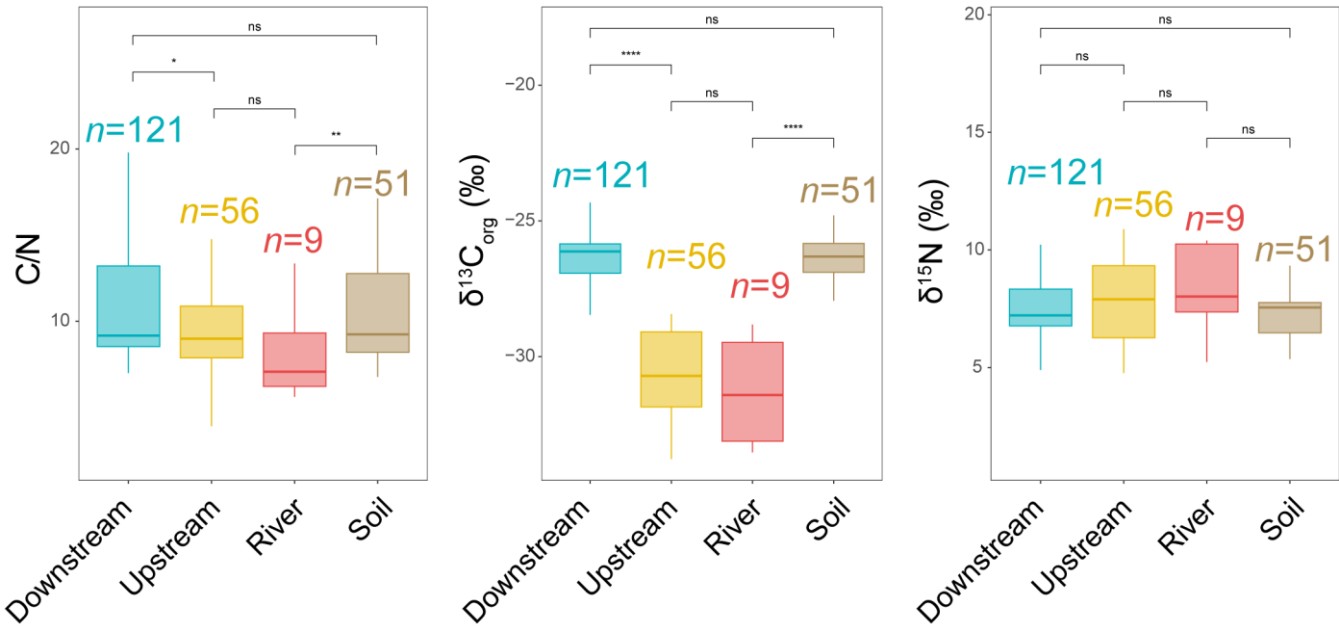

**Figure 2: Distribution of bulk parameters (C/N, $\delta^{13}C_{org}$ and $\delta^{15}N$) from soils (surficial soils and mudflat sediments) as well as river, upstream estuary and downstream estuary samples across the Seine River basin. Box plots of upstream and downstream estuary samples are based on SPM and sediments, whereas those of river samples are based only on SPM. Boxes show the upper and lower quartiles of the data, and whiskers show the range of the data, which are color-coded based on the sample type (river in red, upstream estuary in yellow, and downstream estuary in blue). The center-line in the boxes indicates the median value of the dataset. Statistical testing was performed by a Wilcoxon test (\*$p < 0.05$; \*\*$p < 0.01$; \*\*\*$p < 0.001$; \*\*\*\*$p < 0.0001$; ns, not significant, $p > 0.05$).**

The total organic carbon (TOC) content was significantly higher in the upstream estuary (4.64±1.42 %, based on SPM and river channel sediments) than in downstream estuary (3.30±1.69 %, based on SPM and sediments), soils (3.03±3.49 %, based on surficial soils and mudflat sediments) and river (2.88±1.14 %, based on SPM) (Table S1). The total nitrogen (TN) content was higher in the upstream estuary (0.51±0.17 %, based on SPM and sediments) than in the river (0.37±0.15 %, based on SPM), downstream estuary (0.31±0.14 %, based on SPM and river channel sediments), and soils (0.24±0.17 %, based on surficial soils and mudflat sediments) (Table S1). Lower C/N values were observed in the river (8.04±4.31, based on SPM) and upstream estuary (9.42±3.67, based on SPM and sediments) compared to the downstream estuary (10.73±3.59, based on SPM and sediments) and soils (11.59±4.79, based on surficial soils and mudflat sediments) (Fig. 2). Much lower values of $\delta^{13}C_{org}$ were observed in river (-31.30±1.91 ‰, based on SPM) and upstream estuary (-30.62±1.66 ‰, based on SPM and sediments) than in the downstream estuary (-26.45±1.34 ‰, based on SPM and sediments) and soils (-26.55±1.13 ‰, based on surficial soils and mudflat sediments) (Fig. 2). In addition, no significant differences in $\delta^{15}N$ were observed along the river basin (Fig. 2).

## 3.2 Distribution of brGDGTs from land to sea

The different brGDGTs (IIIa$_5$, IIIb$_5$, IIIc$_5$, IIa$_5$, IIb$_5$, IIc$_5$, IIIa$_6$, IIIb$_6$, IIIc$_6$, IIa$_6$, IIb$_6$, IIc$_6$, IIIa$_7$, IIIb$_7$, IIa$_7$, Ia, Ib, Ic, 1050d, and 1036d) were detected in all studied samples. The brGDGT chromatograms from upstream samples (SPM and river channel sediments) differed markedly from downstream estuarine samples (SPM and sediments). For example, 6-methyl brGDGTs were much more abundant than 5-methyl brGDGTs in the river (SPM) and upstream estuary (SPM), whereas the strong predominance of 6-methyl *vs.* 5-methyl brGDGTs decreased in the downstream estuarine SPM samples (Fig. S2). Furthermore, the peaks of the recently described 7-methyl brGDGTs and their late-eluting isomers (i.e. 1050d) were more pronounced in the downstream estuary than in the rest of the Seine basin (Fig. S2).

The relative abundances of the brGDGTs were determined all along the Seine River basin (Fig. 3 and Figs. S3, S4). The 6-methyl brGDGTs (IIIa$_6$, IIa$_6$, IIIb$_6$, IIb$_6$, and IIIc$_6$) were significantly higher in river (SPM) and upstream estuary (SPM and river channel sediments) than in soils (surficial soils and mudflat sediments) and downstream estuary (SPM and river channel sediments). In addition, the relative abundances of 7-methyl brGDGTs (IIIa$_7$ and IIa$_7$) and their late-eluting compound (1050d) were significantly higher in downstream estuary (SPM and river channel sediments) and soils (surficial soils and mudflat sediments) than in river (SPM) and the upstream estuary (SPM and river channel sediments).

The concentration of total brGDGTs also showed differences along the land to sea continuum (Fig. S5a). The total brGDGTs concentration decreased from river (10.51 ± 5.91 µg/g organic carbon (C$_{org}$), based on SPM samples) to upstream estuary (7.52 ± 5.09 µg/g C$_{org}$, based on SPM and sediments) and downstream estuary (4.95 ± 4.09 µg/g C$_{org}$, based SPM and sediments). In soils from all the Seine basin, the concentration in total brGDGTs (1.55 ± 1.61 µg/g C$_{org}$, based on surficial soils and mudflat sediments) was significantly lower than that in SPM and sediments (Fig. S5a).

A Principal Component Analysis (PCA) was performed to statistically compare the fractional abundances of brGDGTs from different location (river, upstream and downstream estuary, based on SPM and sediments collected in the river channel), which explained 54.1% of the variance in the first two dimensions (Fig. 4a). The first axis (PC1) explained 40.9% of the variance, with negative loadings for most of the 6-methyl brGDGTs and positive loadings for the remaining brGDGTs (Fig. 4a). Samples from the downstream estuary clustered apart from those from the river and upstream parts. Specifically, the brGDGT distribution was dominated by 6-methyl brGDGTs (IIIa$_6$, IIIb$_6$, IIIc$_6$, IIa$_6$, and IIb$_6$) in river and upstream estuarine samples, whereas in downstream estuary, it was driven by 5-methyl brGDGTs (III$_5$, IIa$_5$, IIc$_5$, IIb$_5$ and IIIb$_5$), tetramethylated brGDGTs (Ia, Ib, and Ic), 7-methyl brGDGTs (IIIa$_7$, IIa$_7$, and IIb$_7$), and their late-eluting compounds (1050d and 1036d). The brGDGT distributions of soils (surficial soils and mudflat sediments) were included in the PCA biplot performed on SPM and river channel sediments. This revealed that the brGDGT distribution in soils mostly overlap with the one in downstream estuarine SPM and river channel sediments (Fig. 4a).

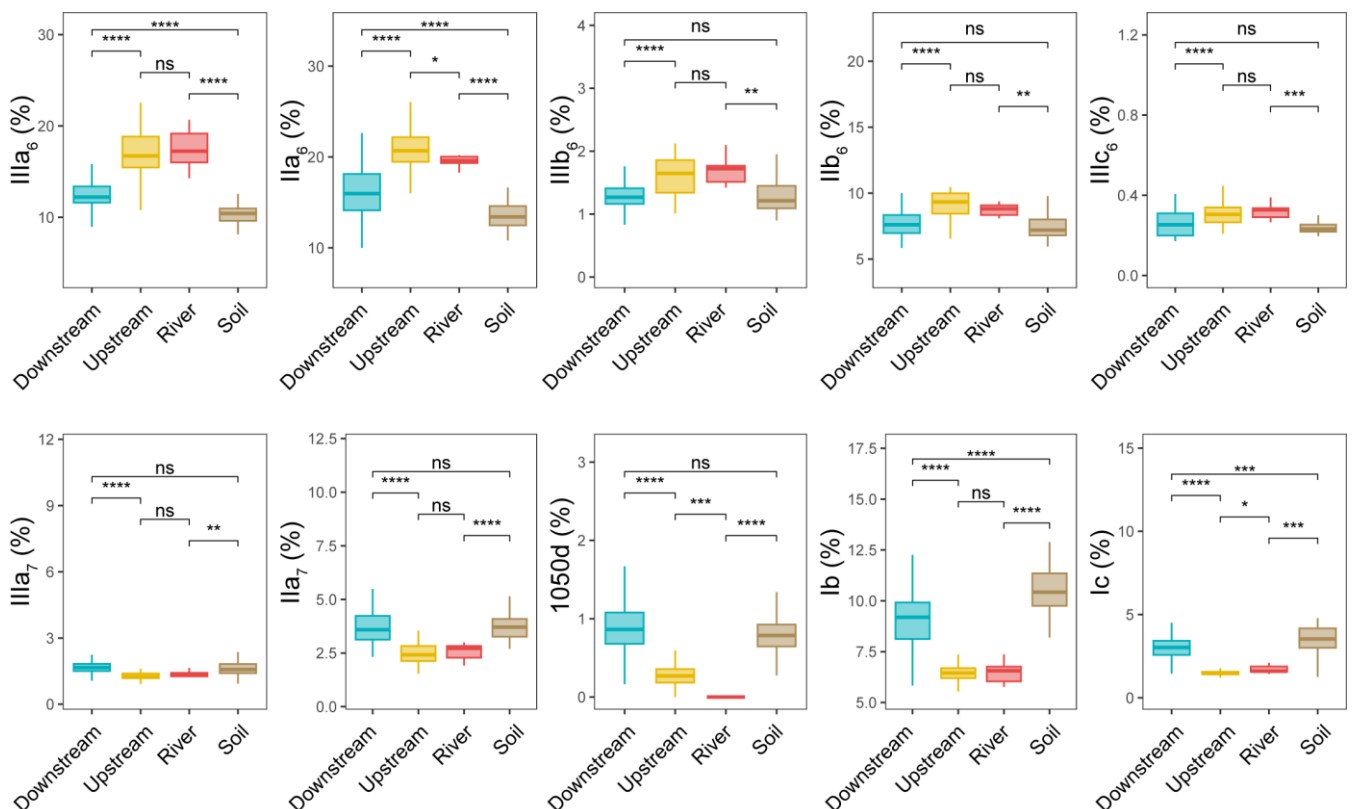

**Figure 3: Relative abundances of selected individual brGDGTs over 20 brGDGTs (IIIa5, IIIb5, IIIc5, IIa5, IIb5, IIc5, IIIa6, IIIb6, IIIc6, IIa6, IIb6, IIc6, IIIa7, IIIb7, IIa7, Ia, Ib, Ic, 1050d, and 1036d) from soils (surficial soils and mudflat soils/sediments, *n*=51), river (*n*=9), upstream estuary (*n*=56), and downstream estuary (*n*=121) samples across the Seine River basin: cyclopentane-containing tetramethylated brGDGTs (Ib and Ic), 6-methyl brGDGTs (IIa6, IIIa6, IIb6, IIIb6, and IIIc6), 7-methyl brGDGTs (IIa7 and IIIa7) and brGDGT 1050d. Box plots of upstream and downstream estuary samples are based on SPM and sediments, whereas those of**
**river samples are based only on SPM. Boxes show the upper and lower quartiles of the data, and whiskers show the range of the data, which are color-coded based on the sample type (river in red, upstream estuary in yellow, and downstream estuary in blue). The center-line in the boxes indicates the median value of the dataset. Statistical testing was performed by a Wilcoxon test (*$P < 0.05$; **$P < 0.01$; ***$P < 0.001$; ****$P < 0.0001$; ns, not significant, $P > 0.05$).**

A Redundancy analysis (RDA) was performed to investigate the influence of the environmental factors (TOC, TN, temperature, water discharge and salinity) on the brGDGT distributions in SPM samples (Fig. 5a and Table 2). It allowed to explain 39.79% of the variability through two dimensions. The RDA triplot (Fig. 5) showed how these factors correlate to the distributions of individual brGDGTs. The first axis of the RDA explained 33.16% of the variability and was primarily correlated with salinity and TN, whereas the second axis explained 6.63% of the variability and was associated with temperature, water discharge and
TOC (Fig. 5a and Table 2). Based on hierarchical partitioning, salinity and TN were the two most important variables in explaining the brGDGT variations (individual importance of 14.97 % for salinity and 13.47 % for TN; Fig. 5b and Table 2).

Compared with the salinity and TN, other available parameters have much lower individual importance (3.68 % for water discharge, 3.6 % for temperature and 2.12 % for TOC; Fig. 5b and Table 2).

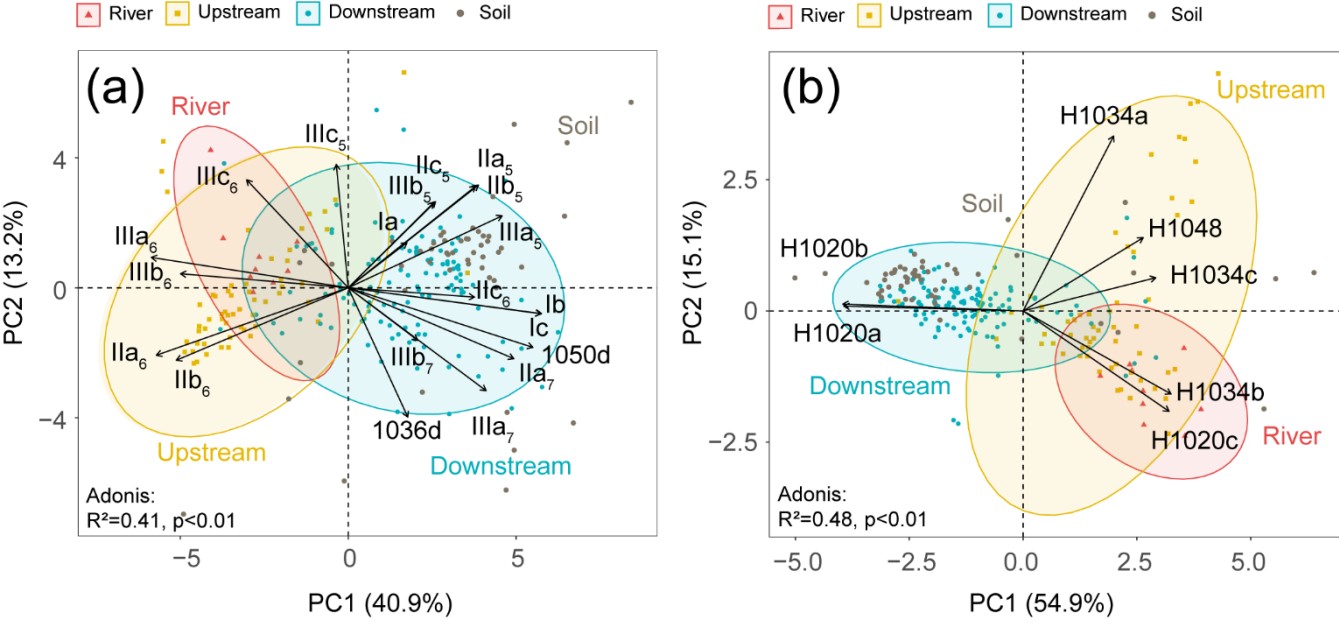

Figure 4: PCA analysis of fractional abundances of (a) brGDGTs and (b) brGMGTs. The coordinates of passive individuals (soils) are added passively as an overlay. They are predicted based on the information provided by the existing PCA performed on SPM and sediments (active individuals). Adonis analysis was used to evaluate how variation can be explained by the variables (999 permutations).

Table 2. RDA results

|  | variables | RDA scores | | Individual Importance (%) |
|---|---|---|---|---|
|  |  | Axis 1 | Axis 2 |  |
| brGDGTs | TOC | 0.17 | -0.49 | 2.12 * |
|  | TN | 0.72 | -0.23 | 13.47 *** |
|  | Temperature | 0.41 | -0.82 | 3.6 ** |
|  | Salinity | -0.73 | -0.10 | 14.97 *** |
|  | Water discharge | -0.30 | 0.93 | 3.68 ** |
| brGMGTs | TOC | 0.08 | 0.68 | 3.5 ** |
|  | TN | -0.34 | 0.09 | 4.18 ** |
|  | Temperature | 0.04 | 0.64 | 1.17 ns |
|  | Salinity | 0.83 | -0.50 | 17.45 *** |
|  | Water discharge | -0.17 | -0.71 | 2.16 * |


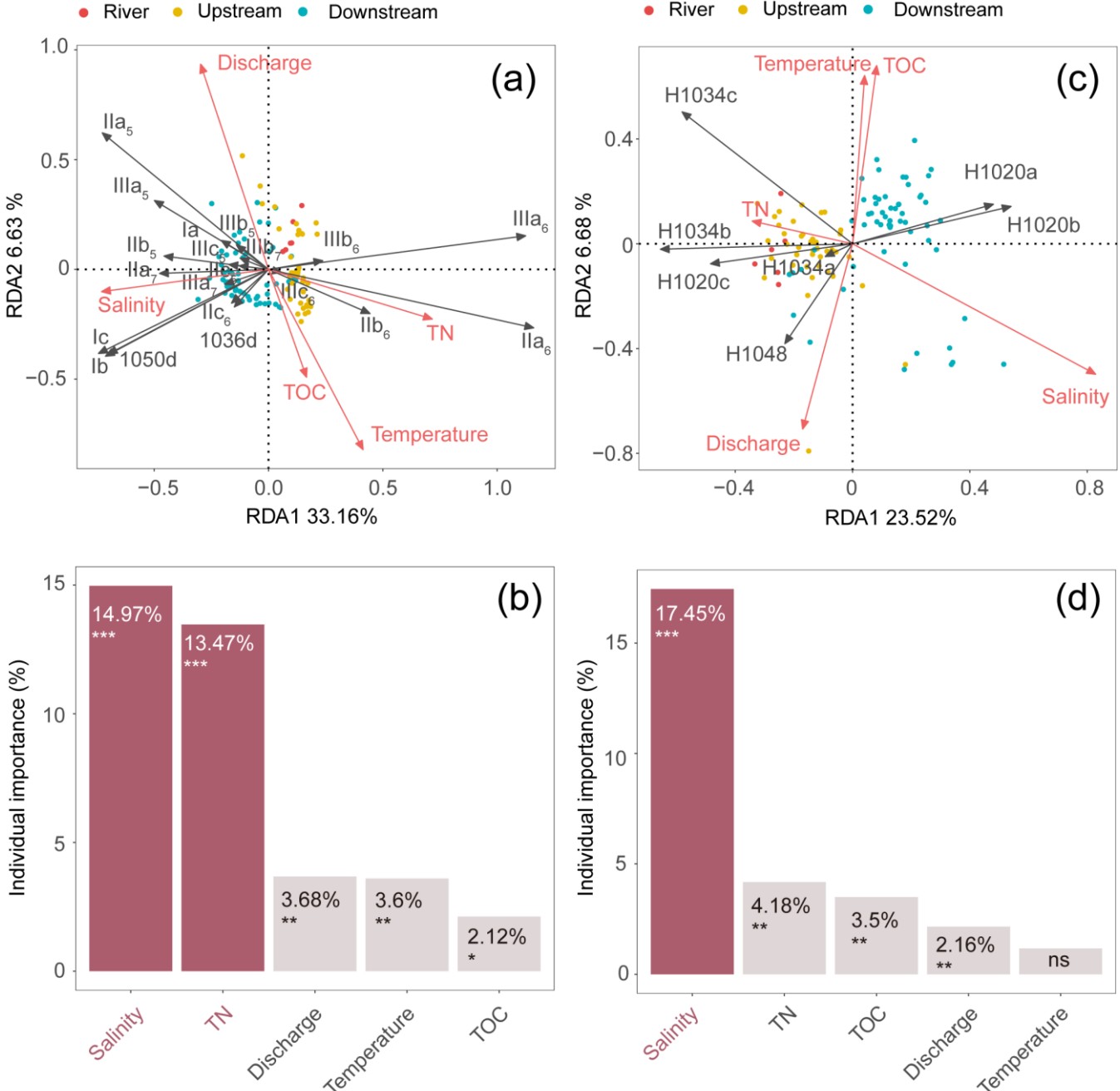

**Figure 5: RDA analysis showing relationships between environmental factors (TN, TOC, salinity, temperature, discharge, red arrows) and fractional abundances of (a) brGDGTs and (c) brGMGTs. The individual importance of the environmental factors (TN, TOC, salinity, temperature, and discharge) explaining the variation in (b) brGDGT and (d) brGMGT distributions was determined by hierarchical partitioning analysis. The dataset used for RDA analysis is composed of SPM from river ($n$=6; red), upstream estuary ($n$=42; yellow) and downstream estuary ($n$=59; blue). Significance level is indicated by asterisks: *$p$ < 0.05; **$p$ < 0.01; ***$p$ < 0.001; ns, not significant, $p$ > 0.05. $p$-values are derived from permutation tests (999 randomizations).**

### 3.3 Distribution of brGMGTs from land to sea

The brGMGTs (H1020a, H1020b, H1020c, H1034a, H1034b, H1034c, and H1048) identified by Baxter et al. (2019) were
detected in the samples collected across the Seine River basin. H1034a is the least abundant isomer and is below detection
limit for most of the SPM and sediment samples in the Seine River basin (Fig. S6 and S7). The chromatograms revealed
distinct distributions in brGMGTs in the different parts of the basin (SPM and sediments), with e.g. a higher intensity for the
homologue H1020c in the river samples (SPM) than in those from the upstream (SPM) and downstream estuary (SPM) (Fig.
S2). These spatial variations were apparent when calculating the fractional abundances of the individual brGMGTs (Fig. 6 and
Figs. S5, S6). From river to downstream estuary, the relative abundances in H1020a and H1020b increased, whereas those in
1020c and H1034b decreased (Fig. 6). In SPM and river channel sediments, the total brGMGT concentration was observed to
be slightly (but not significantly) higher in the riverine part ($0.26 \pm 0.24$ µg/g $C_{org}$) than in downstream estuary ($0.20 \pm 0.13$
µg/g $C_{org}$) and upstream estuary samples ($0.17 \pm 0.18$ µg/g $C_{org}$; Fig. S5b). The total brGMGT concentrations were the lowest
in soils (surficial soils and mudflat sediments) all over the basin ($0.07 \pm 0.09$ µg/g $C_{org}$; Fig. S5b).


The PCA analysis based on the brGMGT relative abundances (Fig. 4b) explained 70 % of the variance in the first two
dimensions, which separates samples from different parts of the basin. The first axis explained 54.9 % of the variance,
separating downstream estuarine samples from riverine and upstream estuarine samples, with negative loadings for two
brGMGTs (H1020a and H1020b), and positive loadings for the remaining brGMGTs (H1020c, H1034a, H1034b, H1034c,
and H1048). The second axis explained 15.1% of the variance and mainly separated the riverine and upstream estuarine
samples, with higher relative abundances of compounds H1020c and H1034b in riverine samples (Fig. 4b). The soil brGMGT
distributions were passively added to the PCA biplot based on SPM and sediments, revealing that the soils largely overlap
with the SPM and sediments collected in the downstream estuary (Fig. 4b).

The RDA was performed to investigate the factors that could explain the variability of brGMGT distributions in SPM samples
(Fig. 5c and Table 2) and explains 30.2 % of the variance in the first two axes. The RDA triplot showed that the first axis,
accounting for 23.52 % of the variability, was mainly associated with salinity and to a lesser extent TN, while the second axis
(6.68 %) was mainly driven by temperature, TOC and water discharge (Fig. 5c and Table 2). Based on hierarchical partitioning,
salinity had the highest individual importance (17.45 %) in explaining the variability of brGMGT distribution followed by TN
(4.18 %), TOC (3.5 %), and water discharge (2.16 %) (Fig. 5d and Table 2).

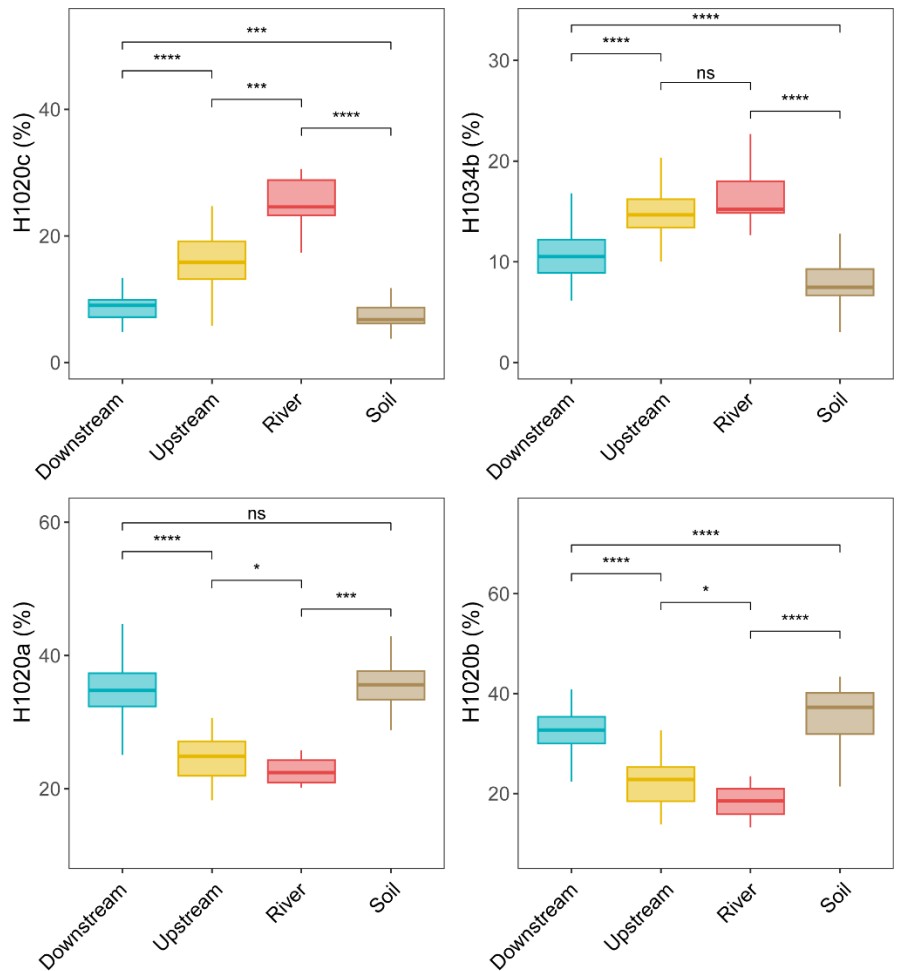

**Figure 6: Relative abundance of selected individual brGMGTs over 7 brGMGTs (H1020a, H1020b, H1020c, H1034a, H1034b, H1034c, and H1048) from soils (surficial soils and mudflat soils/sediments, *n*=51), river (*n*=9), upstream estuary (*n*=56) and downstream estuary (*n*=121) across the Seine River basin. Box plots of upstream and downstream estuary are composed of SPM and river channel sediments, whereas those of river are composed of SPM. Boxes show the upper and lower quartiles of the data, and whiskers show the range of the data, which are color-coded based on the sample type (river in red, upstream estuary in yellow, and downstream estuary in blue). The center-line in the boxes indicates the median value of the dataset. Statistical testing was performed by a Wilcoxon test (\*$p < 0.05$; \*\*$p < 0.01$; \*\*\*$p < 0.001$; \*\*\*\*$p < 0.0001$; ns, not significant, $p > 0.05$).**

## 4 Discussion

### 4.1 Sources of brGDGTs and environmental controls on their distribution

#### 4.1.1 Sources of brGDGTs

In order to determine the predominant origin of brGDGTs in the Seine River basin, the overall brGDGT concentrations and distributions in SPM and river channel sediments (*n*=186) were compared with those in soils (surficial soils and mudflat

sediments, $n=51$). The brGDGT concentrations (normalized to $C_{org}$) and relative abundances of several brGDGTs (i.e. $IIIa_6$, $IIa_6$, $IIIb_6$, $IIb_6$, and $IIIc_6$) in the SPM and sediments were significantly higher than those in soils ($p<0.05$, Wilcoxon test; Fig. S5a and Fig. 3). Such differences in brGDGT concentrations and relative abundances between soils and aquatic settings (SPM and sediments) imply that at least part of the brGDGTs in the water column and sediments of the Seine River basin is produced *in situ*. This is in agreement with previous findings which suggested an *in situ* aquatic contribution to the brGDGT pool (Peterse

et al., 2009; De Jonge et al., 2015; Crampton-Flood et al., 2021; Kirkels et al., 2022b).

More specifically, the fractional abundances of the two major 6-methyl brGDGTs ($IIa_6$ and $IIIa_6$) are significantly higher in the Seine River and upstream estuary than in soils (Fig. 3). This confirms that these brGDGTs are mostly produced within the river, adding to the growing body of evidence supporting riverine 6-methyl brGDGT production in water column and/or

sediment (De Jonge et al., 2015; Bertassoli et al., 2022; Kirkels et al., 2022b). A subsequent shift in the brGDGT distributions in the downstream estuary compared to the upstream areas is observed in the Seine River basin. The PCA analysis shows a separation of downstream estuarine samples (influenced by seawater intrusion) from riverine and upstream estuary ones (without significant seawater intrusion) (Fig. 4a). This difference is predominantly driven by the higher abundances of 6-methyl brGDGTs in riverine and upstream estuarine samples *vs.* higher abundances of 5- and 7-methyl brGDGTs as well as

compounds Ib, Ic, and late eluting brGDGTs 1050d, 1036d in downstream estuarine samples (Figs. 3, 4a and Fig. S4). The decrease in the fractional abundance of 6-methyl brGDGTs from the upstream estuary to the downstream estuary cannot be explained by the dam located at Poses (Fig. 1a). This dam separates the riverine part of the Seine from the upstream estuarine section. Even during the low-flow season (Fig. 1b), at least part of the water from the Seine River upstream of Poses flows into the estuary (Romero et al., 2019). Thus, the dam should not prevent (part of) the riverine brGDGTs associated with SPM

from reaching the estuary. It cannot be excluded that part of the riverine sediments is trapped by this dam. Nevertheless, all our estuarine samples were collected downstream of the dam, implying that the observed changes in brGDGT abundance and distribution within the estuary are intrinsic to the biogeochemical functioning of the Seine estuary and cannot be attributed to the dam.

Instead, a shift in brGDGTs along the land-sea continuum may reflect the fact that riverine 6-methyl brGDGTs are more easily degraded than soil-derived homologues and only partially transferred downstream. This hypothesis is based in a previous study, which showed a shift in brGDGT distribution from the Yenisei River to the Kara Sea (De Jonge et al., 2015). They interpreted this to be a preferential degradation of labile (riverine) 6-methyl brGDGTs and the enrichment in less labile (soil-derived) 5-methyl brGDGTs during transport (De Jonge et al., 2015). This suggests that only limited amounts of riverine 6-methyl

brGDGTs are transferred to the ocean, as also shown in other recent studies (Cao et al., 2022; Kirkels et al., 2022b). Such preferential degradation of 6-methyl brGDGTs over other brGDGTs could be attributed to variations in how these molecules are attached to soil particles (Huguet et al., 2008). Indeed, the higher degradation of 6-methyl brGDGTs upstream could be attributed to their different attachment to particles compared to downstream. The median diameter of the SPM was monitored

between February 2015 and June 2016 in both the upstream (sites 7 and 10) and downstream (sites 15 and 17) parts of the Seine Estuary (Druine, 2018). The particle size showed only slight dispersion (80-110 µm) under various hydrological conditions in the upstream estuarine section. The homogeneity in particle size in the upstream estuary likely reflects its predominantly continental origin (i.e. from the Seine River before the dam at Poses). In contrast, a large variability in the size of SPM particles was observed in the downstream estuary (15-20 µm to 80-90 µm), attributed to the complex flocculation and defragmentation processes of particles in this part of the estuary (Druine, 2018). Hence, the variability in the size of SPM particles from upstream to downstream could influence the distribution of brGDGTs in the Seine estuary.

In addition to this hypothesis, a shift in brGDGT distribution during downstream transport could be explained by mixing with autochthonous (i.e. estuarine-produced) brGDGTs (Crampton-Flood et al., 2021). The relative abundance of several brGDGTs (i.e. Ib, Ic, $IIIa_7$, $IIa_7$ and 1050d) in the downstream part of the Seine River basin is indeed significantly higher than the one in the upstream part ($p<0.05$, Wilcoxon test; Fig. 3), suggesting *in situ* brGDGT production in saltwater. Such a saltwater contribution can be visualized by the PCA based on brGDGT distribution, showing the positive score of the aforementioned compounds with the first axis (Fig. 4a). This axis is dominated by downstream samples influenced by seawater intrusion in the Seine Estuary (Fig. 4a).

It should be noted that brGDGT distributions in soils were roughly similar to those observed in downstream estuarine samples (SPM and river channel sediments) based on the PCA (Fig. 4a). Additionally, no significant differences were observed in the fractional abundances of several brGDGTs ($IIIb_6$, $IIb_6$, $IIIc_6$, $IIIa_7$, $IIa_7$, 1050d, $IIIa_5$, $IIIb_5$, $IIIb_7$, $IIIc_5$, $IIc_6$, and Ia) between soils and downstream samples (Fig. 3 and Fig. S4). This similarity in brGDGT distributions may be due to the influx of brGDGTs from the downstream soils into the downstream estuary, as 82% of the soils were collected downstream (Fig. 1a and Table 1). Hence, it cannot be excluded that brGDGTs detected in downstream estuarine samples are at least partly derived from soils of the watershed. Nevertheless, the soil-derived brGDGT contribution to the downstream estuarine samples is expected to be much lower than the autochthonous one, as the average brGDGT concentration in soils was ca. 3 times lower than the one in downstream estuarine (i.e. SPM and river channel sediment) samples (Fig. S5a).

In order to further assess whether downstream estuarine samples could be distinguished from soils, we applied the machine learning model (BigMac) developed by Martínez-Sosa et al. (2023) to our dataset with isoGDGT and brGDGT data as input. Most of our samples (SPM, sediments, and soils) were predicted as lake-type, with only one soil sample (soil6) collected at site B predicted as soil-type. This model suggests that, when considered altogether, the isoGDGT and brGDGT distributions are similar in aquatic and soil samples from the Seine estuary and differ from the soil-type samples described by Martínez-Sosa et al. (2023). Since the BigMac model does not include a river-type or estuary-type category (Martínez-Sosa et al., 2023), further inclusion of both isoGDGT and brGDGT data from global riverine/estuarine samples in the BigMac model may help to enhance predictions for river-type or estuary-type SPM and sediment samples.

The BigMac model distinguishes the type of samples using $IIa_6$ and crenarchaeol as the two most important predicting variables.
When accounting for both isoGDGTs and brGDGTs in the Seine River basin, the fractional abundance of crenarchaeol *vs.* total GDGTs (i.e. isoGDGTs + brGDGTs) varies significantly, whereas the one of $IIa_6$ does not differ significantly between the downstream estuary and soils (Fig. S8). Hence, the inclusion of isoGDGTs in the model may highly reduce the differences between sample types, as we observe significant differences in the fractional abundance of $IIa_6$ when calculated vs. total brGDGTs only (Fig. 3). As the BigMac model relies on both isoGDGT and brGDGT distribution, with no option of using
brGDGTs alone, we chose to perform an independent analysis to assess the similarity in brGDGT relative abundance between downstream SPM and sediment samples on the one hand and soils from the Seine River basin on the other hand. This model was developed using the same algorithm (random forest) as Martínez-Sosa et al. (2023). Binary classification (downstream estuary *vs.* soils) was performed based on fractional abundances of brGDGTs. The trained model (Fig. S9) indicated distinguishable brGDGT distributions between downstream estuary (SPM and sediments) and soil samples, supporting the *in*
*situ* production of brGDGTs in the downstream estuary. Although most of our soil samples were collected from the downstream estuary and showed similarity with the downstream SPM and sediment samples through PCA and comparison of fractional abundances, we were able to distinguish their brGDGT compositions using machine learning.

### 4.1.2 Environmental controls on the brGDGT distribution

As several individual brGDGTs are suggested to be preferentially produced either in the riverine or estuarine parts of the Seine
basin, their distribution might be related to ambient environmental factors. The RDA (performed on SPM samples) highlights the relationships between the available environmental variables (salinity, TN, TOC, water discharge and temperature) and the relative abundances of brGDGTs. Hierarchical partitioning indicates that salinity is the most important factor influencing the brGDGT distribution (14.97 %) in the Seine River basin (Fig. 5b and Table 2). Salinity is related to the relative abundances of compounds Ib, Ic, 7-methyl brGDGTs and the late-eluting homologs 1050d and 1036d that scored negatively on the first axis
of the RDA (Fig. 5a). This is in line with the positive significant correlations between salinity and the relative abundances of these compounds (Fig. S10). This trend also supports the assumption about the aquatic production of ring-containing tetramethylated brGDGTs (Ib and Ic) in Svalbard fjords which was thought to be linked to a salinity change (Dearing Crampton-Flood et al., 2019). The 7-methyl brGDGTs and their late-eluting isomers were also shown to be much more abundant in hypersaline lakes than those of lower salinity (Wang et al., 2021). Such a salinity-dependent brGDGT composition
has previously been interpreted by membrane adaptation to salinity changes or by a shift in bacterial community composition (Dearing Crampton-Flood et al., 2019; Wang et al., 2021). Hence, the significant positive correlations between salinity and these compounds in the Seine River basin suggest that brGDGT-producing bacteria have similar physiological mechanisms (i.e., membrane adaptation) to those reported in other aquatic settings (lakes and fjords) and/or that the diversity of these bacteria is changing along the river-sea continuum. The salinity proxy ($IR_{6+7me}$) proposed by Wang et al. (2021) does not show
significant correlations with salinity in this study ($p > 0.05$, Wilcoxon test; Fig. S10). This suggests that the $IR_{6+7me}$ index is

relatively insensitive in the Seine Estuary, potentially due to the preferential production of 6-methyl brGDGTs in specific estuarine regions (i,e. KP 255.6-337).

Indeed, the significant negative correlations between the salinity and the relative abundance of 6-methyl brGDGTs is observed in the Seine basin (Fig. S10), which suggests that the bacteria producing 6-methyl brGDGTs are preferentially present in the low salinity area of the estuary. To explore this further, we investigated the spatio-temporal variations of the 6-methyl $vs.$ 5-methyl brGDGTs ratio: $IR_{6Me}$ (Fig. 7). High $IR_{6Me}$ values ($0.69 \pm 0.10$) are associated with enhanced $in\ situ$ production of 6-methyl brGDGTs within the Yenisei river (De Jonge et al., 2015). In the Seine River basin, seasonal variation in $IR_{6me}$ is observed. Specifically, much higher $IR_{6Me}$ values are observed in a specific zone of the estuary ($260 < KP < 340$) with a low salinity range ($1.18 \pm 2.71$) during low-flow season (Fig. 7), suggesting that 6-methyl brGDGTs are preferentially produced in this zone when water discharge is low. Similarly, preferential production of 6-methyl brGDGT at low discharges was previously observed in other river systems, including the Amazon River basin (Kirkels et al., 2020; Crampton-Flood et al., 2021; Bertassoli et al., 2022) as well as Black and White Rivers (Dai et al., 2019). It was suggested that the enhanced 6-methyl brGDGT production at low flows was due to slow flow velocity and reduced soil mobilization. Although these hypotheses could account for the temporal variation in $IR_{6Me}$ in the Seine River basin, they are unlikely to explain the substantially high $IR_{6Me}$ values in this specific zone. Other environmental variables such as dissolved oxygen contents (Wu et al., 2021) and pH (De Jonge et al., 2014, 2015) were previously suggested to have a potential influence on 6-methyl brGDGT distributions. Nevertheless, these two environmental factors do not co-vary with $IR_{6Me}$ in the present study and can be ruled out as causes of variation in 6-methyl brGDGT distribution along the Seine river-sea continuum (Fig. 7). Hence, the production of 6-methyl brGDGTs in this zone of the Seine Estuary has to be triggered by other factors, such as the nutrient concentration.

High nutrient levels were shown to favor the production of 6-methyl versus 5-methyl brGDGTs in the water column of mesocosm experiments (Martínez-Sosa and Tierney, 2019). As the nutrient concentration is higher in the upstream part of the Seine estuary (Wei et al., 2022), and this zone is characterized by high proportions of agricultural land use (Flipo et al., 2021), the substantial production of 6-methyl brGDGT observed in the aforementioned zone ($260 < KP < 340$; Fig. 7) during low flows could be attributed to elevated nutrient levels, particularly nitrogen, resulting from intense agricultural activities. This is supported by the RDA triplot showing strong correlation of TN with the brGDGT distribution in the Seine basin, with the major 6-methyl brGDGTs (i.e. $IIa_6$ and $IIIa_6$) plotting close to TN in the RDA triplot (Fig. 5a). In addition, TN and $\delta^{15}N$ are observed to co-vary with $IR_{6Me}$ and to peak in the same zone ($260 < KP < 340$; Fig. 7) during the low-flow season. Nitrate from sewage effluents and manure are generally enriched in $^{15}N$ compared to other sources, leading to much elevated $\delta^{15}N$ values (10–25‰) (Leavitt et al., 2006; Andrisoa et al., 2019). Nutrients, in the form of nitrogen, can be concentrated at low discharges, thus triggering phytoplankton blooms (Romero et al., 2019). Hence, the elevated TN and $\delta^{15}N$ signals in a specific zone of the estuary ($260 < KP < 340$) could be attributed to the increase of nitrogen loadings and $^{15}N$-enriched nitrate uptake by phytoplankton developing intensively during the low-flow season. The much higher chlorophyll $a$ concentration in this

zone under low discharge conditions supports the hypothesis of phytoplankton blooms (Fig. 7). This high phytoplankton biomass might consequently create an environment that accelerates the growth and production of heterotrophic bacteria, which can in turn transform phytoplankton-derived organic matter (Buchan et al., 2014). As the brGDGT-producers were suggested to have a heterotrophic lifestyle (Weijers et al., 2010; Huguet et al., 2017; Blewett et al., 2022), they may transform phytoplankton-derived organic matter and thus participate in N-cycling during blooms. Hence, the co-variations of all the

parameters (IR$_{6Me}$, TN, $\delta^{15}$N, and Chl *a* concentration) peaking in the low salinity area during low-flow season suggest that low salinity range and high phytoplankton productivity represent favorable conditions for 6-methyl brGDGT production.

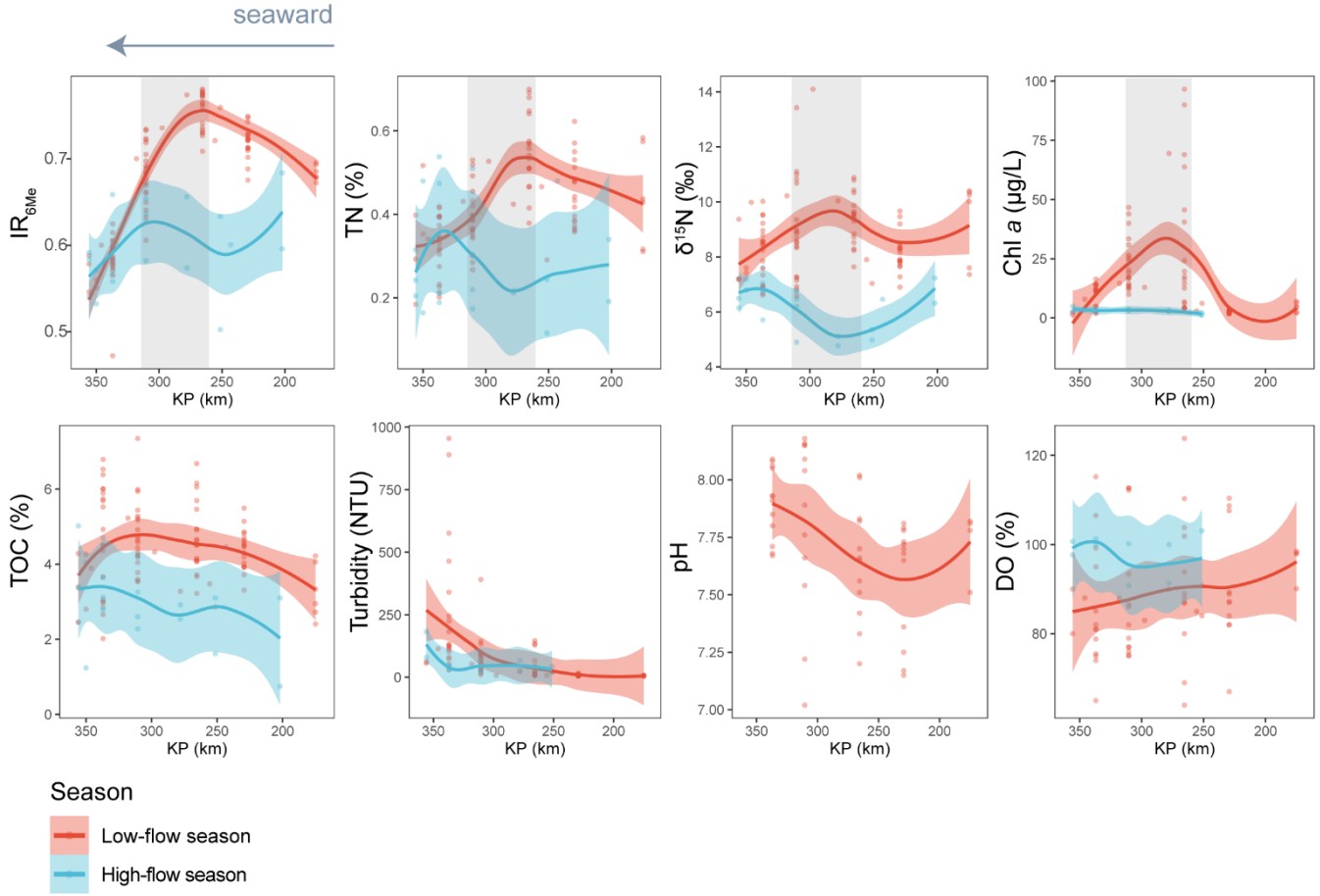

**Figure 7: Spatio-temporal variations of IR$_{6Me}$ and several environmental factors, including TN (%), $\delta^{15}$N (‰), Chla (µg/L), TOC (%), turbidity (NTU), pH, and dissolved oxygen saturation (DO, %). The trends showing variations were based on locally estimated**
**scatterplot smoothing (LOESS) method with 95% confidence intervals. KP (kilometric point) represents the distance in kilometers from the city of Paris (KP 0). Dataset is composed of SPM. The shaded area highlights a zone (KP 255.6-337) where IR$_{6Me}$ and several environmental parameters co-vary.**

### 4.2 Sources of brGMGTs and environmental controls on their distribution

**4.2.1 Sources of brGMGTs**

Similarly to the brGDGTs, the brGMGTs can also be produced *in situ* within the water column and/or sediments (Baxter et al., 2021; Kirkels et al., 2022a). In previous studies, brGMGTs were detected only in part of the soils surrounding the Godavari River basin (India; Kirkels et al., 2022a) and Lake Chala (East Africa; Baxter et al., 2021), suggesting a limited brGMGT production in soils in comparison to aquatic settings. Consistently, in the Seine River basin, concentrations of brGMGTs in

SPM and sediment samples are significantly higher than those in soils ($p<0.05$, Wilcoxon test; Fig. S5), pointing out their predominant aquatic source.

A notable compositional shift in brGMGT distribution is observed along the Seine River basin, as revealed by the separation of riverine, upstream and downstream estuarine samples in the PCA (Fig. 4b). The relative abundance of 2 brGMGTs (H1020c and H1034b) gradually decreases across the basin (Fig. 6) and is significantly correlated with those of 6-methyl brGDGTs

(Fig. S11). As 6-methyl brGDGTs are mainly produced in freshwaters in the Seine basin, this suggests that brGMGTs H1020c and H1034b and 6-methyl brGDGTs have a common freshwater origin and that the mixture of fresh and marine waters along the estuary leads to the dilution of these compounds during downstream transport. H1020c is the dominant brGMGT homologue in SPM from the riverine zone of the Seine and one of the most abundant brGMGT in the upstream part of the

estuary (Fig. 6). Such a trend was also observed in SPM and riverbed sediments from the upper part of the Godavari River basin, which was attributed to *in situ* riverine brGMGT production of this compound (Kirkels et al., 2022a).

The fractional abundance of H1020a and H1020b homologues gradually increases along the Seine River basin. This is consistent with the higher abundances of H1020a and H1020b previously reported in marine sediments from the Bay of Bengal

(Kirkels et al., 2022a). The predominance of these compounds in such samples was attributed to their *in situ* production in the marine realm. In line with this hypothesis, the relative abundances of brGMGTs H1020a and H1020b positively correlate with brGDGTs Ib, Ic, IIIa$_7$, IIa$_7$ and 1050d (Fig. S11) in the Seine Estuary, suggesting a similar marine origin.

**4.2.2 Environmental controls on the distribution of brGMGTs**

The current knowledge on the parameters controlling the brGMGT distributions in the terrestrial and marine realm is still

limited, as there is little literature available (Kirkels et al., 2022a). The correlations between the brGDGT and brGMGT relative abundances in the Seine River basin (Fig. S11) suggest that both types of compounds might be derived from overlapping source microorganisms, with common environmental factors controlling their membrane lipid composition. In the Seine River basin, salinity is shown to be the main environmental parameter influencing the brGMGT distribution, as also observed for brGDGTs (Fig. 5). This is reflected in the significant ($p<0.05$) increase in the relative abundances of homologues H1020a and

H1020b with salinity and a concomitant significant negative correlation between this parameter and the relative abundances

of homologues H1020c and H1034b ($p<0.05$, Wilcoxon test; Fig. 8, a-d). Nevertheless, the individual effect of TN on brGMGT relative abundances is observed to be much lower compared to that observed for brGDGTs (Fig. 5 and Table 2). This implies that, while having common controlling factors such as the salinity, they are also influenced by distinct parameters (i.e. TN), likely indicating distinct sources. This is consistent with a recent study showing that brGDGTs and brGMGTs likely originate from overlapping, but not identical origins (Elling et al., 2023).

The shift in brGMGT distribution observed across the Seine River basin (Fig. 4) could be due to a change in the diversity of brGMGT-producing bacteria and/or to an adaptation of these microorganisms to environmental changes occurring from upstream to downstream. The latter hypothesis seems unlikely, as a physiological adaptation of a given bacterial community would make it difficult to explain why the relative abundance of three isomers of compound H1020, which share a similar structure, varies differently in response to salinity changes. Hence, a shift in brGMGT-producing bacterial communities across the basin is more likely. Compounds H1020c and H1034b could predominantly be produced by bacteria preferentially growing in freshwater, and homologues H1020a and H1020b by bacteria preferentially living in brackish or saltwater.

## 4.3 Potential implications for brGMGTs as a proxy for riverine runoff in modern systems

The distinct brGMGT distributions in freshwater and saltwater could be used to trace the Organic Matter (OM) produced upstream all along the Seine basin. To trace such a riverine runoff signal, we propose a new proxy, the Riverine IndeX (RIX), based on the fractional abundances of brGMGTs H1020c and H1034b versus H1020a and H1020b (Eq. 6):

$$\text{RIX} = \frac{H1020c + H1034b}{H1020c + H1034b + H1020a + H1020b} \tag{6}$$

The rationale for RIX as a riverine runoff signal is that in freshwater environments, the pool of brGMGTs is dominated by H1020c and H1034b, whereas H1020a and H1020b prevail in saltwater environments. This is further supported by the significant negative correlation between RIX and salinity (Fig. 8e).

Since the other salinity proxies (i.e. ACE and $IR_{6+7Me}$) have shown positive correlations with salinity in previous studies (Turich and Freeman, 2011; Wang et al., 2021), they were expected to be positively correlated with salinity and negatively correlated with RIX in the Seine River basin. However, the ACE index (Turich and Freeman, 2011) and $IR_{6+7Me}$ (Wang et al., 2021) do not show significant correlations with salinity in the Seine River basin ($p>0.05$, Wilcoxon test; Fig. S10) and show weak but significant relations with the RIX (Fig. S13). This could be attributed to the influence of other factors than salinity on these indices (i.e. ACE and $IR_{6+7Me}$). Indeed, while ACE has been successfully applied in hypersaline systems (Turich and Freeman, 2011), it performs less effectively in certain saline settings due to the complex sources of archaeol and GDGTs (Huguet et al., 2015) and/or distinct ionization efficiencies between these compounds (He et al., 2020; Wang et al., 2021). Similarly, the

$IR_{6+7Me}$ may be influenced by the preferential production of 6-methyl brGDGTs related with nitrogen nutrient loadings in a specific region of the estuary (KP 255.6-337), as discussed in 4.1.2. Consequently, only the RIX successfully tracks salinity variations in this basin, while ACE and $IR_{6+7Me}$ show relative insensitivity. However, quantitatively reconstructing salinity with RIX is an important step forward that warrants further investigation. This requires comparing brGMGT distributions from various aquatic environments (e.g. estuaries and lakes) across salinity gradients.

The RIX was calculated for the SPM and sediment samples from the Seine River basin, showing an obvious decreasing trend from upstream to downstream (Fig. 8f). The RIX in river ($0.51\pm0.06$, SPM) and upstream estuarine ($0.40\pm0.07$, SPM and sediments) samples is significantly higher than for downstream estuarine ($0.23\pm0.06$, SPM and sediments) samples. RIX values around 0.50 could therefore be considered reflecting the riverine endmember, while those below 0.30 could represent the saltwater endmember. A significant overlap between brGMGT distribution soils and downstream samples was observed (Fig. 4b). This suggests that a portion of the brGMGT signal in the water masses of the Seine may be partially derived from surrounding soils. Hence, RIX was also calculated for the soil samples. The RIX values of the soil samples were $0.21\pm0.13$, close to those of the downstream estuarine samples. A large variance in the soil brGMGT concentration was observed (Fig. S5b), suggesting that further investigation is needed to better understand the environmental controls on the brGMGT production in soils. As with brGDGTs, we applied a random forest algorithm to distinguish brGMGT distributions between downstream estuary and soil samples. This trained model accurately distinguishes soils from downstream estuarine samples (Fig. S12), indicating in situ production of brGMGTs in the downstream estuary. Given the significantly low brGMGT concentrations in soils ($p<0.05$, Wilcoxon test; Fig. S5b) and the distinct distributions between brGMGT in soils and aquatic settings identified through PCA (Fig. 4) and machine learning (Fig. S12), it can be assumed that the impact of soil-derived brGMGTs on the observed RIX signal in the water column of the Seine basin is low.

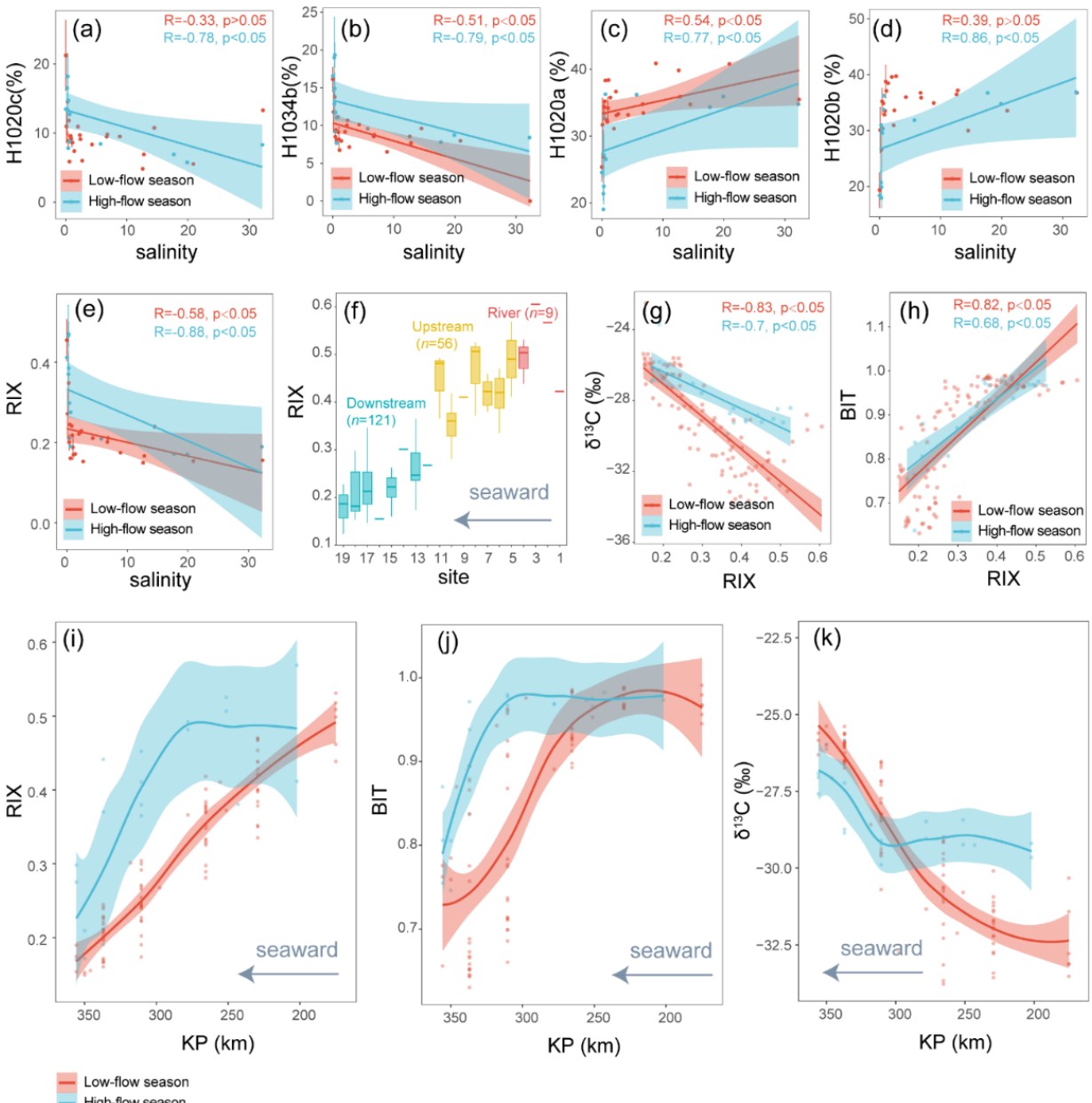

Figure 8: (a-e) The correlation between salinity and the relative abundance of brGMGTs and RIX, analyzed through linear regression. Shaded area represents 95% confidence intervals. Vertical error bars indicate mean ± s.d for samples with the same salinity. Dataset is composed of SPM. (f) Distribution of RIX across the Seine River basin. Boxes show the upper and lower quartiles of the data, and whiskers show the range of the data, which are color-coded based on the sample type (river in red, upstream estuary in yellow, and downstream estuary in blue). The center-line in the boxes indicates the median value of the dataset. Dataset is composed of SPM and sediments. (g-h) RIX plotted versus $\delta^{13}C$ and BIT through the linear regression. Shaded area represents 95% confidence intervals. (i-k) Spatio-temporal variations of RIX and several other terrestrial proxies, including BIT and $\delta^{13}C$ (‰). The trends showing spatio-temporal variations were based on locally estimated scatterplot smoothing (LOESS) method with 95% confidence intervals. KP (kilometric point) represents the distance in kilometers from the city of Paris (KP 0). Dataset is composed of SPM.

In order to test the general applicability of the RIX, it was then applied to soils, riverine and marine samples (SPM and sediments) collected in the Godavari River basin and Bay of Bengal (Kirkels et al., 2022a). This site represents the only other river-sea continuum besides the Seine basin for which brGMGT data are presently available. Significant differences in RIX between the soils, SPM and sediment samples from the Godavari River basin are observed ($p<0.05$, Wilcoxon test; Fig. 9). RIX values in soils (0.49±0.16) around the Godavari River basin are significantly lower than those the river samples ($p<0.05$,

Wilcoxon test; Fig. 9). Therefore, the potential soil contribution would dilute the riverine brGMGT signal, further decreasing the RIX in marine sediments. This is consistent with the observations in the Seine River basin. However, given the distinct distributions between soil and aquatic samples and the lower brGMGT concentration in soils (Kirkels et al., 2022a), the influence of soil-derived brGMGTs on riverine RIX values may be limited. In addition, 96% of the RIX values in riverine SPM and riverbed sediments from the Godavari basin exceed 0.5, whereas all of the RIX values observed in marine sediments

from the Bay of Bengal are below 0.3. This suggests that the RIX cutoff values defined using the samples from the Seine basin may be broadly applicable and valid across other river-sea continuums. This deserves further studies.

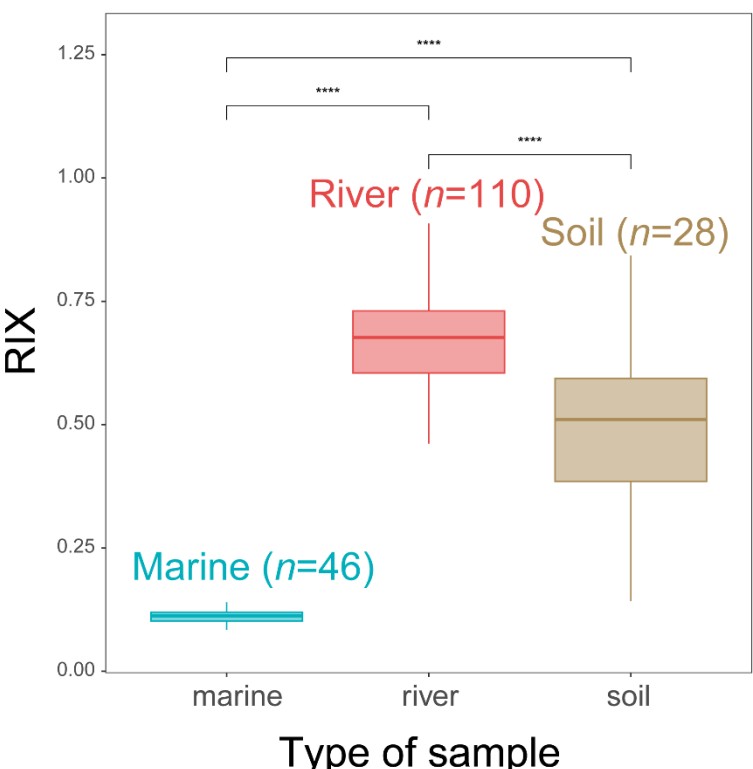

**Figure 9: RIX in the soils, SPM and sediment samples from Godavari River basin (India) and Bay of Bengal sediments (data from Kirkels et al. (2022a)). Statistical testing was performed by a Wilcoxon test. Boxes show the upper and lower quartiles of the data,**
**and whiskers show the range of the data, which are color-coded based on the sample type (river in red, marine in blue, and soil in brown). The center-line in the boxes indicates the median value of the dataset.**

Further confirmation of the RIX potential as a tracer of riverine OM comes from the significant correlations observed between this index and other commonly used proxies for tracing OM sources, i.e. the BIT and $\delta^{13}C_{org}$ ($p<0.05$, Wilcoxon test; Fig. 8, g-h). These proxies show roughly similar spatial and temporal variations in the Seine River basin. It is worth noting that another terrestrial proxy (C/N) was not included because it may be ineffective in tracing terrestrial OM in this anthropogenic estuary.

In the low-flow season, RIX and BIT gradually decrease while $\delta^{13}C_{org}$ increase across the basin (Fig. 8, i-k). Such trends during the low discharge periods likely reflects the continuous dilution process of riverine OM caused by the mixing of fresh and marine water masses (Thibault et al., 2019). The gradual dilution of the riverine OM signal along the Seine River basin could be due to the increase of seawater intrusion, and thus marine-derived OM, at low discharges (Ralston and Geyer, 2019; Kolb et al., 2022). In contrast, during the high-flow season, no such gradual dilution trend is observed. Instead, at high discharges, the RIX, BIT and $\delta^{13}C_{org}$ remain roughly stable from KP 202 to 310.5, before, steeply decreasing for BIT index and RIX, and increasing for $\delta^{13}C_{org}$. This trend can be explained by the fact that at high flow rates, the limit of saltwater intrusion in the estuary shifts seawards rather than landwards, allowing the riverine OM to be flushed further downstream than under low discharge conditions. After this region, the riverine OM is diluted because of the mixing with marine water masses, as observed during the low-flow season. The trends observed in the Seine Estuary are consistent with previous studies in other regions, showing that terrestrial OM was only effectively transported downstream at high flow rates (Kirkels et al., 2020, 2022b).

Although the BIT is successfully used in the Seine River basin as well as in previous studies to trace riverine (terrestrial) OM inputs (Hopmans et al., 2004; Xu et al., 2020), this index can be biased by *in situ* production of brGDGTs in the water column and/or sediments (Sinninghe Damsté, 2016; Dearing Crampton-Flood et al., 2019) and selective degradation of crenarchaeol *vs.* brGDGTs (Smith et al., 2012). Hence, high BIT values do not necessarily indicate higher contribution of terrestrial OM in some settings (Smith et al., 2012). Unlike the BIT index, based on two different families of compounds (isoGDGTs and brGDGTs), the RIX is based on 4 compounds from the same family (brGMGTs) that likely have similar degradation rates and therefore not influenced by selective degradation. Furthermore, the RIX is based on the relative abundances of abundant brGMGTs which are all predominantly produced in aquatic settings, two of them (H1020c and H1034b) being mainly produced in freshwater and two of them (H1020a and H1020b) mainly in saltwater. Therefore, the RIX is based on compounds which are more specifically produced in the two endmembers (freshwater or saltwater), which could avoid the biases encountered with the BIT. Overall, our work shows that, in addition to the BIT and $\delta^{13}C_{org}$, the RIX successfully captures the spatio-temporal dynamics of riverine OM in the Seine River basin, making this proxy a promising and complementary one tracing riverine runoff in modern samples.

**4.4 Application of the RIX to a paleorecord across the upper Paleocene and lower Eocene**

To further test the applicability of the RIX as a riverine proxy in paleorecords, we calculated RIX and compared this proxy with BIT using the published brGMGT and GDGT data (Sluijs et al., 2020) from IODP Expedition 302 Hole 4A (cf. location

in Fig. S14) across the upper Paleocene and lower Eocene. This core is considered to record significant changes in terrestrial inputs (i.e. continental spores and pollen) due to sea level changes over time (Sluijs et al., 2009, 2006), making it a suitable paleorecord for testing runoff proxies.

In the late Palaeocene, the relative abundance of terrestrial palynomorphs (spores and pollen) remains at high levels (Fig. 10a), indicating enhanced terrestrial inputs during this period. This is consistent with the presence of abundant amorphous organic matter in this interval that is presumed to have originated from terrestrial sources (Sluijs et al., 2006). Enhanced freshwater contribution is also successfully captured by RIX and BIT (Fig. 10b), with values exceeding 0.3 for both proxies.

Palynomorph assemblages from the body of Paleocene-Eocene Thermal Maximum (PETM) are characterized by substantially lower terrestrial palynomorphs (Fig. 10a). This indicates a relative decrease of riverine runoff, which is also evidenced by a drop of both RIX and BIT indices (Fig. 10b). Such decreased runoff during the PETM body was previously attributed to a rise in sea level (Sluijs et al., 2006), which has been recorded in many other sites worldwide (Speijer and Morsi, 2002; Harding et al., 2011; Sluijs et al., 2014). During the PETM recovery, increased runoff is reflected by the gradual increase in relative abundance of terrestrial palynomorphs (Fig. 10b), which was interpreted as a consequence of a drop in sea level (Sluijs et al., 2006). Higher freshwater influx during this period is also indicated by increases in RIX and BIT (Fig. 10b).

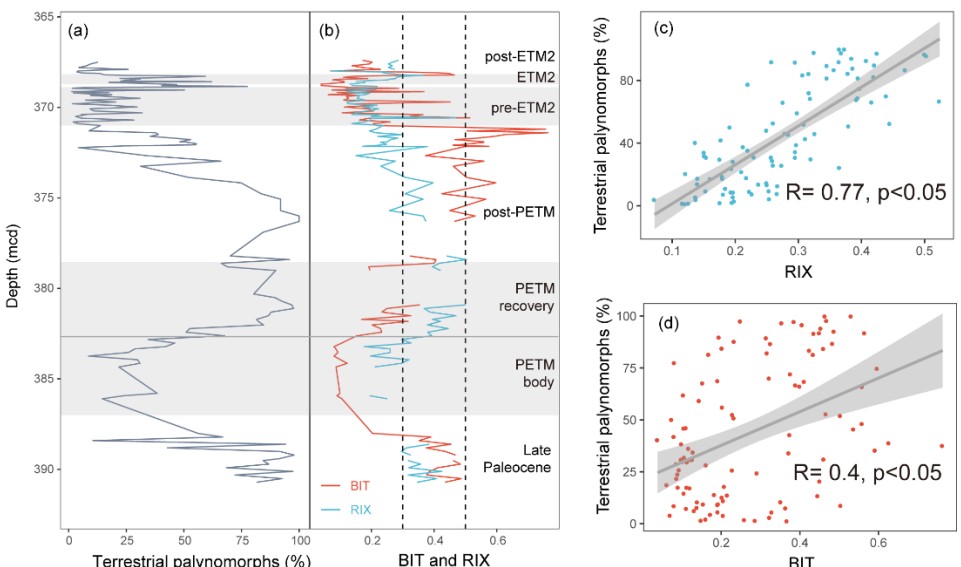

**Figure 10: Comparison between (a) terrestrial palynomorphs (%) and (b) BIT and RIX across the upper Paleocene and lower Eocene between 391 and 367 meters composite depth below sea floor (mcd) of IODP Expedition 302 Hole 4A. Terrestrial palynomorphs data are from Sluijs et al. (2006) and Sluijs et al. (2009). RIX and BIT were calculated using data from Sluijs et al. (2020). Grey shading represents Eocene Thermal Maximum 2 (ETM2), pre-ETM2 interval, and Paleocene-Eocene Thermal Maximum (PETM). Dotted line represents cutoff values of RIX (below 0.3 for marine contribution and above 0.5 for riverine contribution). Linear regression of the RIX (c) and BIT (d) against the terrestrial palynomorphs. Shaded area represents 95% confidence intervals.**

The relative abundance of terrestrial palynomorphs decreases after the termination of the PETM (Fig. 10a). Similar decreasing trends are also evident for RIX during post-PETM (Fig. 10b). However, BIT levels remain high until the pre-Eocene Thermal Maximum 2 (ETM2), showing distinct patterns compared to terrestrial palynomorphs and RIX. One possible explanation for this distinction could be the sedimentary *in situ* production of brGDGTs (Peterse et al., 2009). Indeed, several studies have indicated that elevated BIT values may not accurately represent high levels of terrestrial inputs in marine environments (Smith et al., 2012). In contrast, the similar trends observed between RIX and terrestrial palynomorphs highlight the reliability of RIX as a valuable complementary runoff proxy (Fig. 10b).

At the onset of the pre-ETM2, there is a significant decrease in terrestrial palynomorphs (Fig. 10a). Meanwhile, there is a sharp increase in the proportions of normal marine dinocysts, which was interpreted as a transgressive signal (Sluijs et al., 2008; Willard et al., 2019; Sluijs et al., 2009). Throughout the pre-ETM2 interval, the relative abundance of terrestrial palynomorphs remains consistently low (Fig. 10a). Additionally, the dinocyst assemblages suggest normal marine conditions for this period (Sluijs et al., 2009). These normal marine conditions are also well documented by RIX (Fig. 10b), as most of the samples demonstrate low values (below 0.3). In contrast, BIT values exhibit some fluctuation, with several samples displaying high values (Fig. 10b). This variability in BIT values could also be attributed to the sedimentary *in situ* production of brGDGTs.

During the ETM2 interval, increase of terrestrial palynomorphs suggests an increased runoff from the continent to the site (Fig. 10a). Enhanced runoff signal during ETM2 is also supported by the dominance of low-salinity dinocyst taxa and the presence of massive amorphous organic matter (Sluijs et al., 2009; Willard et al., 2019). Both RIX and BIT show increasing trends for this interval (Fig. 10b), indicating that both indices are reflecting a runoff signal during this period.

Following the ETM2, there is a decline in the relative abundance of terrestrial palynomorphs (Fig. 10a), which indicates a shift toward normal marine conditions. This shift is also supported by the dominance of normal marine dinocysts and low concentrations of massive amorphous organic matter (Sluijs et al., 2009; Willard et al., 2019). Additionally, this shift towards normal marine conditions is in line with the lower values (below 0.3) observed for both RIX and BIT (Fig. 10b).

Overall, RIX and BIT exhibit similar trends with terrestrial palynomorphs in the late Palaeocene, PETM, ETM2, and post-ETM2 periods. Both lipid proxies are likely reliable indicators of riverine runoff for these intervals. However, distinctions between RIX and BIT become more apparent especially in the post-PETM and pre-ETM2 periods when normal marine conditions prevail, and sedimentary *in situ* production of brGDGTs may occur, resulting in high BIT values. RIX proves to be particularly valuable for these intervals, avoiding the possible biases associated with BIT. This indicates that RIX performs better in this core compared with BIT, which is further supported by a higher correlation coefficient observed between RIX and terrestrial palynomorphs (0.77; Fig. 10c) compared with BIT and terrestrial palynomorphs (0.4; Fig. 10d).

**5 Conclusions**

In this study, the brGDGT and brGMGT concentrations and distributions in soils, SPM and sediments ($n$=237) across the Seine River basin were investigated. Higher concentrations and distinct distributions of brGDGTs and brGMGTs in SPM and sediments compared with soils imply that both types of compounds can be produced *in situ* in aquatic settings. The distribution of both brGDGTs and brGMGTs are largely related to salinity, but only brGDGT distributions are significantly influenced by nitrogen nutrient loadings. In addition, covariations of $IR_{6Me}$, TN, $\delta^{15}N$, and Chl *a* concentration within the low salinity region

suggest that riverine (6-methyl) brGDGT production is favored by low-salinity and high-productivity conditions.

    In the Seine River basin, salinity correlates positively with H1020a and H1020b, and negatively with H1020c and H1034b. This indicates that compounds H1020c and H1034b could be produced by bacteria that preferentially grow in freshwater, while homologues H1020a and H1020b could be produced by bacteria that mainly live in saltwater. Based on this, a novel proxy,

the Riverine IndeX (RIX) is proposed to trace riverine OM input. The average value of RIX for the riverine samples is 0.51, which is much higher than that in downstream estuarine (0.23 on average) samples. This suggests that RIX values over 0.5 imply considerable riverine contributions, whereas RIX values below 0.3 indicate higher marine contributions. This cutoff value defined in the Seine River basin also works in the Godavari River basin (India), which implies that this novel proxy based on brGMGTs may be broadly applicable in modern samples. Finally, RIX was applied to a paleorecord across the upper

Paleocene and lower Eocene and showed similar trends with terrestrial palynomorphs. Altogether, this showed the potential of the RIX as a riverine runoff proxy in modern and deep times. Additional studies are needed to test its general applicability in modern samples and paleorecords.

## Data availability

The brGDGT and brGMGT data will be archived in PANGAEA by the time of publication.

## Author contribution

ZXZ, EP, and AH conceptualized this study. AC conducted LC-MS analyses. JM and AML collected and analyzed wetland soils and mudflat sediments. ZXZ, AH, EP, AT, and EB participated in the fieldwork. ZXZ performed the laboratory work, data analyses and wrote the article in consultation with all the other authors.

## Competing interests

The authors declare that they have no conflict of interest.

## Acknowledgments

We would like to thank members of the SARTRE project for their assistance during the field campaigns. We appreciate the logistical support from Robert Lafite, Frédéric Azémar, Anaëlle Bernard and Mahaut Sourzac. We are grateful to Pascal Claquin and Léo Chasselin for their support in water quality measurements and Micky Tackx for the scientific coordination of the SARTRE project. This study was funded by GIP Seine-Aval (MOSAIC, PHARESEE, FEREE and SARTRE projects) and CNRS/INSU and OFB (EC2CO RUNTIME project). Zhe-Xuan Zhang's PhD scholarship is funded by China Scholarship Council (No. 202004910406) and ASDB from Sorbonne Université. This work was supported by CESAM (UIDP/50017/2020 + UIDB/50017/2020 + LA/P/0094/2020). We would like to thank Vincent Grossi for helpful discussion. We also thank Appy Sluijs for providing the palynological data for the IODP Expedition 302 Hole 4A.

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
