# Peer review of "Environmental controls on the brGDGT and brGMGT distributions across the Seine River basin (NW France): Implications for bacterial tetraethers as a proxy for riverine runoff"

_EGUsphere, 2023_

## Author Comment (AC1)

Response to comments by reviewer #1

In this manuscript, Zhang et al propose a new proxy to reconstruct fluvial organic matter inputs to coastal marine settings. They suggest that brGMGTs are produced in-situ in rivers and estuaries and that the distribution of brGMGTs is principally controlled by salinity. Based on these facts they generate a new Riverine Index (RIX) using the fractional abundances of H1020c and H1034b versus H1020a and H1020b. To validate the RIX in deep time they compare RIX values to the BIT index and terrestrial pollen/spore deposits deposited during the PETM from IODP Expedition 302 Hole 4A. They report a closer relationship between RIX and terrestrial pollen abundance than BIT and terrestrial pollen abundance, indicating that at least in this site RIX outperforms BIT in accurately reconstructing riverine inputs. In all, this is an interesting study that will likely be of interest to BG readers. I have a number of comments that aim to strengthen the manuscript.

We would like to thank the reviewer for all the constructive comments and the positive assessment on the significance of our manuscript. A point-by-point reply to all the reviewer's comment is provided below and is colored blue. The text which would be added into the revised manuscript is shown in orange italics.

General comments:

In some sections (see specific comments) the use of English in the paper is poor and obfuscates the meaning of the text. I suggest that the authors carefully read through the manuscript to catch all typos and grammatical errors. Likewise figure quality varies considerably. In some cross plots, it is impossible to see the data as the marker size is so small (see specific comments). Characters that should be superscripted/subscripted are left as regular text (see specific comments). Lines of best fit are drawn through data but there is no information as to how these lines were constructed (see specific comments). As such, this manuscript would benefit greatly from more attention to detail from the authors.

In a revised version, we would carefully check the manuscript for any typos and grammatical errors to improve its readability and clarity. Regarding the figure quality, we would address all the issues highlighted by the reviewer, including marker size, superscripting/subscripting, and would provide detailed information (e.g. lines of best fit) in the figure caption.

Additionally, as the authors are proposing a new GDGT salinity index, I would like them to calculate and report previously formulated salinity indices from their samples. Specifically, the ACE index (Turich and Freeman 2011) and the IR6+7me (Wang et al 2021) both have been calibrated against water salinity in marine saline ponds and hypersaline lakes respectively. I know the author's brief touched on comparing IR6me from this study to values from Wang et al (2021) in the text but a more thorough examination of prior GDGT-derived water salinity reconstructions would strengthen the manuscript. Readers will be interested to see how these indices compare against RIX in reconstructing salinity from an estuarine environment.

We agree with the suggestion by the reviewer. In a revised version, we would calculate the ACE and $IR_{6+7me}$ indices and would present them in a supplementary figure (presented below), allowing the comparison of the corresponding values with those of the RIX. The ACE and $IR_{6+7me}$ indices do not correlate with salinity in the Seine River basin.

[Figure]

*Supplementary figure. Salinity plotted versus ACE, $IR_{6+7Me}$, relative abundance of 6-methyl and 7-methyl brGDGTs ($IIIa_6$, $IIa_6$, $IIb_6$, $IIIa_7$ and $IIa_7$) as well as compounds 1050d, 1036d, Ib, and Ic through the linear regression. Shaded area represents 95% confidence intervals. Vertical error bars indicate mean ± s.d for samples with the same salinity. Dataset is composed of SPM.*

We would add the following text in the revised manuscript:
*"Compared with RIX, other salinity proxies, including the ACE index (Turich and Freeman, 2011) and $IR_{6+7Me}$ (Wang et al., 2021), do not show significant correlations with salinity in the Seine River basin (p>0.05, Wilcoxon test; Fig. S7). Although ACE has been successfully applied in hypersaline systems (Turich and Freeman, 2011), it performs less effectively in certain saline settings due to the complex sources of archaeol and GDGTs (Huguet et al., 2015) and/or distinct ionization efficiencies between these compounds (He et al., 2020; Wang et al., 2021). Similarly, $IR_{6+7Me}$ may be influenced by the preferential production of 6-methyl brGDGTs related with nitrogen nutrient loadings in a specific region of the estuary, as discussed in 4.1.2. Consequently, only RIX successfully tracks salinity variations in this basin, while ACE and $IR_{6+7Me}$ show relative insensitivity."*

Additionally, the evidence for in situ production of brGDGTs and brGMGTs in downstream estuary sites is pretty weak. This is demonstrated by Fig 2 where we see that distributions of d13Corg and d15N in soils and downstream estuary samples are very similar in addition to Fig 5 where your PCA on sample brGDGT and brGMGT distributions cannot separate out soils from downstream estuary samples. Yes you see (on average) higher concentrations of brGDGTs and brGMGTs in downstream estuary samples than in soils but the actual distributions of brGDGT and brGMGT abundance in soils are pretty large, indicating that some soils have pretty substantial quantities of these compounds. A great way to add more clarity to this sourcing issue is to train a random forest model using a similar method to Martinez-Sosa et al (2023)

on your brGDGT and brGMGT samples (and isoGDGTs as these should be available to you). If the random forest model can accurately separate out soils from downstream estuary samples then you can be pretty sure that your downstream estuary samples were produced in situ. This won't require much additional work and can be implemented easily using python (https://scikit-learn.org/) or another language of your choice.

We thank the reviewer for this suggestion. In a revised manuscript, we would use a similar approach as the one proposed by Martinez-Sosa et al. (2023) by applying a random forest model to our brGDGT and brGMGT datasets. As the objective of our manuscript is to investigate the sources and controlling factors of brGDGTs and brGMGTs, we would use brGDGTs and brGMGTs separately for training the machine learning model. The application of this model to brGDGT and brGMGT datasets accurately separates downstream (SPM and sediment) estuarine samples from soil ones, indeed supporting *in situ* production of these lipids in downstream Seine Estuary.

In the material and methods, we would add a new machine-learning section describing the model, as follows:

*"Our lipid dataset was split into a training set (75%) and a test set (25%). We then used a supervised machine-learning algorithm (random forest) to train models. This algorithm was applied to classify the downstream estuary and soil samples based on brGDGTs or brGMGTs as input, implemented using the scikit-learn library (https://github.com/scikit-learn/) (Pedregosa et al., 2011) in Python (version 3.10.12). Hyperparameter tuning was conducted using a randomized search approach implemented through the RandomizedSearchCV function in scikit-learn.*

*SHapley Additive exPlanations (SHAP) is a game-theoretical method used to interpret machine learning models (Lundberg et al., 2020). SHAP analysis was applied to identify which compounds were important for the classifications, implemented by the SHAP library in Python. A higher SHAP value indicates a more substantial contribution of the feature (brGDGTs or brGMGTs) to the predicted outcome (downstream estuary or soils)."*

Two figures (one for brGDGTs, another for brGMGTs) showing the performance of the model would be added to the supplement:

[Figure]

*Supplementary Figure (brGDGTs). Evaluation of the random forest model based on brGDGTs through the confusion matrix (a), classification report (b), and receiver operating characteristic (ROC) curve (c). SHAP summary plots (d-e) show the feature importance obtained from the random forest algorithm and the SHAP library. Each bullet in the plot represents a single sample in the training set, with the color indicating the feature value (fractional abundance of the brGDGTs) from low (blue) to high (pink). The bullets positioned on the right side of the SHAP summary plot correspond to positive SHAP values, indicating a positive effect on the model output (downstream estuary or soils). The bullets on the left side of the plot indicate negative SHAP values, suggesting a negative effect on the model output. The variables (brGDGTs) with higher impact on the model performance are shown at higher positions. Training sets include downstream SPM and sediment samples (d) and soils (e).*

[Figure]

*Supplementary Figure (brGMGTs). Evaluation of the random forest model based on brGMGTs through the confusion matrix (a), classification report (b), and receiver operating characteristic (ROC) curve (c). SHAP summary plots (d-e) show the feature importance obtained from the random forest algorithm and the SHAP library. Each bullet represents a single sample within the training set, with the color representing the feature value (fractional abundance of the brGMGTs) ranging from low (blue) to high (pink). The bullets positioned on the right side of the SHAP summary plot correspond to positive SHAP values, indicating a positive effect on the model output (downstream estuary or soils). The bullets on the left side of the plot indicate negative SHAP values, suggesting a negative effect on the model output. The variables (brGDGTs) with higher impact on the model performance are shown at higher positions. The training sets include downstream SPM and sediment samples (d) as well as soils (e).*

We would add the following text in a revised manuscript to describe and discuss the results related to the application of the model to the brGDGT dataset:

*"Martínez-Sosa et al. (2023) showed the efficiency of random forest algorithm to distinguish the origin of GDGTs from a global dataset comprised of soil, peat, marine and lake samples. Consequently, we chose to use this algorithm to train and classify brGDGTs from downstream estuary and soil samples in the Seine River basin. Binary classification (downstream estuary vs. soils) was performed using a random forest model based on fractional abundances of brGDGTs. This trained model (Fig. S8) indicated distinguishable brGDGT distributions between downstream estuary (SPM and sediments) and soil samples, supporting in situ production of brGDGTs in the downstream estuary."*

We would also add the following text in a revised manuscript to describe and discuss the results related to the application of the model to the brGMGT dataset:

*"As with brGDGTs, we applied a random forest algorithm to distinguish brGMGT distributions between downstream estuary and soil samples. This trained model accurately distinguishes soils from downstream estuarine samples (Fig. S11), indicating in situ production of brGMGTs in the downstream estuary. Given the significantly low brGMGT concentrations in soils ($p<0.05$, Wilcoxon test; Fig. S5b) and the distinct distributions between brGMGT in soils and aquatic settings identified through PCA (Fig. 4) and machine learning (Fig. S11), it can be assumed that the impact of soil-derived brGMGTs on the observed RIX signal in the water column of the Seine basin is low."*

Last, isoGDGT data could be discussed in our future work, but this is beyond the scope of the present manuscript.

Specific comments

Line 35: This complicates paleoenvironmental interpretations in SOME aquatic settings not ALL aquatic settings
We agree with this suggestion, this would be corrected.

Line 37: "all along this basin, from land to sea" awkward phrasing
We would rephrase this sentence as follows:

*"BrGDGTs and brGMGTs were analyzed in soils, Suspended Particulate Matter (SPM), and sediments (n=237) collected along the land-sea continuum of the Seine basin."*

Line 40: "Redundancy analysis further shows that both salinity and nitrogen loadings dominantly control the brGDGT distributions." No, the loadings indicate that SALINITY (not salinity loadings) controls the brGDGT distribution.
This would be corrected.

Line 40-43: "Furthermore, the relative abundance of 6- methyl vs. 5-methyl brGDGTs (IR6Me ratio), Total Nitrogen (TN), δ 15N and chlorophyll a concentration co-vary in a specific zone with low salinity"
Is this zone geographical, in your redundancy analysis, or something else?
This zone is geographical. We would propose the following sentence in a revised manuscript:
*"Furthermore, the relative abundance of 6-methyl vs. 5-methyl brGDGTs (IR$_{6Me}$ ratio), Total Nitrogen (TN), $\delta^{15}N$ and chlorophyll a concentration co-vary in a specific geographical zone with low salinity,*

*suggesting that 6-methyl brGDGTs are preferentially produced under low-salinity and high-productivity conditions."*

Line 44-45: "Salinity is positively correlated with homologs H1020a and H1020b, 45 and negatively correlated with compounds H1020c and H1034b." Is this in soils, sediments or SPM?
This correlation was found in SPM. This would be specified as follows:
*"Salinity is positively correlated with homologues H1020a and H1020b, and negatively correlated with compounds H1020c and H1034b in SPM."*

Line 45: "This suggests that bacteria thriving…" thriving is not the correct word (carries implications of a value judgment) replace with "living".
*This would be corrected.*

Line 45-47: It seems like you aren't mentioning results from soils and sediments, only from SPM? Or maybe all your sediment samples are exclusively from rivers? The reader is unclear on this.
Correlation between salinity and lipid distribution is based on SPM samples. We would modify the abstract as follows:
*"Both types of compounds (i.e. brGDGTs and brGMGTs) are shown to be produced in situ, in both freshwater and saltwater, based on their high concentrations and distinct distributions in aquatic settings (SPM and sediments) vs. soils.*

Line 51-52: "a paleorecord across the upper Paleocene and lower Eocene," You should name this record and say where it is.
We would add the name (Arctic Coring Expedition) and location (Lomonosov Ridge) of this record.

Lines 51: "showing its potential applicability in both modern samples and in paleorecords." Perhaps you could evaluate its usage in both these cases e.g. - we successfully/unsuccessfully applied RIX in …
We would rephrase this sentence as follows:
*"We successfully applied RIX to the Godavari River basin (India) and a paleorecord across the upper Paleocene and lower Eocene from the Arctic Coring Expedition at Lomonosov Ridge, showing its potential applicability in both modern samples and in paleorecords."*

Line 55: ", although some of them were attributed to the phylum Acidobacteria" Imprecise language.
Thank you for the comment. We would rephrase this sentence into:
*"Branched glycerol dialkyl glycerol tetraethers (brGDGTs) are membrane lipids produced by bacteria, some of them belonging to the phylum Acidobacteria."*

Line 57-58: "The distribution of brGDGTs (number of cyclopentane moieties and methyl groups; cf. structures in Fig. S1) was empirically linked with pH and Mean Annual Air Temperature" Again, imprecise language. The phrase "empirically linked" doesn't convey much useful information.
We would replace "empirically" by "has been":
*"The distribution of brGDGTs (number of cyclopentane moieties and methyl groups; cf. structures in Fig. S1) has been linked with pH and Mean Annual Air Temperature (MAAT) in soils."*

Line 60: Should really cite some earlier lake GDGT papers in addition to Martinez-Sosa et al., 2021.

Thank you for this suggestion. We would cite some earlier work: Tierney et al., 2010 GCA and Russell et al., 2018 OG.

Lines 60-61: "The brGDGT-based proxies (i.e. MBT'5ME and CBT') have been largely applied to reconstruct MAAT and pH from sedimentary archives (Coffinet et al., 2018; Harning et al., 2020; Wang et al., 2020)." Not quite true - Martinez-Sosa et al (2021) and Dearing Crampton-Flood et al (2020) generated Bayesian linear regressions between the Mean temperature of months above freezing and MBT'5me. These BayMBT models have been used widely in the community since their publication.

We agree that the new models were not applied to records available before their publication. We would modify this sentence as follows:

*"The brGDGT-based proxies (i.e. MBT'$_{5ME}$ and CBT') have been largely applied to reconstruct paleoclimate from sedimentary archives (Coffinet et al., 2018; Harning et al., 2020; Wang et al., 2020)."*

Line 62-63: "In aquatic settings, brGDGTs were initially suggested to be predominantly derived from watershed soils and transported by erosion in the sediments (Hopmans et al., 2004)." Maybe you mean "transported by erosion to the sediments"?

We agree and would rephrase this sentence as follows:

*"In aquatic settings, brGDGTs were initially suggested to be transported by erosion to the sediments."*

Lines 63-78: The use of English throughout this paragraph is poor and hard to follow. Needs copyediting.

We would rephrase this paragraph as follows:

*"In aquatic settings, brGDGTs were initially suggested to be transported by erosion to the sediments (Hopmans et al., 2004). Based on this assumption, the Branched and Isoprenoid Tetraethers (BIT) index was defined as the abundance ratio of the major brGDGTs to crenarchaeol (isoprenoid GDGT mainly produced by marine Nitrososphaerota). The BIT index ranges between 0 and 1, with high BIT values (around 1) reflecting a higher contribution of terrestrial organic matter compared to marine organic matter (Hopmans et al., 2004).*

*Over the last few years, the BIT index has been broadly used to quantify the relative contribution of terrestrial organic matter in aquatic systems (Xu et al., 2020; Yedema et al., 2023) and to evaluate the reliability of the TEX$_{86}$ palaeothermometer (Cramwinckel et al., 2018). However, several studies have shown that brGDGTs can also be produced in situ in aquatic settings, including rivers (e.g., De Jonge et al., 2015; Freymond et al., 2017; Kim et al., 2015; Zell et al., 2014, 2013), lakes (Tierney and Russell, 2009), and marine environments (Dearing Crampton-Flood et al., 2019; Zeng et al., 2023). This adds complexity to the identification of brGDGT sources in aquatic ecosystems and to the application of brGDGTs as (paleo)environmental proxies, including the BIT index.*

*The BIT values have all the more to be carefully interpreted, especially considering the potential influence of the selective degradation of branched vs. isoprenoid GDGTs (Smith et al., 2012). Thus, complementary molecular proxies for quantifying the input of terrestrial organic matter to aquatic settings are still needed. These proxies may cross-validate other available terrestrial proxies, such as the $\delta^{13}C$ of organic carbon (Lamb et al., 2006), heterocyst glycolipids (Kang et al., 2023), and long-chain diols (Lattaud et al., 2017)."*

Lines 63-78: You should read and cite Martinez-Sosa et al (2023) here for their work on a Random Forest

approach to classify GDGT sources (i.e. Marine, Soil, Lake etc).

*Thank you for this suggestion. We would refer to the work by Martínez-Sosa et al. (2023) as follows:*

*"Recently, a machine-learning approach (BIGMaC model) was proposed to infer the origin of environmental samples (e.g. soil, peat, marine and lake settings) based on their GDGT distribution (Martínez-Sosa et al., 2023). While such an approach shows potential for differentiating distinct sources of GDGTs, its application to aquatic systems has not yet been extensively explored."*

Line 80-83: "The improvement of analytical methods allowed the separation and quantification of 5-, 6- and 7-methyl brGDGTs (methyl groups at the fifth, sixth, and seventh positions; Fig. S1), that in previous chromatographic protocols co-eluted (De Jonge et al., 2013, 2014; Ding et al., 2016)." No real link between the previous paragraph and this one. Also, which methods? How did they improve?

This comment would be taken into account as follows:

*"The improvement of chromatographic methods allowed the separation and quantification of 5-, 6- and 7-methyl brGDGTs (methyl groups at the fifth, sixth, and seventh positions; Fig. S1) that previously co-eluted (De Jonge et al., 2013, 2014; Ding et al., 2016). This led to the development of new brGDGT-based proxies based on these specific brGDGT isomers (De Jonge et al., 2014)."*

Lines 86-87: "In addition to temperature and pH, other environmental factors may influence brGDGT distributions in terrestrial and aquatic settings and hence the application and interpretation of brGDGT-derived proxies" This is a repetition from earlier in the introduction.

*This sentence would be removed from the revised manuscript.*

Lines 91-99: You should mention that brGMGTs have previously been called H-brGDGTs in the literature.

We would mention this as follows:

*"Compared with brGDGTs, the branched glycerol monoalkyl glycerol tetraethers (brGMGTs, also referred as H-brGDGTs) are a much less studied group of lipids."*

Lines 91-111: This paragraph was very well written and is an example of the standard the entire manuscript should meet.

As said above, we would carefully check the English quality of our revised manuscript.

Lines 117-123: You go from talking about the hypothesis you aim to test in the paper to talking about the aims of the paper. Surely your aim is to test the hypothesis you have just laid out - why do we need to talk about more aims here?

Thank you very much for this comment. We consider as appropriate to transition from stating the hypothesis to clearly discussing the aims of the paper in this paragraph.

Line 125-126: "by high population density". High population density of what?

Population density refers to the number of human beings who live in a given region. We would remove this as it is redundant with the next half sentence.

Line 127: "macrotidal". Please define this term.

Macrotidal means large tidal range. We would define this term in a revised manuscript.

Figure 1: I really like this figure - it nicely summarizes your water sampling campaign.
Thank you for this comment.

Line 167: "Both decarbonated and non-decarbonated samples (~6 mg for SPM and ~20 mg for soils) were enclosed in a tin capsule" You should mention that you will split your samples and decarbonate one aliquot. Otherwise, the reader is confused as to where your non-decarbonated samples are coming from.
Thank you for the suggestion. We would add the following sentence:
*"The samples were split, and one aliquot was decarbonated."*

Line 172-174: "The isotopic composition (δ 13C or δ 15N) was expressed as the relative difference between isotopic ratios in samples and in standards (Vienna Pee Dee Belemnite for carbon or atmospheric N2 for nitrogen)" Should be "and atmospheric N2…".
This would be corrected.

Line 176: What were these "additional…analyses"? Do you mean the same analyses aforementioned but on different samples, or different analyses on different samples?
The elemental and isotopic data of the SPM and sediments collected in 2015 and 2016 (*n*=84) were published by Thibault et al. (2019). To avoid confusion, we would rephrase this sentence as follows:
*"Additional elemental and isotopic data based on SPM and sediments collected in 2015 and 2016 (n=84) were obtained from Thibault et al. (2019)."*

Line 177: "(4-20g, n=51)" Looks to me like you've used the minus sign, not the en dash here. If so use the en dash.
This would be corrected.

Line 180-183: "The total lipid extracts were then separated into fractions of increasing polarity on an activated silica gel column, using (i) 30 mL of heptane, (ii) 30 mL of heptane:DCM (1/4, v/v), and (iii) 30 mL of DCM/MeOH (1/1, v/v) as eluents." That seems like a nonstandard amount of solvent. Are you using very large columns here? If so state how many g of silica gel were used.
We use glass columns with a total volume of ca. 10 mL to separate the lipids. The amount of solvent is three times the column volume. We would modify the text as follows:
*"The total lipid extracts were then separated into fractions of increasing polarity on an activated silica gel column (ca. 10 mg), using (i) 30 mL of heptane, (ii) 30 mL of heptane:DCM (1/4, v/v), and (iii) 30 mL of DCM/MeOH (1/1, v/v) as eluents."*

Line 233: Vegan should be vegan. No capital V.
This would be corrected.

Lines 240-243: I don't think you effectively explain how your hierarchical partitioning method actually works here. As some readers won't be familiar with this method, more details are needed.
We agree with the reviewer and would add more information about the hierarchical partitioning method as follows:
*"Briefly, this approach suggests that shared variance can be decomposed into equal components based on the number of involved predictors (environmental factors), allowing for the estimation of the relative*

*importance of each predictor by adding its partial $R^2$ to the sum of all allocated average shared $R^2$. While most selection procedures, such as forward selection, use predictor ordering to assess variable importance, hierarchical partitioning calculates individual importance (the sum of unique and total average shared effects) from all subset models, generating an unordered assessment of variable importance (Lai et al., 2022)."*

Figure 2. I really don't like how the axis of these plots has been extended to include chart labels. The top left panel scale is completely distorted by the addition of these labels. You should also define the features of your "boxes" in your box plot. These comments apply to all boxplots in the manuscript.
We agree with the reviewer and would modify the plots (notably by changing the scales) and captions of the figures accordingly.

Line 268: "The different brGDGTs were detected in all studied samples" Which brGDGTs?
We would specify all the brGDGTs which were detected in the following sentence:
*"The different brGDGTs ($IIIa_5$, $IIIb_5$, $IIIc_5$, $IIa_5$, $IIb_5$, $IIc_5$, $IIIa_6$, $IIIb_6$, $IIIc_6$, $IIa_6$, $IIb_6$, $IIc_6$, $IIIa_7$, $IIIb_7$, $IIa_7$, Ia, Ib, Ic, 1050d, and 1036d) were detected in all studied samples."*

Line 275: "The relative abundances of the brGDGTs were determined all along the Seine River basin" I feel like this sentence should be at the start of this section not in the second paragraph.
*We would prefer to keep the current arrangement. We still prefer placing the description of chromatogram at the beginning of the section and then fractional abundances, which emphasizes the logical flow of information.*

Line 290: "which explained 40.9% of the variance in two dimensions" Which two dimensions are these?
Thank you for the comment. These are the first two dimensions. We would correct a typo here and modify this paragraph as follows:
*"A Principal Component Analysis (PCA) was performed to statistically compare the fractional abundances of brGDGTs from different location (river, upstream and downstream estuary, based on SPM and sediments collected in the river channel), which explained 54.1% of the variance in the first two dimensions (Fig. 4a). The first axis (PC1) explained 40.9% of the variance, with negative loadings for most of the 6-methyl brGDGTs and positive loadings for the remaining brGDGTs (Fig. 4a)."*

Line 291: "Samples from the downstream estuary clustered well" Colloquial language, you should describe the data using words that don't convey a value judgment.
We would remove the word "well" from this sentence.

Line 314-315: "It allowed to explain 39.79% of the variability through two dimensions." Doesn't make sense - please proofread your manuscript.
We would remove this sentence.

I feel like you have just randomly placed the figures in the text. You should line up the first in-text citation of a figure with the location of the figure in the manuscript. Currently, the text and the figures are out of sequence which makes reading this document a challenge.
We have carefully checked our figures and in-text citations. They were appropriately positioned. We

present the PCA (RDA) plots for brGDGTs and brGMGTs together to avoid adding too many figures to the manuscript. In the text, we describe the results related to brGDGTs first and then those related to brGMGTs. The in-text citations indeed correspond to the order of the figures.

Figure 5: Visually this figure is quite busy. I don't think having the brGDGT names in blue (the same colour used for the downstream bubble) helps. I would use black for these names and also the arrows. We agree with the reviewer and would change the color of these names and arrows into black.

Line 336: "The brGMGTs identified in previous studies" Which brGMGTs and which studies? This lack of precise usage of language is present throughout the text.
This would be corrected in the following sentence:
*"The brGMGTs (H1020a, H1020b, H1020c, H1034a, H1034b, H1034c, and H1048) identified by Baxter et al. (2019) were detected in the samples collected across the Seine River basin."*

Line 343-345: "In SPM and river channel sediments, the total brGMGT concentration was observed to be slightly higher in the riverine part ($0.26 \pm 0.24$ µg/g Corg) than in downstream ($0.20 \pm 0.13$ µg/g Corg) and upstream estuary samples ($0.17 \pm 0.18$ µg/g Corg; Fig. S4b)." Slightly higher but not significantly higher. If it's not significant you should say so.
The difference in brGMGT concentration along the estuary is not significant. This would be acknowledged as follows:
*"In SPM and river channel sediments, the total brGMGT concentration was observed to be slightly (but not significantly) higher in the riverine part ($0.26 \pm 0.24$ µg/g $C_{org}$) than in downstream estuary ($0.20 \pm 0.13$ µg/g $C_{org}$) and upstream estuary samples ($0.17 \pm 0.18$ µg/g $C_{org}$; Fig. S5b). The total brGMGT concentrations were the lowest in soils (surficial soils and mudflat sediments) all over the basin ($0.07 \pm 0.09$ µg/g $C_{org}$; Fig. S5b)."*

Line 346: "The PCA analysis based on the brGMGT relative abundances (Fig. 5b) explained 70 % of the variance". I'm unsure what the authors are trying to say here but I think they mean that the first two PCs sum to 70%. The second half of the sentence "which allows to observe that samples from the different parts of the basin clustered well apart from each other." doesn't make sense and I'm unsure what the authors are trying to say.
Yes, the first two PCs sum to 70%. To clarify this point, the sentence would be rephrased as follows:
*"The PCA analysis based on the brGMGT relative abundances (Fig. 4b) explained 70 % of the variance in the first two dimensions, which separate samples from different parts of the basin."*

Line 357: "allows to explain" This phrase doesn't make sense in this context - please remove all uses of it from the manuscript.
This would be corrected.

Lines 406-408: "The similarity in distributions between soils and downstream samples may be due to the overrepresentation of downstream soil samples, as 82% of the soils were collected downstream (Fig. 1a and Table 1)." I don't understand your point here. Are you saying that the similarity between downstream estuary brGDGT distributions and soil brGDGTs is because the downstream estuary predominantly receives brGDGTs from downstream soils?

We thank the reviewer for this comment. We would rephrase the sentence as follows to clarify this point:
*"The similarity in brGDGT distributions between soils and downstream samples may be due to the influx of brGDGTs from the downstream soils into the downstream estuary, as 82% of the soils were collected downstream (Fig. 1a and Table 1)."*

Lines 409-412: "Nevertheless, the soil-derived brGDGT contribution to the downstream samples is expected to be much lower than the autochthonous one, as the average brGDGT concentration in soils was ca. 3 times lower than the one in downstream (i.e. SPM and river channel sediment) samples (Fig. S4a)." Right, but it's curious that the distributions are so similar between brGDGTs in soils and downstream estuaries. To bring more clarity to this point it would be interesting to see you attempt a machine learning approach (see Martinez-Sosa et al 2023, PP) to investigate whether (or not) a random forest model can distinguish soil samples from downstream estuary samples.

As previously said, we applied a machine learning approach, similar to that of Martinez-Sosa et al. (2023), to our dataset. Additional figures would be added to the supplementary material, as well as text to the discussion (cf. reply to main comments above).

Lines 426-429: It would be great to see you calculate and report IR6+7me following Wang et al (2021) to determine if these indices correlate to salinity in an estuarine location.

We have calculated $IR_{6+7me}$ as suggested by the reviewer. We would modify the figures and main text accordingly (cf. reply to main comments above) and would notably add the following sentence:

*"The salinity proxy ($IR_{6+7me}$) proposed by Wang et al. (2021) does not show significant correlation with salinity in this study ($p>0.05$, Wilcoxon test; Fig. S9). This suggests that $IR_{6+7me}$ is relatively insensitive in the Seine Estuary, potentially due to the preferential production of certain brGDGTs in specific estuarine regions."*

433-436: "The distinct behavior of 6-methyl brGDGTs between lakes and the Seine river-sea continuum might be due to the lower salinity range in the Seine River basin (0-32 psu) vs. the lakes (0-376 psu) 435 investigated by Wang et al. (2021). This suggests that the limited range of salinity variation in the Seine River basin might be insufficient to trigger significant 6-methyl brGDGT production, as observed in hypersaline lakes." This is actually incorrect. Wang et al 2021 report that IR6me is sensitive to salinity in the range of 5-1000 (mg/L) but relatively insensitive beyond this range.

We agree with the reviewer and would modify the text accordingly by removing the reference to the publication by Wang et al. (2021) here:

*"Indeed, the significant negative correlations between the salinity and the relative abundance of 6-methyl brGDGTs is observed in the Seine basin (Fig. S9), which suggests that the bacteria producing 6-methyl brGDGTs are preferentially present in the low salinity area of the estuary."*

458-460: "As the nutrient concentration is higher in the upstream part of the Seine estuary (Wei et al., 2022), the substantial 6-methyl brGDGT production observed in the aforementioned zone (260 460 < KP < 340, Fig. 8)" Right but why would the nutrient runoff be higher for this specific section of the basin? Do we see more agricultural activity here or something? It would be good to try and flesh out this point.

This specific region of the estuary is indeed characterized by intense agricultural activity, which could at least partly explain the high nutrient concentration in this zone, especially during the low-flow season. The text of the manuscript would be revised as follows:

*"As the nutrient concentration is higher in the upstream part of the Seine estuary (Wei et al., 2022), and this zone is characterized by high proportions of agricultural land use (Flipo et al., 2021), the substantial production of 6-methyl brGDGT observed in the aforementioned zone (260 < KP < 340, Fig. 8) during low flows could be attributed to elevated nutrient levels, particularly nitrogen, resulting from intense agricultural activities."*

Figure 8 and throughout: Make sure to superscript 15 in d15N and subscript 6 in IR6me.
This would be corrected.

509-510: "The current knowledge on the parameters controlling the brGMGT distributions in the terrestrial and marine realm is still limited." Why is it limited? Be specific.
Thank you for the comment. This group of lipids (brGMGTs) has only recently gained attention. Consequently, there are many aspects (e.g. controlling factors) still unknown for brGMGTs compared to brGDGTs. To be more specific, we would rephrased our sentence as follows:
*"The current knowledge on the parameters controlling the brGMGT distributions in the terrestrial and marine realm is still limited, as there is little literature available (Kirkels et al., 2022a)."*

Fig 9: Almost impossible to see the data points on some of the figure panels (e.g. panel e). Make the points bigger. Also, keep a consistent label text size to make the figure look neater. Also, you should say in the caption how you constructed the straight lines drawn through the data in some panels (e for instance). I'm assuming this is a linear regression but you have to inform the reader of your methods.
Thank you for this suggestion. To enhance visibility, we would increase the size of the data points in figure panels, especially in panel e. Additionally, we would standardize the label text size across all figure panels. Furthermore, we would provide more information in the figure captions.

557-558: " However, the average concentrations of brGMGTs are an order of magnitude lower in the soils than in the river channel sediments and SPM samples of the Seine basin (Fig. S4b)." Maybe it is, but visually it doesn't look like that, so include the numbers in this sentence. You can also argue that the brGMGT abundance within soils varies by an order of magnitude. Do you know what is driving such a large variance in the soil brGMGT abundance?
We agree with the reviewer that the brGMGT concentration in soil samples shows large variance. This highlights the need for further investigation into the environmental controls on brGMGT concentration and distribution in soils. However, as shown by the boxplot, the upper and lower quartiles as well as the median value of the soil brGMGT data are low compared to the river, upstream, and downstream samples. In any case, downstream (SPM and sediment) samples and soils display distinct distribution and concentrations, also captured by the application of the machine learning model to the brGMGT dataset (cf. reply to the main comments above).
We would consider the comment of the reviewer in a revised manuscript through the following sentence:
*"A large variance in the soil brGMGT concentration was observed (Fig. S5b), suggesting that further investigation is needed to better understand the environmental controls on the brGMGT production in soils."*

589: Missing the word "index" after BIT
This would be corrected.

You need a map showing the location of IODP 302 Hole 4A

We would add the following map showing the location of the core in the supplement:

[Figure]

Lines 605-607: "This core is considered proximal to the coast and has considerable changes in terrestrial inputs (i.e. continental spores and pollen) over time (Sluijs et al., 2009, 2006), making it a suitable paleorecord for testing runoff proxies." Again would be great to have some specifics. The readers will be interested in how close this core site was to the coast around the PETM. You should also say why there was a considerable change in terrestrial inputs (I'm assuming large changes in sea level are responsible).

We thank the reviewer for this comment. The changes of sea level are indeed responsible for the changing terrestrial inputs. We would rephrase this sentence as follows in a revised manuscript:

*"This core is considered to record significant changes in terrestrial inputs (i.e. continental spores and pollen) due to sea level changes over time (Sluijs et al., 2009, 2006), making it a suitable paleorecord for testing runoff proxies."*

Lines 616-617: "Such decreased runoff during the PETM body was previously attributed to a local sea level rise" Ah here is the explanation - this should have been in the previous paragraph. Also, be specific, are you saying there was decreased runoff during the PETM, OR did your sediment core record decreased runoff due to a change in sea level? These are two different things.

In addition to this core (Sluijs et al., 2008), a rise in sea level during the PETM has been recorded in many other sites worldwide (Speijer and Morsi, 2002; Harding et al., 2011; Sluijs et al., 2014). We would rephrase this sentence to clarify this point in a revised manuscript:

*"Such decreased runoff during the PETM body was previously attributed to a rise in sea level (Sluijs et*

*al., 2006), which has been recorded in many other sites worldwide (Speijer and Morsi, 2002; Harding et al., 2011; Sluijs et al., 2014)."*

References:

Martínez-Sosa, P., Tierney, J. E., Pérez-Angel, L. C., Stefanescu, I. C., Guo, J., Kirkels, F., ... & Reyes, A. V. (2023). Development and application of the Branched and Isoprenoid GDGT Machine learning Classification algorithm (BIGMaC) for paleoenvironmental reconstruction. Paleoceanography and Paleoclimatology, 38(7), e2023PA004611.

Wang, H., Liu, W., He, Y., Zhou, A., Zhao, H., Liu, H., Cao, Y., Hu, J., Meng, B., Jiang, J., Kolpakova, M., Krivonogov, S., and Liu, Z.: Salinity-controlled isomerization of lacustrine brGDGTs impacts the associated MBT5ME' terrestrial temperature index, Geochimica et Cosmochimica Acta, 305, 33–48, https://doi.org/10.1016/j.gca.2021.05.004, 2021.

---

## Author Response (AR1)

Dear Zhe-Xuan Zhang, I have received two independent reviews of your submitted manuscript, and have evaluated your response to their comments. In general, your replies to the reviewers' comments are satisfactory.

I still advise to address the reviewer comments better by doing the following:

1) Apply the BigMac model to your dataset, instead of doing an independent analysis. This will allow to i) test the BigMac model along an interesting environmental gradient, ii) have an additional line of evidence to assign provenance to your downstream estuary samples (L 409: soil or in-situ?), that is not skewed by your localized soil sampling scheme. This is needed for both aims of the paper 'environmental controls' and 'development of salinity proxy'.

We would like to thank the editor for her valuable and constructive comments. A point-by-point reply to the comments is provided below and is colored blue. The text has been added into the revised manuscript is shown in orange italics. The line numbers correspond to those of the manuscript with tracked changes.

We have now applied the BigMac model to our GDGT dataset (including both isoGDGTs and brGDGTs). A figure and additional discussion have been added to the revised manuscript as follows (lines 529-557):

*"In order to further assess whether downstream estuarine samples could be distinguished from soils, we applied the machine learning model (BigMac) developed by Martínez-Sosa et al. (2023) to our dataset with isoGDGT and brGDGT data as input. Most of our samples (SPM, sediments, and soils) were predicted as lake-type, with only one soil sample (soil6) collected at site B predicted as soil-type. This model suggests that, when considered altogether, the isoGDGT and brGDGT distributions are similar in aquatic and soil samples from the Seine estuary and differ from the soil-type samples described by Martínez-Sosa et al. (2023). Since the BigMac model does not include a river-type or estuary-type category (Martínez-Sosa et al., 2023), further inclusion of both isoGDGT and brGDGT data from global riverine/estuarine samples in the BigMac model may help enhance predictions for river-type or estuary-type SPM and sediment samples.*

*The BigMac model distinguishes the type of samples using $IIa_6$ and crenarchaeol as the two most important predicting variables. When accounting for both isoGDGTs and brGDGTs in the Seine River basin, the fractional abundance of crenarchaeol vs. total GDGTs (i.e. isoGDGTs + brGDGTs) varies significantly, whereas the one of $IIa_6$ does not differ significantly between the downstream estuary and soils (Fig. S8). Hence, the inclusion of isoGDGTs in the model may highly reduce the differences between sample types, as we observe significant differences in the fractional abundance of $IIa_6$ when calculated vs. total brGDGTs only (Fig. 3). As the BigMac model relies on both isoGDGT and brGDGT distribution, with no option of using brGDGTs alone, we chose to perform an independent analysis to assess the similarity in brGDGT relative abundance between downstream SPM and sediment samples on the one hand and soils from the Seine River basin on the other hand. This model was developed using the same algorithm (random forest) as Martínez-Sosa et al. (2023). Binary classification (downstream estuary vs. soils) was performed based on fractional abundances of brGDGTs. The trained model (Fig. S9) indicated distinguishable brGDGT distributions between downstream estuary (SPM and sediments) and soil samples, supporting the in situ production of brGDGTs in the downstream estuary. Although most of our soil samples were collected from the downstream estuary and showed similarity with the downstream*

*SPM and sediment samples through PCA and comparison of fractional abundances, we were able to distinguish their brGDGT compositions using machine learning."*

[Figure]

*Fig. S8. Relative abundance of $IIa_6$ (a) and crenarchaeol (b) over 19 GDGTs (GDGT-0, GDGT-1, GDGT-2, GDGT-3, Crenarchaeol, Crenarchaeol', $IIIa_5$, $IIIa_6$, $IIIb_5$, $IIIb_6$, $IIa_5$, $IIa_6$, $IIb_5$, $IIb_6$, $IIc_5$, $IIc_6$, Ia, Ib, and Ic) used in the BigMac model. Boxes show the upper and lower quartiles of the data, and whiskers show the range of the data, which are color-coded based on the sample type (downstream estuary in blue and soil in brown). The center-line in the boxes indicates the median value of the dataset. Statistical testing was performed by a Wilcoxon test (\*\*\*p < 0.001; ns, not significant, p >0.05).*

I agree with reviewer 2 that the similarity in distribution of brGDGTs present in soils and downstream estuary sediments can not be determined based on the PCA (L 296: your PCA is done correctly, but only reflects a part of the variance, as thus does not allow a straightforward comparison). Please compared fractional abundances or a set of ratios that summarizes the GDGT variability.

We agree with the editor that the PCA alone cannot determine the similarity of brGDGT distribution between soils and downstream samples. We have also compared the brGDGT fractional abundances, and included additional discussion in the revised manuscript as follows (lines 519-523):

*"Additionally, no significant differences were observed in the fractional abundances of several brGDGTs ($IIIb_6$, $IIb_6$, $IIIc_6$, $IIIa_7$, $IIa_7$, 1050d, $IIIa_5$, $IIIb_5$, $IIIb_7$, $IIIc_5$, $IIc_6$, and Ia) between soils and downstream samples (Fig. 3 and Fig. S4). This similarity in brGDGT distributions may be due to the influx of brGDGTs from the downstream soils into the downstream estuary, as 82% of the soils were collected downstream (Fig. 1a and Table 1)."*

2) Please compare your RIX values directly with the IR6+7ME and the ACE values (not just ACE and IR6+7ME values vs salinity).

Thank you for this suggestion. We added a figure in Supplementary material (Fig. S13) showing the
correlations between the RIX and ACE on the one hand and RIX and IR6+7Me on the other hand:

[Figure]

*Fig. S13. RIX plotted versus ACE and $IR_{6+7Me}$ through the linear regression. Shaded area represents*
*95% confidence intervals. Dataset is composed of SPM.*

Additional discussion was added to the revised manuscript as follows (lines 701-715):
*"Since the other salinity proxies (i.e. ACE and $IR_{6+7Me}$) have shown positive correlations with salinity in*
*previous studies (Turich and Freeman, 2011; Wang et al., 2021), they were expected to be positively*
*correlated with salinity and negatively correlated with RIX in the Seine River basin. However, the ACE*
*index (Turich and Freeman, 2011) and $IR_{6+7Me}$ (Wang et al., 2021) do not show significant correlations*
*with salinity in the Seine River basin (p>0.05, Wilcoxon test; Fig. S10) and show weak but significant*
*relations with the RIX (Fig. S13).This could be attributed to the influence of other factors than salinity on*
*these indices (i.e. ACE and $_{IR6+7Me}$). Indeed, while ACE has been successfully applied in hypersaline*
*systems (Turich and Freeman, 2011), it performs less effectively in certain saline settings due to the*
*complex sources of archaeol and GDGTs (Huguet et al., 2015) and/or distinct ionization efficiencies*
*between these compounds (He et al., 2020; Wang et al., 2021). Similarly, $IR_{6+7Me}$ may be influenced by*
*the preferential production of 6-methyl brGDGTs related with nitrogen nutrient loadings in a specific*
*region of the estuary (KP 255.6-337), as discussed in 4.1.2. Consequently, only RIX successfully tracks*
*salinity variations in this basin, while ACE and $IR_{6+7Me}$ show relative insensitivity."*

3) Comment on the potential to do a quantitative reconstruction of salinity based on the RIX index, and
whether the Seine estuary is a good location to develop this index. An additional few words on the
potential impact of soil-derived brGDGTs would be beneficial (L601).
Thank you for this comment. In the current manuscript, we introduced RIX as a proxy for riverine runoff
(river-derived organic matter). We have indeed observed significant correlations between salinity and
various brGMGTs, with different isomers showing distinct behaviors in response to salinity changes. This
observation forms the basis and rationale for our RIX index. However, a quantitative reconstruction of salinity using this index needs further exploration. Specific suggestions for this future work and an assessment of whether the Seine Estuary is a suitable location for salinity calibration have been included in the revised manuscript as follows (lines 715-717):

*"However, quantitatively reconstructing salinity with RIX is an important step forward that warrants further investigation. This requires comparing brGMGT distributions from various aquatic environments (e.g. estuaries and lakes) across salinity gradients."*

Additionally, we assume that the editor referred to soil-derived brGMGTs (not brGDGTs) here (L601). The RIX values in soils were compared with those from river, upstream estuarine, and downstream estuarine samples. Our findings indicate that RIX values in soils are close to those in downstream estuarine samples and are lower than those in river and upstream estuarine samples. This suggests that potential soil contributions would dilute the riverine brGMGT signal, further decreasing RIX. Such potential soil effects (dilution of riverine signal) in the Seine River basin are also observed in the Bay of Bengal. However, given the differences in distributions between soil and aquatic samples, as well as the lower brGMGT concentration in soils, soils may have only a limited influence on the value of RIX.

Fig. 9 was modified to include soils from the Godavari basin:

[Figure]

*Figure 9: RIX in the soils, SPM and sediment samples from Godavari River basin (India) and Bay of Bengal sediments (data from Kirkels et al. (2022a)). Statistical testing was performed by a Wilcoxon test. Boxes show the upper and lower quartiles of the data, and whiskers show the range of the data, which are color-coded based on the sample type (river in red, marine in blue, and soil in brown). The center-line in the boxes indicates the median value of the dataset.*

Additional discussion was included in the revised manuscript as follows (lines 794-798):

*"RIX values in soils (0.49±0.16) around the Godavari River basin are significantly lower than those of the river samples (p<0.05, Wilcoxon test; Fig. 9). Therefore, the potential soil contribution would dilute the riverine brGMGT signal, further decreasing RIX in marine sediments. This is consistent with the observations in the Seine River basin. However, given the distinct distributions between soil and aquatic samples and the lower brGMGT concentration in soils (Kirkels et al., 2022a), the influence of soil-derived brGMGTs on riverine RIX values may be limited."*

My own comments:

1) Based on the study at the Seine River, do the authors propose that this proxy traces salinity, terrestrial organic matter or river-derived organic matter?

Thank you for this comment. We have addressed it above. Please kindly refer to our earlier response.

2) Could you use C/N instead of TOC and TN as a commonly used geochemical proxy for soil input?

In the revised manuscript, we have replaced the boxplot of TOC and TN by a boxplot of C/N:

[Figure]

*Figure 2: Distribution of bulk parameters (C/N, $\delta^{13}C_{org}$ and $\delta^{15}N$) from soils (surficial soils and mudflat sediments) as well as river, upstream estuary and downstream estuary samples across the Seine River basin. Box plots of upstream and downstream estuary samples are based on SPM and sediments, whereas those of river samples are based only on SPM. Boxes show the upper and lower quartiles of the data, and whiskers show the range of the data, which are color-coded based on the sample type (river in red, upstream estuary in yellow, and downstream estuary in blue). The center-line in the boxes indicates the median value of the dataset. Statistical testing was performed by a Wilcoxon test (\*p < 0.05; \*\*p < 0.01; \*\*\*p < 0.001; \*\*\*\*p < 0.0001; ns, not significant, p > 0.05).*

These data are described in the result section of revised manuscript (lines 339-341):

*"Lower C/N values were observed in the river (8.04±4.31, based on SPM) and upstream estuary (9.42±3.67, based on SPM and sediments) compared to the downstream estuary (10.73±3.59, based on SPM and sediments) and soils (11.59±4.79, based on surficial soils and mudflat sediments) (Fig. 2)."*

Additional discussion is provided as follows (lines 810-820):

*"It is worth noting that another terrestrial proxy (C/N) was not included because it may be ineffective in tracing terrestrial OM in this anthropogenic estuary. The C/N ratio is commonly used as a bulk indicator of terrestrial OM, with higher values indicating a greater terrestrial contribution than marine sources (Bianchi and Canuel, 2011). However, other parameters such as decomposition processes, remineralization, and distinct sources could complicate its application (Lamb et al., 2006). In the Seine River basin, C/N values were significantly lower in the river and upstream estuary than in downstream samples (Fig. 2). Given that anthropogenic OM contains more nitrogen than natural OM, an increase in*

*anthropogenic sources would result in a decrease in C/N values (Van Den Hende et al., 2011; Liu et al., 2020). As a result, the observed decrease in C/N values in river and upstream estuarine samples could be attributed to a higher contribution of nitrogen from anthropogenic sources, possibly due to intense agricultural activities as discussed in 4.1.2. As BIT, RIX, and δ13Corg provide similar information, they may be more reliable tracers of terrestrial OM compared to C/N in the Seine River basin."*

3) Can you constrain for the Bay of Bengal at all what are the RIX values of the soils are? How would a change in soil input impact the RIX values, and does this complicate the interpretation of the RIX index as a salinity tracer? Even if it is a minor process in the Seine estuary, it might be an important driver of downcore changes in the Bay of Bengal?

Thank you very much for your comment, which was addressed above.

Minor comments:

Fig. 3: Please include what compound the name "1036d" refers to. 7-methyl brGDGTs are not included in the Fig.S1, please update. Specify that for some compounds the structure has not been described yet.

Thank you for your comment. To date, the structures of 1050d and 1036d have not been described. Therefore, we tentatively refer to these compounds as 1050d and 1036d. We have added related information into the caption of Fig. 3 (which has now been moved to Fig. S2 as suggested by reviewer #2) as follows (lines 944-945):

*"1050d and 1036d represent compounds eluting later than IIIa$_7$ and IIa$_7$, respectively."*

Additionally, the structures of 7-methyl brGDGTs (i.e., IIIa$_7$, IIa$_7$, IIIb$_7$) have been included in Fig. S1. We have also specified that, for some compounds eluting later than 7-methyl brGDGTs, their structures have not been described yet. The relevant information has been added to the caption of Fig. S1 as follows (lines 939-940):

*"Note that the structures of brGMGTs and compounds eluting later than 7-Methyl brGDGTs (1050d and 1036d) have not been described."*

L 340: Were brGMGTs present in all samples, or did some compounds fall below detection level in a certain samples type? Please include this description.

Some of the brGMGTs, especially H1034a, are below detection level in most of the SPM and sediment samples. We have now included related information in the revised manuscript as follows (lines 425-426):

*"H1034a is the least abundant isomer and is below detection limit for most of the SPM and sediment samples in the Seine River basin."*

Response to comments by reviewer #1

In this manuscript, Zhang et al propose a new proxy to reconstruct fluvial organic matter inputs to coastal marine settings. They suggest that brGMGTs are produced in-situ in rivers and estuaries and that the distribution of brGMGTs is principally controlled by salinity. Based on these facts they generate a new Riverine Index (RIX) using the fractional abundances of H1020c and H1034b versus H1020a and H1020b. To validate the RIX in deep time they compare RIX values to the BIT index and terrestrial pollen/spore deposits deposited during the PETM from IODP Expedition 302 Hole 4A. They report a closer relationship between RIX and terrestrial pollen abundance than BIT and terrestrial pollen abundance, indicating that at least in this site RIX outperforms BIT in accurately reconstructing riverine inputs. In all, this is an interesting study that will likely be of interest to BG readers. I have a number of comments that aim to strengthen the manuscript.

We would like to thank the reviewer for all the constructive comments and the positive assessment on the significance of our manuscript. A point-by-point reply to all the reviewer's comment is provided below and is colored blue. The text has been added into the revised manuscript is shown in orange italics. The line numbers correspond to those of the manuscript with tracked changes.

General comments:

In some sections (see specific comments) the use of English in the paper is poor and obfuscates the meaning of the text. I suggest that the authors carefully read through the manuscript to catch all typos and grammatical errors. Likewise figure quality varies considerably. In some cross plots, it is impossible to see the data as the marker size is so small (see specific comments). Characters that should be superscripted/subscripted are left as regular text (see specific comments). Lines of best fit are drawn through data but there is no information as to how these lines were constructed (see specific comments). As such, this manuscript would benefit greatly from more attention to detail from the authors.

We have carefully checked the manuscript for any typos and grammatical errors to improve its readability and clarity. Regarding the figure quality, we have addressed all the issues highlighted by the reviewer, including marker size, superscripting/subscripting, and provided detailed information (e.g. lines of best fit) in the figure caption.

Additionally, as the authors are proposing a new GDGT salinity index, I would like them to calculate and report previously formulated salinity indices from their samples. Specifically, the ACE index (Turich and Freeman 2011) and the IR6+7me (Wang et al 2021) both have been calibrated against water salinity in marine saline ponds and hypersaline lakes respectively. I know the author's brief touched on comparing IR6me from this study to values from Wang et al (2021) in the text but a more thorough examination of prior GDGT-derived water salinity reconstructions would strengthen the manuscript. Readers will be interested to see how these indices compare against RIX in reconstructing salinity from an estuarine environment.

We agree with the suggestion by the reviewer. We have calculated the ACE and $IR_{6+7me}$ indices and have presented them in a supplementary figure (presented below), allowing the comparison of the corresponding values with those of the RIX. The ACE and $IR_{6+7me}$ indices do not correlate with salinity in the Seine River basin.

[Figure]

*Fig. S10. Salinity plotted versus ACE, IR$_{6+7Me}$, relative abundance of 6-methyl and 7-methyl brGDGTs*
*(IIIa$_6$, IIa$_6$, IIb$_6$, IIIa$_7$ and IIa$_7$) as well as compounds 1050d, 1036d, Ib, and Ic through the linear*
*regression. Shaded area represents 95% confidence intervals. Vertical error bars indicate mean ± s.d for*
*samples with the same salinity. Dataset is composed of SPM.*

We have added the following text in the revised manuscript (lines 701-715):
*"Since the other salinity proxies (i.e. ACE and IR$_{6+7Me}$) have shown positive correlations with salinity in*
*previous studies (Turich and Freeman, 2011; Wang et al., 2021), they were expected to be positively*
*correlated with salinity and negatively correlated with RIX in the Seine River basin. However, the ACE*
*index (Turich and Freeman, 2011) and IR$_{6+7Me}$ (Wang et al., 2021) do not show significant correlations*
*with salinity in the Seine River basin (p>0.05, Wilcoxon test; Fig. S10) and show weak but significant*
*relations with the RIX (Fig. S13). This could be attributed to the influence of other factors than salinity*
*on these indices (i.e. ACE and IR$_{6+7Me}$). Indeed, while ACE has been successfully applied in hypersaline*
*systems (Turich and Freeman, 2011), it performs less effectively in certain saline settings due to the*
*complex sources of archaeol and GDGTs (Huguet et al., 2015) and/or distinct ionization efficiencies*
*between these compounds (He et al., 2020; Wang et al., 2021). Similarly, the IR$_{6+7Me}$ may be influenced*
*by the preferential production of 6-methyl brGDGTs related with nitrogen nutrient loadings in a specific*
*region of the estuary (KP 255.6-337), as discussed in 4.1.2. Consequently, only the RIX successfully tracks*
*salinity variations in this basin, while ACE and IR$_{6+7Me}$ show relative insensitivity."*

Additionally, the evidence for in situ production of brGDGTs and brGMGTs in downstream estuary sites
is pretty weak. This is demonstrated by Fig 2 where we see that distributions of d13Corg and d15N in
soils and downstream estuary samples are very similar in addition to Fig 5 where your PCA on sample
brGDGT and brGMGT distributions cannot separate out soils from downstream estuary samples. Yes you see (on average) higher concentrations of brGDGTs and brGMGTs in downstream estuary samples than in soils but the actual distributions of brGDGT and brGMGT abundance in soils are pretty large, indicating that some soils have pretty substantial quantities of these compounds. A great way to add more clarity to this sourcing issue is to train a random forest model using a similar method to Martinez-Sosa et al (2023) on your brGDGT and brGMGT samples (and isoGDGTs as these should be available to you). If the random forest model can accurately separate out soils from downstream estuary samples then you can be pretty sure that your downstream estuary samples were produced in situ. This won't require much additional work and can be implemented easily using python (https://scikit-learn.org/) or another language of your choice.

We thank the reviewer for this suggestion. As suggested by the editor, we have applied first the BigMac model (based on isoGDGTs and brGDGTs). However, the inclusion of isoGDGTs in the Seine River basin may highly reduce differences between sample types. Consequently, we used independent models for brGDGTs and brGMGTs, respectively. Please see our response to the editor (lines 16-54 in this response letter).
-
The application of this model to brGDGT and brGMGT datasets accurately separates downstream (SPM and sediment) estuarine samples from soil ones, indeed supporting *in situ* production of these lipids in downstream Seine Estuary.

In the material and methods, we added a new machine-learning section describing the model, as follows (Lines 307-319):

*"The BigMac model, developed by Martínez-Sosa et al. (2023) based on brGDGT and isoGDGT distribution, was applied. Subsequently, using the same algorithm (random forest), we developed our own model based on either brGDGTs or brGMGTs.*

*For independent models, our lipid dataset was split into a training set (75%) and a test set (25%). We then used a supervised machine-learning algorithm (random forest) to train models. This algorithm was applied to classify the downstream estuary and soil samples based on brGDGTs or brGMGTs as input, implemented using the scikit-learn library (https://github.com/scikit-learn/) (Pedregosa et al., 2011) in Python (version 3.10.12). Hyperparameter tuning was conducted using a randomized search approach implemented through the RandomizedSearchCV function in scikit-learn.*

*SHapley Additive exPlanations (SHAP) is a game-theoretical method used to interpret machine learning models (Lundberg et al., 2020). SHAP analysis was applied to identify which compounds were important for the classifications, implemented by the SHAP library in Python. A higher SHAP value indicates a more substantial contribution of the feature (brGDGTs or brGMGTs) to the predicted outcome (downstream estuary or soils)."*

Two figures (one for brGDGTs, another for brGMGTs) showing the performance of the model have been added to the supplement:

[Figure]

*Fig. S9 (brGDGTs). Evaluation of the random forest model based on brGDGTs through the confusion matrix (a), classification report (b), and receiver operating characteristic (ROC) curve (c). SHAP summary plots (d-e) show the feature importance obtained from the random forest algorithm and the SHAP library. Each bullet in the plot represents a single sample in the training set, with the color indicating the feature value (fractional abundance of the brGDGTs) from low (blue) to high (pink). The bullets positioned on the right side of the SHAP summary plot correspond to positive SHAP values, indicating a positive effect on the model output (downstream estuary or soils). The bullets on the left side of the plot indicate negative SHAP values, suggesting a negative effect on the model output. The variables (brGDGTs) with higher impact on the model performance are shown at higher positions. Training sets include downstream SPM and sediment samples (d) and soils (e).*

[Figure]

*Fig. S12 (brGMGTs). Evaluation of the random forest model based on brGMGTs through the confusion matrix (a), classification report (b), and receiver operating characteristic (ROC) curve (c). SHAP summary plots (d-e) show the feature importance obtained from the random forest algorithm and the SHAP library. Each bullet represents a single sample within the training set, with the color representing the feature value (fractional abundance of the brGMGTs) ranging from low (blue) to high (pink). The bullets positioned on the right side of the SHAP summary plot correspond to positive SHAP values, indicating a positive effect on the model output (downstream estuary or soils). The bullets on the left side of the plot indicate negative SHAP values, suggesting a negative effect on the model output. The variables (brGDGTs) with higher impact on the model performance are shown at higher positions. The training sets include downstream SPM and sediment samples (d) as well as soils (e).*

We added the following text in a revised manuscript to describe and discuss the results related to the application (i) of the Bigmac model to the brGDGT and isoGDGT dataset and (ii) our independent model applied to the brGDGT dataset (lines 529-557):

*"In order to further assess whether downstream estuarine samples could be distinguished from soils, we applied the machine learning model (BigMac) developed by Martínez-Sosa et al. (2023) to our dataset with isoGDGT and brGDGT data as input. Most of our samples (SPM, sediments, and soils) were predicted as lake-type, with only one soil sample (soil6) collected at site B predicted as soil-type. This model suggests that, when considered altogether, the isoGDGT and brGDGT distributions are similar in aquatic and soil samples from the Seine estuary and differ from the soil-type samples described by Martínez-Sosa et al. (2023). Since the BigMac model does not include a river-type or estuary-type category (Martínez-Sosa et al., 2023), further inclusion of both isoGDGT and brGDGT data from global riverine/estuarine samples in the BigMac model may help enhance predictions for river-type or estuary-type SPM and sediment samples.*

*The BigMac model distinguishes the type of samples using $IIa_6$ and crenarchaeol as the two most important predicting variables. When accounting for both isoGDGTs and brGDGTs in the Seine River basin, the fractional abundance of crenarchaeol vs. total GDGTs (i.e. isoGDGTs + brGDGTs) varies significantly, whereas the one of $IIa_6$ does not differ significantly between the downstream estuary and soils (Fig. S8). Hence, the inclusion of isoGDGTs in the model may highly reduce the differences between sample types, as we observe significant differences in the fractional abundance of $IIa_6$ when calculated vs. total brGDGTs only (Fig. 3). As the BigMac model relies on both isoGDGT and brGDGT distribution, with no option of using brGDGTs alone, we chose to perform an independent analysis to assess the similarity in brGDGT relative abundance between downstream SPM and sediment samples on the one hand and soils from the Seine River basin on the other hand. This model was developed using the same algorithm (random forest) as Martínez-Sosa et al. (2023). Binary classification (downstream estuary vs. soils) was performed based on fractional abundances of brGDGTs. The trained model (Fig. S9) indicated distinguishable brGDGT distributions between downstream estuary (SPM and sediments) and soil samples, supporting the in situ production of brGDGTs in the downstream estuary. Although most of our soil samples were collected from the downstream estuary and showed similarity with the downstream SPM and sediment samples through PCA and comparison of fractional abundances, we were able to distinguish their brGDGT compositions using machine learning."*

We also added the following text to describe and discuss the results related to the application of the model to the brGMGT dataset (lines 728-734):

*"As with brGDGTs, we applied a random forest algorithm to distinguish brGMGT distributions between downstream estuary and soil samples. This trained model accurately distinguishes soils from downstream estuarine samples (Fig. S12), indicating in situ production of brGMGTs in the downstream estuary. Given the significantly low brGMGT concentrations in soils ($p<0.05$, Wilcoxon test; Fig. S5b) and the distinct distributions between brGMGT in soils and aquatic settings identified through PCA (Fig. 4) and machine learning (Fig. S12), it can be assumed that the impact of soil-derived brGMGTs on the observed RIX signal in the water column of the Seine basin is low."*

Specific comments

Line 35: This complicates paleoenvironmental interpretations in SOME aquatic settings not ALL aquatic settings

We agree with this suggestion, this has been corrected.

Line 37: "all along this basin, from land to sea" awkward phrasing

We have rephrased this sentence as follows (lines 37-38):

*"BrGDGTs and brGMGTs were analyzed in soils, Suspended Particulate Matter (SPM), and sediments (n=237) collected along the land-sea continuum of the Seine basin."*

Line 40: "Redundancy analysis further shows that both salinity and nitrogen loadings dominantly control the brGDGT distributions." No, the loadings indicate that SALINITY (not salinity loadings) controls the brGDGT distribution.

This has been corrected.

Line 40-43: "Furthermore, the relative abundance of 6- methyl vs. 5-methyl brGDGTs (IR6Me ratio), Total Nitrogen (TN), $\delta$ 15N and chlorophyll a concentration co-vary in a specific zone with low salinity" Is this zone geographical, in your redundancy analysis, or something else?

This zone is geographical. We have added the following sentence in a revised manuscript (lines 43-46):

*"Furthermore, the relative abundance of 6-methyl vs. 5-methyl brGDGTs ($IR_{6Me}$ ratio), Total Nitrogen (TN), $\delta^{15}N$ and chlorophyll a concentration co-vary in a specific geographical zone with low salinity, suggesting that 6-methyl brGDGTs are preferentially produced under low-salinity and high-productivity conditions."*

Line 44-45: "Salinity is positively correlated with homologs H1020a and H1020b, 45 and negatively correlated with compounds H1020c and H1034b." Is this in soils, sediments or SPM?

This correlation was found in SPM. This has been specified as follows (lines 47-48):

*"Salinity is positively correlated with homologues H1020a and H1020b, and negatively correlated with compounds H1020c and H1034b in SPM."*

Line 45: "This suggests that bacteria thriving…" thriving is not the correct word (carries implications of a value judgment) replace with "living".

*This has been corrected.*

Line 45-47: It seems like you aren't mentioning results from soils and sediments, only from SPM? Or maybe all your sediment samples are exclusively from rivers? The reader is unclear on this.

Correlation between salinity and lipid distribution is based on SPM samples. We have modified the abstract as follows (lines 40-42):

*"Both types of compounds (i.e. brGDGTs and brGMGTs) are shown to be produced in situ, in freshwater and saltwater, based on their high concentrations and distinct distributions in aquatic settings (SPM and sediments) vs. soils."*

Line 51-52: "a paleorecord across the upper Paleocene and lower Eocene," You should name this record and say where it is.

We have added the name (Arctic Coring Expedition) and location (Lomonosov Ridge) of this record.

Lines 51: "showing its potential applicability in both modern samples and in paleorecords." Perhaps you could evaluate its usage in both these cases e.g. - we successfully/unsuccessfully applied RIX in …

We have rephrased this sentence as follows (lines 53-55):

*"We successfully applied RIX to the Godavari River basin (India) and a paleorecord across the upper Paleocene and lower Eocene from the Arctic Coring Expedition at Lomonosov Ridge, showing its potential applicability in both modern samples and in paleorecords."*

Line 55: ", although some of them were attributed to the phylum Acidobacteria" Imprecise language.

Thank you for the comment. We have rephrased this sentence into (lines 58-60):

*"Branched glycerol dialkyl glycerol tetraethers (brGDGTs) are membrane lipids produced by bacteria, some of them belonging to the phylum Acidobacteria."*

Line 57-58: "The distribution of brGDGTs (number of cyclopentane moieties and methyl groups; cf. structures in Fig. S1) was empirically linked with pH and Mean Annual Air Temperature" Again, imprecise language. The phrase "empirically linked" doesn't convey much useful information.

We have replaced "empirically" by "has been" (lines 64-66):

*"The distribution of brGDGTs (number of cyclopentane moieties and methyl groups; cf. structures in Fig. S1) has been linked with pH and Mean Annual Air Temperature (MAAT) in soils."*

Line 60: Should really cite some earlier lake GDGT papers in addition to Martinez-Sosa et al., 2021.

Thank you for this suggestion. We have cited some earlier work: Tierney et al., 2010 GCA and Russell et al., 2018 OG.

Lines 60-61: "The brGDGT-based proxies (i.e. MBT'5ME and CBT') have been largely applied to reconstruct MAAT and pH from sedimentary archives (Coffinet et al., 2018; Harning et al., 2020; Wang et al., 2020)." Not quite true - Martinez-Sosa et al (2021) and Dearing Crampton-Flood et al (2020) generated Bayesian linear regressions between the Mean temperature of months above freezing and MBT'5me. These BayMBT models have been used widely in the community since their publication.

We agree that the new models were not applied to records available before their publication. We have modified this sentence as follows (lines 64-66):

*"The brGDGT-based proxies (i.e. MBT'$_{5ME}$ and CBT') have been largely applied to reconstruct paleoclimate from sedimentary archives (Coffinet et al., 2018; Harning et al., 2020; Wang et al., 2020)."*

Line 62-63: "In aquatic settings, brGDGTs were initially suggested to be predominantly derived from watershed soils and transported by erosion in the sediments (Hopmans et al., 2004)." Maybe you mean "transported by erosion to the sediments"?

We agree and have rephrased this sentence as follows (line 68):

*"In aquatic settings, brGDGTs were initially suggested to be transported by erosion to the sediments."*

Lines 63-78: The use of English throughout this paragraph is poor and hard to follow. Needs copyediting.

We have rephrased this paragraph as follows (lines 68-89):

*"In aquatic settings, brGDGTs were initially suggested to be transported by erosion to the sediments (Hopmans et al., 2004). Based on this assumption, the Branched and Isoprenoid Tetraethers (BIT) index*

*was defined as the abundance ratio of the major brGDGTs to crenarchaeol (isoprenoid GDGT mainly produced by marine Nitrososphaerota). The BIT index ranges between 0 and 1, with high BIT values (around 1) reflecting a higher contribution of terrestrial organic matter compared to marine organic matter (Hopmans et al., 2004).*

*Over the last few years, the BIT index has been broadly used to quantify the relative contribution of terrestrial organic matter in aquatic systems (Xu et al., 2020; Yedema et al., 2023) and to evaluate the reliability of the TEX$_{86}$ palaeothermometer (Cramwinckel et al., 2018). However, several studies have shown that brGDGTs can also be produced in situ in aquatic settings, including rivers (e.g., De Jonge et al., 2015; Freymond et al., 2017; Kim et al., 2015; Zell et al., 2014, 2013), lakes (Tierney and Russell, 2009), and marine environments (Dearing Crampton-Flood et al., 2019; Zeng et al., 2023). This adds complexity to the identification of brGDGT sources in aquatic ecosystems and to the application of brGDGTs as (paleo)environmental proxies, including the BIT index.*

*The BIT values have all the more to be carefully interpreted, especially considering the potential influence of the selective degradation of branched vs. isoprenoid GDGTs (Smith et al., 2012). Thus, complementary molecular proxies for quantifying the input of terrestrial organic matter to aquatic settings are still needed. These proxies may cross-validate other available terrestrial proxies, such as the $\delta^{13}C$ of organic carbon (Lamb et al., 2006), heterocyst glycolipids (Kang et al., 2023), and long-chain diols (Lattaud et al., 2017)."*

Lines 63-78: You should read and cite Martinez-Sosa et al (2023) here for their work on a Random Forest approach to classify GDGT sources (i.e. Marine, Soil, Lake etc).

*Thank you for this suggestion. We have referred to the work by Martínez-Sosa et al. (2023) as follows (lines 88-93):*

*"Recently, a machine-learning approach (BIGMaC model) was proposed to infer the origin of environmental samples (e.g. soil, peat, marine and lake settings) based on their GDGT distribution (Martínez-Sosa et al., 2023). While such an approach shows potential for differentiating distinct sources of GDGTs, its application to aquatic systems has not yet been extensively explored."*

Line 80-83: "The improvement of analytical methods allowed the separation and quantification of 5-, 6- and 7-methyl brGDGTs (methyl groups at the fifth, sixth, and seventh positions; Fig. S1), that in previous chromatographic protocols co-eluted (De Jonge et al., 2013, 2014; Ding et al., 2016)." No real link between the previous paragraph and this one. Also, which methods? How did they improve?

This comment has been taken into account as follows (lines 96-102):

*"The improvement of chromatographic methods allowed the separation and quantification of 5-, 6- and 7-methyl brGDGTs (methyl groups at the fifth, sixth, and seventh positions; Fig. S1) that previously co-eluted (De Jonge et al., 2013, 2014; Ding et al., 2016). This led to the development of new brGDGT-based proxies based on these specific brGDGT isomers (De Jonge et al., 2014)."*

Lines 86-87: "In addition to temperature and pH, other environmental factors may influence brGDGT distributions in terrestrial and aquatic settings and hence the application and interpretation of brGDGT-derived proxies" This is a repetition from earlier in the introduction.

*This sentence has been removed from the revised manuscript.*

Lines 91-99: You should mention that brGMGTs have previously been called H-brGDGTs in the literature.

We have mentioned this as follows (lines 111-112):

*"Compared with brGDGTs, the branched glycerol monoalkyl glycerol tetraethers (brGMGTs, also referred as H-brGDGTs) are a much less studied group of lipids."*

Lines 91-111: This paragraph was very well written and is an example of the standard the entire manuscript should meet.

As said above, we have carefully checked the English quality of our revised manuscript.

Lines 117-123: You go from talking about the hypothesis you aim to test in the paper to talking about the aims of the paper. Surely your aim is to test the hypothesis you have just laid out - why do we need to talk about more aims here?

Thank you very much for this comment. We consider as appropriate to transition from stating the hypothesis to clearly discussing the aims of the paper in this paragraph.

Line 125-126: "by high population density". High population density of what?

Population density refers to the number of human beings who live in a given region. We have removed this as it is redundant with the next half sentence.

Line 127: "macrotidal". Please define this term.

Macrotidal means large tidal range, as specified in the sentence where it is used.

Figure 1: I really like this figure - it nicely summarizes your water sampling campaign.

Thank you for this comment.

Line 167: "Both decarbonated and non-decarbonated samples (~6 mg for SPM and ~20 mg for soils) were enclosed in a tin capsule" You should mention that you will split your samples and decarbonate one aliquot. Otherwise, the reader is confused as to where your non-decarbonated samples are coming from.

Thank you for the suggestion. We have added the following sentence (lines 202-203):

*"The samples were split, and one aliquot was decarbonated."*

Line 172-174: "The isotopic composition ($\delta$ 13C or $\delta$ 15N) was expressed as the relative difference between isotopic ratios in samples and in standards (Vienna Pee Dee Belemnite for carbon or atmospheric N2 for nitrogen)" Should be "and atmospheric N2…".

This has been corrected.

Line 176: What were these "additional…analyses"? Do you mean the same analyses aforementioned but on different samples, or different analyses on different samples?

The elemental and isotopic data of the SPM and sediments collected in 2015 and 2016 ($n$=84) were published by Thibault et al. (2019). To avoid confusion, we have rephrased this sentence as follows (lines 212-214):

*"Additional elemental and isotopic data based on SPM and sediments collected in 2015 and 2016 (n=84) were obtained from Thibault et al. (2019)."*

Line 177: "(4-20g, n=51)" Looks to me like you've used the minus sign, not the en dash here. If so use the en dash.

This has been corrected.

Line 180-183: "The total lipid extracts were then separated into fractions of increasing polarity on an activated silica gel column, using (i) 30 mL of heptane, (ii) 30 mL of heptane:DCM (1/4, v/v), and (iii) 30 mL of DCM/MeOH (1/1, v/v) as eluents." That seems like a nonstandard amount of solvent. Are you using very large columns here? If so state how many g of silica gel were used.

We use glass columns with a total volume of ca. 10 mL to separate the lipids. The amount of solvent is three times the column volume. We have modified the text as follows (lines 219-221):

*"The total lipid extracts were then separated into fractions of increasing polarity on an activated silica gel column (ca. 10 mg), using (i) 30 mL of heptane, (ii) 30 mL of heptane:DCM (1/4, v/v), and (iii) 30 mL of DCM/MeOH (1/1, v/v) as eluents."*

Line 233: Vegan should be vegan. No capital V.

This has been corrected.

Lines 240-243: I don't think you effectively explain how your hierarchical partitioning method actually works here. As some readers won't be familiar with this method, more details are needed.

We agree with the reviewer and have added more information about the hierarchical partitioning method as follows (lines 294-300):

*"Briefly, this approach suggests that shared variance can be decomposed into equal components based on the number of involved predictors (environmental factors), allowing for the estimation of the relative importance of each predictor by adding its partial $R^2$ to the sum of all allocated average shared $R^2$. While most selection procedures, such as forward selection, use predictor ordering to assess variable importance, hierarchical partitioning calculates individual importance (the sum of unique and total average shared effects) from all subset models, generating an unordered assessment of variable importance (Lai et al., 2022)."*

Figure 2. I really don't like how the axis of these plots has been extended to include chart labels. The top left panel scale is completely distorted by the addition of these labels. You should also define the features of your "boxes" in your box plot. These comments apply to all boxplots in the manuscript.

We agree with the reviewer and have modified the plots (notably by changing the scales) and captions of the figures accordingly.

Line 268: "The different brGDGTs were detected in all studied samples" Which brGDGTs?

We have specified all the brGDGTs which were detected in the following sentence (lines 347-348):

*"The different brGDGTs (IIIa$_5$, IIIb$_5$, IIIc$_5$, IIa$_5$, IIb$_5$, IIc$_5$, IIIa$_6$, IIIb$_6$, IIIc$_6$, IIa$_6$, IIb$_6$, IIc$_6$, IIIa$_7$, IIIb$_7$, IIa$_7$, Ia, Ib, Ic, 1050d, and 1036d) were detected in all studied samples."*

Line 275: "The relative abundances of the brGDGTs were determined all along the Seine River basin" I feel like this sentence should be at the start of this section not in the second paragraph.

*We prefer to keep the current arrangement. We still prefer placing the description of chromatogram at the beginning of the section and then fractional abundances, which emphasizes the logical flow of information.*

Line 290: "which explained 40.9% of the variance in two dimensions" Which two dimensions are these?
Thank you for the comment. These are the first two dimensions. We have corrected a typo here and modify this paragraph as follows (lines 368-371):
*"A Principal Component Analysis (PCA) was performed to statistically compare the fractional abundances of brGDGTs from different location (river, upstream and downstream estuary, based on SPM and sediments collected in the river channel), which explained 54.1% of the variance in the first two dimensions (Fig. 4a). The first axis (PC1) explained 40.9% of the variance, with negative loadings for most of the 6-methyl brGDGTs and positive loadings for the remaining brGDGTs (Fig. 4a)."*

Line 291: "Samples from the downstream estuary clustered well" Colloquial language, you should describe the data using words that don't convey a value judgment.
We have removed the word "well" from this sentence.

Line 314-315: "It allowed to explain 39.79% of the variability through two dimensions." Doesn't make sense - please proofread your manuscript.
We have removed this sentence.

I feel like you have just randomly placed the figures in the text. You should line up the first in-text citation of a figure with the location of the figure in the manuscript. Currently, the text and the figures are out of sequence which makes reading this document a challenge.
We have carefully checked our figures and in-text citations. They were appropriately positioned. We present the PCA (RDA) plots for brGDGTs and brGMGTs together to avoid adding too many figures to the manuscript. In the text, we describe the results related to brGDGTs first and then those related to brGMGTs. The in-text citations indeed correspond to the order of the figures.

Figure 5: Visually this figure is quite busy. I don't think having the brGDGT names in blue (the same colour used for the downstream bubble) helps. I would use black for these names and also the arrows.
We agree with the reviewer and have changed the color of these names and arrows into black.

Line 336: "The brGMGTs identified in previous studies" Which brGMGTs and which studies? This lack of precise usage of language is present throughout the text.
This has been corrected in the following sentence (lines 424-425):
*"The brGMGTs (H1020a, H1020b, H1020c, H1034a, H1034b, H1034c, and H1048) identified by Baxter et al. (2019) were detected in the samples collected across the Seine River basin."*

Line 343-345: "In SPM and river channel sediments, the total brGMGT concentration was observed to be slightly higher in the riverine part (0.26 ± 0.24 µg/g Corg) than in downstream (0.20 ± 0.13 µg/g Corg) and upstream estuary samples (0.17 ± 0.18 µg/g Corg; Fig. S4b)." Slightly higher but not significantly higher. If it's not significant you should say so.
The difference in brGMGT concentration along the estuary is not significant. This has been acknowledged as follows (lines 431-433):
*"In SPM and river channel sediments, the total brGMGT concentration was observed to be slightly (but not significantly) higher in the riverine part (0.26 ± 0.24 µg/g C$_{org}$) than in downstream estuary (0.20 ±*

*0.13 µg/g Corg) and upstream estuary samples (0.17 ± 0.18 µg/g Corg; Fig. S5b). The total brGMGT*
*concentrations were the lowest in soils (surficial soils and mudflat sediments) all over the basin (0.07 ±*
*0.09 µg/g Corg; Fig. S5b)."*

Line 346: "The PCA analysis based on the brGMGT relative abundances (Fig. 5b) explained 70 % of the variance". I'm unsure what the authors are trying to say here but I think they mean that the first two PCs sum to 70%. The second half of the sentence "which allows to observe that samples from the different parts of the basin clustered well apart from each other." doesn't make sense and I'm unsure what the authors are trying to say.

Yes, the first two PCs sum to 70%. To clarify this point, the sentence has been rephrased as follows (lines 436-437):

*"The PCA analysis based on the brGMGT relative abundances (Fig. 4b) explained 70 % of the variance in the first two dimensions, which separate samples from different parts of the basin."*

Line 357: "allows to explain" This phrase doesn't make sense in this context - please remove all uses of it from the manuscript.

This has been corrected.

Lines 406-408: "The similarity in distributions between soils and downstream samples may be due to the overrepresentation of downstream soil samples, as 82% of the soils were collected downstream (Fig. 1a and Table 1)." I don't understand your point here. Are you saying that the similarity between downstream estuary brGDGT distributions and soil brGDGTs is because the downstream estuary predominantly receives brGDGTs from downstream soils?

We thank the reviewer for this comment. We have rephrased the sentence as follows to clarify this point (lines 521-523):

*"This similarity in brGDGT distributions may be due to the influx of brGDGTs from the downstream soils into the downstream estuary, as 82% of the soils were collected downstream (Fig. 1a and Table 1)"*

Lines 409-412: "Nevertheless, the soil-derived brGDGT contribution to the downstream samples is expected to be much lower than the autochthonous one, as the average brGDGT concentration in soils was ca. 3 times lower than the one in downstream (i.e. SPM and river channel sediment) samples (Fig. S4a)." Right, but it's curious that the distributions are so similar between brGDGTs in soils and downstream estuaries. To bring more clarity to this point it would be interesting to see you attempt a machine learning approach (see Martinez-Sosa et al 2023, PP) to investigate whether (or not) a random forest model can distinguish soil samples from downstream estuary samples.

As previously said, we applied a machine learning approach, similar to that of Martinez-Sosa et al. (2023), to our dataset. Additional figures have been added to the supplementary material, as well as text to the discussion (cf. reply to main comments above).

Lines 426-429: It would be great to see you calculate and report IR6+7me following Wang et al (2021) to determine if these indices correlate to salinity in an estuarine location.

We have calculated IR6+7me as suggested by the reviewer. We have modified the figures and main text accordingly (cf. reply to main comments above) and notably added the following sentence (lines 583-586):

*"The salinity proxy ($IR_{6+7me}$) proposed by Wang et al. (2021) does not show significant correlation with salinity in this study (p>0.05, Wilcoxon test; Fig. S10). This suggests that $IR_{6+7me}$ is relatively insensitive in the Seine Estuary, potentially due to the preferential production of 6-methyl brGDGTs in specific estuarine regions (i,e. KP 255.6-337)"*

433-436: "The distinct behavior of 6-methyl brGDGTs between lakes and the Seine river-sea continuum might be due to the lower salinity range in the Seine River basin (0-32 psu) vs. the lakes (0-376 psu) 435 investigated by Wang et al. (2021). This suggests that the limited range of salinity variation in the Seine River basin might be insufficient to trigger significant 6-methyl brGDGT production, as observed in hypersaline lakes." This is actually incorrect. Wang et al 2021 report that IR6me is sensitive to salinity in the range of 5-1000 (mg/L) but relatively insensitive beyond this range.

We agree with the reviewer and have modified the text accordingly by removing the reference to the publication by Wang et al. (2021) here (lines 596-598):

*"Indeed, the significant negative correlations between the salinity and the relative abundance of 6-methyl brGDGTs is observed in the Seine basin (Fig. S10), which suggests that the bacteria producing 6-methyl brGDGTs are preferentially present in the low salinity area of the estuary."*

458-460: "As the nutrient concentration is higher in the upstream part of the Seine estuary (Wei et al., 2022), the substantial 6-methyl brGDGT production observed in the aforementioned zone (260 460 < KP < 340, Fig. 8)" Right but why would the nutrient runoff be higher for this specific section of the basin? Do we see more agricultural activity here or something? It would be good to try and flesh out this point.

This specific region of the estuary is indeed characterized by intense agricultural activity, which could at least partly explain the high nutrient concentration in this zone, especially during the low-flow season. The text of the manuscript has been revised as follows (lines 616-619):

*"As the nutrient concentration is higher in the upstream part of the Seine estuary (Wei et al., 2022), and this zone is characterized by high proportions of agricultural land use (Flipo et al., 2021), the substantial production of 6-methyl brGDGT observed in the aforementioned zone (260 < KP < 340, Fig. 8) during low flows could be attributed to elevated nutrient levels, particularly nitrogen, resulting from intense agricultural activities."*

Figure 8 and throughout: Make sure to superscript 15 in d15N and subscript 6 in IR6me.

This has been corrected.

509-510: "The current knowledge on the parameters controlling the brGMGT distributions in the terrestrial and marine realm is still limited." Why is it limited? Be specific.

Thank you for the comment. This group of lipids (brGMGTs) has only recently gained attention. Consequently, there are many aspects (e.g. controlling factors) still unknown for brGMGTs compared to brGDGTs. To be more specific, we have rephrased our sentence as follows (lines 669-670):

*"The current knowledge on the parameters controlling the brGMGT distributions in the terrestrial and marine realm is still limited, as there is little literature available (Kirkels et al., 2022a)."*

Fig 9: Almost impossible to see the data points on some of the figure panels (e.g. panel e). Make the points bigger. Also, keep a consistent label text size to make the figure look neater. Also, you should say in the caption how you constructed the straight lines drawn through the data in some panels (e for instance).

I'm assuming this is a linear regression but you have to inform the reader of your methods.

Thank you for this suggestion. To enhance visibility, we have increased the size of the data points in figure panels, especially in panel e. Additionally, we have standardized the label text size across all figure panels. Furthermore, we have provided more information in the figure captions.

557-558: " However, the average concentrations of brGMGTs are an order of magnitude lower in the soils than in the river channel sediments and SPM samples of the Seine basin (Fig. S4b)." Maybe it is, but visually it doesn't look like that, so include the numbers in this sentence. You can also argue that the brGMGT abundance within soils varies by an order of magnitude. Do you know what is driving such a large variance in the soil brGMGT abundance?

We agree with the reviewer that the brGMGT concentration in soil samples shows large variance. This highlights the need for further investigation into the environmental controls on brGMGT concentration and distribution in soils. However, as shown by the boxplot, the upper and lower quartiles as well as the median value of the soil brGMGT data are low compared to the river, upstream, and downstream samples.

In any case, downstream (SPM and sediment) samples and soils display distinct distribution and concentrations, also captured by the application of the machine learning model to the brGMGT dataset (cf. reply to the main comments above).

We have considered the comment of the reviewer in a revised manuscript through the following sentence (lines 727-728):

*"A large variance in the soil brGMGT concentration was observed (Fig. S5b), suggesting that further*

*investigation is needed to better understand the environmental controls on the brGMGT production in*

*soils."*

589: Missing the word "index" after BIT

This has been corrected.

You need a map showing the location of IODP 302 Hole 4A

We have added the following map showing the location of the core in the supplement:

[Figure]

Lines 605-607: "This core is considered proximal to the coast and has considerable changes in terrestrial
inputs (i.e. continental spores and pollen) over time (Sluijs et al., 2009, 2006), making it a suitable
paleorecord for testing runoff proxies." Again would be great to have some specifics. The readers will be
interested in how close this core site was to the coast around the PETM. You should also say why there
was a considerable change in terrestrial inputs (I'm assuming large changes in sea level are responsible).
We thank the reviewer for this comment. The changes of sea level are indeed responsible for the changing
terrestrial inputs. We have rephrased this sentence as follows (lines 851-854):
*"This core is considered to record significant changes in terrestrial inputs (i.e. continental spores and*
*pollen) due to sea level changes over time (Sluijs et al., 2009, 2006), making it a suitable paleorecord for*
*testing runoff proxies."*
Lines 616-617: "Such decreased runoff during the PETM body was previously attributed to a local sea
level rise" Ah here is the explanation - this should have been in the previous paragraph. Also, be specific,
are you saying there was decreased runoff during the PETM, OR did your sediment core record decreased
runoff due to a change in sea level? These are two different things.
In addition to this core (Sluijs et al., 2008), a rise in sea level during the PETM has been recorded in many
other sites worldwide (Speijer and Morsi, 2002; Harding et al., 2011; Sluijs et al., 2014). We have
rephrased this sentence to clarify this point in a revised manuscript (lines 863-865):
*"Such decreased runoff during the PETM body was previously attributed to a rise in sea level (Sluijs et*
*al., 2006), which has been recorded in many other sites worldwide (Speijer and Morsi, 2002; Harding et*
*al., 2011; Sluijs et al., 2014)."*

References:

Martínez-Sosa, P., Tierney, J. E., Pérez-Angel, L. C., Stefanescu, I. C., Guo, J., Kirkels, F., ... & Reyes,
A. V. (2023). Development and application of the Branched and Isoprenoid GDGT Machine learning
Classification algorithm (BIGMaC) for paleoenvironmental reconstruction. Paleoceanography and
Paleoclimatology, 38(7), e2023PA004611.

Wang, H., Liu, W., He, Y., Zhou, A., Zhao, H., Liu, H., Cao, Y., Hu, J., Meng, B., Jiang, J., Kolpakova,
M., Krivonogov, S., and Liu, Z.: Salinity-controlled isomerization of lacustrine brGDGTs impacts the
associated MBT5ME' terrestrial temperature index, Geochimica et Cosmochimica Acta, 305, 33–48,
https://doi.org/10.1016/j.gca.2021.05.004, 2021.

Response to comments by reviewer #2

The authors analyzed brGDGTs and brGMGTs in soils, suspended particulate matter, and river sediments in the Seine River basin to evaluate the environmental controls on and sources of these lipids. The basin ranges from freshwater to estuarine, allowing the authors to evaluate the effects of salinity on the GDGT compositions. The major motivation seems to be development of a new GMGT index, called "RIX", to detect terrestrial inputs of GMGTs to marine environments. The authors test this index through application of Cenozoic sections of an IODP site.

There is now a relatively large literature on the environmental controls on GDGTs, though there is less on GMGTs, and combining these across a riverine salinity gradient is a strength of the paper. Overall I think the paper does provide some novel contributions and findings that merit publication. That said, there are a number of technical problems that will require major revision before the paper can be published.

We thank the reviewer for his/her detailed comments and for recognizing the novelty and strength of our work. A point-by-point reply to all the reviewer's comment is provided below and is colored blue. The text which has been added into the revised manuscript is shown in orange italics. The line numbers correspond to those of the manuscript with tracked changes.

First, the Seine basin is complicated by the presence of a dam that separates sections of the river influenced by tides (salinity) from sections upstream. The dam also presumably traps upstream sediment and likely presents a barrier for transport of GDGTs (other than SPM). The authors also have relatively few soil sampling sites – there are only 5 sites and the soil samples are dominated by downstream estuarine soils. I don't think these challenges are adequately discussed in the paper. The dam may be a good thing for the study, since it establishes clear environmental boundaries, but it could be tricky to apply a GDGT index from this environment to other sites/time periods.

The dam of Poses (cf. location on the revised map below) is the frontier between the Seine river and estuary. It represents the upstream limit of the fluvial estuary and of the tidal propagation. It was built in 1887 to regulate the water level and to allow navigation of the ships up to Paris, whatever the season. Indeed, the average water flow of the Seine River measured at Poses is ~ 470 m$^3$ s$^{-1}$, with marked intra-annual differences between winter and summer flows (~ 250 m$^3$ s$^{-1}$ in the summer and over 700 m$^3$ s$^{-1}$ in the winter). Whatever the period of the year, at least part of the water from the Seine river upstream Poses flows to the estuary. Therefore, the dam should not prevent (part of) the riverine GDGTs associated to SPM to arrive to the estuary. Nevertheless, it cannot be excluded that part of the riverine sediments are trapped by the dam.

Regarding the estuary itself (downstream Poses), it comprises two major sections: the upstream, freshwater section (from site 5 to 12) and the lower, downstream section influenced by salinity (from site 12 to the coastal zone). All our estuarine samples were (logically) collected downstream of the dam of Poses. Therefore, the observed changes in brGDGT/brGMGT distribution and abundance all along the estuary, with distinct signal in the upstream and downstream estuarine zones, are intrinsic to the biogeochemical functioning of the Seine estuary and cannot be attributed specifically to this dam. Corresponding details were added to the revised manuscript as follows (lines 483-490):

*"The decrease in the fractional abundance of 6-methyl brGDGTs from the upstream estuary to the downstream estuary cannot be explained by the dam located at Poses (Fig. 1a). This dam separates the riverine part of the Seine from the upstream estuarine section. Even during the low-flow season (Fig. 1b), at least part of the water from the Seine River upstream of Poses flows into the estuary (Romero et al., 2019). Thus, the dam should not prevent (part of) the riverine brGDGTs associated with SPM from reaching the estuary. It cannot be excluded that part of the riverine sediments is trapped by this dam. Nevertheless, all our estuarine samples were collected downstream of the dam, implying that the observed changes in brGDGT abundance and distribution within the estuary are intrinsic to the biogeochemical functioning of the Seine estuary and cannot be attributed to the dam."*

Regarding the soils, we agree with the reviewer and acknowledge the limitations of our sampling strategy, with a low number of sampling sites, mainly located downstream. We cannot exclude that the overlay in brGDGT/brGMGT distribution between the soils and the downstream estuary SPM and sediment samples is partly due to the sampling approach. This has been specified in a revised manuscript with the following sentence (lines 521-523):
*"This similarity in brGDGT distributions may be due to the influx of brGDGTs from the downstream soils into the downstream estuary, as 82% of the soils were collected downstream (Fig. 1a and Table 1)."*

Nevertheless, the comparison of the brGDGT/brGMGT concentrations and distributions between soils and downstream estuary samples allows distinguishing the two types of samples, as captured by the application of an independent machine learning approach to our brGMGT/brGDGT datasets.

Last, we kindly disagree with reviewer 2 when saying that "it could be tricky to apply a GDGT index from this environment to other sites/time periods." The RIX index was developed based on samples from the Seine estuary. Nevertheless, it was successfully tested in both modern (Godavari River basin) and past settings (marine sedimentary core IODP 302-4A), showing its potential general applicability.

Second, there are a lot of data / statistical difficulties with this paper, the details of which are discussed below. At times the authors compare concentrations of GDGTs to evaluate in situ production, which is generally not a good way to do this due to the effects of sediment transport from soils to river to estuaries – concentrations may be higher in SPM than soils, for instance, as SPM contains less coarse-grained particles. Although the writing is a bit unclear, the authors appear to compare results of two PCAs, one on soils and one on aquatic samples, to differentiate these two sample types, which is not possible given how PCA works.

In order to better evaluate the *in situ* production of brGDGTs/brGMGTs in the estuary, a machine learning approach GDGT/GMGT datasets, as suggested by Reviewer #1. We have now several lines of evidence supporting the *in situ* production of brGDGT/brGMGTs in the Seine estuary:
1) higher brGDGT/brGMGT concentrations in aquatic environments compared with soils.
2) distinct distributions between soils and aquatic settings (riverine and upstream estuarine samples) identified by PCA.
3) the application of the machine learning approach, which allows distinguishing downstream estuary and soil samples based on brGDGT/brGMGT distributions. This is addressing the overlap observed between downstream estuary and soil samples in the PCA biplot based on brGDGT/brGMGT distributions.

As detailed in the reply to reviewer 1, additional discussion on the *in situ* production of brGDGTs/brGMGTs (based on the above mentioned points) has been added to our revised manuscript.

In addition, we would like to clarify that the PCAs of soils and aquatic samples were not done separately. The biplots do not correspond to a simple overlay. Only active individuals (river, upstream, and downstream estuarine samples) were used for principal component analysis. The coordinates of passive individuals (also known as supplementary individuals) (i.e. soil samples) were just **predicted**/projected using the **existing** PCA information obtained with active ones. This is actually a widely used approach which can be implemented by the R package FactoMineR ([https://cran.r-project.org/web/packages/FactoMineR](https://cran.r-project.org/web/packages/FactoMineR)). It has also been used in a recent GDGT paper (Kirkels et al., 2022 *Biogeosciences*), which aims to compare GDGT distributions in soils and aquatic settings. We prefer this approach as it effectively delivers the key information: brGDGT/brGMGT distributions in riverine and upstream estuarine samples are distinguishable from those in soils. However, in the PCAs based on brGDGT/brGMGT distribution, soils partly overlay with downstream estuary samples. This similarity may be at least partly attributed to our sampling strategy, given that most of the soils were collected around the downstream estuary, as mentioned in the manuscript. Nevertheless, we can efficiently distinguish brGDGT/brGMGT distributions in downstream estuarine samples from those in soils by using an independent machine learning approach as said above.

In the revised manuscript, we have modified the figure caption of the PCAs to better illustrate our methodology (lines 408-410):

*"The coordinates of soils (passive individuals) are added as an overlay and are predicted based on the information provided by the existing PCA performed on SPM and sediments (active individuals)."*

Third, Section 4.4 compares the application of the RIX to IODP site 302 to results from other measurements, such as the BIT and % terrestrial palynomorphs. The comparison is largely qualitative, and it's hard to tell from Figure 11 how well these compare in a statistical sense. Could the authors provide correlation coefficients to show that the RIX captures terrestrial inputs?

We thank the reviewer for this comment. We have provided the correlation coefficients between RIX and % terrestrial palynomorphs as well as between BIT and % terrestrial palynomorphs in our revised manuscript. The corresponding figure is provided below:

[Figure]

*Figure 10: Comparison between (a) terrestrial palynomorphs (%) and (b) BIT and RIX across the upper*
*Paleocene and lower Eocene between 391 and 367 meters composite depth below sea floor (mcd) of*
*IODP Expedition 302 Hole 4A. Terrestrial palynomorphs data are from Sluijs et al. (2006) and Sluijs et*
*al. (2009). RIX and BIT were calculated using data from Sluijs et al. (2020). Grey shading represents*
*Eocene Thermal Maximum 2 (ETM2), pre-ETM2 interval, and Paleocene-Eocene Thermal Maximum*
*(PETM). Dotted line represents cutoff values of RIX (below 0.3 for marine contribution and above 0.5 for*
*riverine contribution). Linear regression of the RIX (c) and BIT (d) against the terrestrial palynomorphs.*
*Shaded area represents 95% confidence intervals.*

We have also added the following sentence to our revised manuscript (lines 912-914):
*"This indicates that RIX performs better in this core compared with BIT, which is further supported by a*
*higher correlation coefficient observed between RIX and terrestrial palynomorphs (0.77; Fig. 10c)*
*compared with BIT and terrestrial palynomorphs (0.4; Fig. 10d)"*

Detailed comments:

Section 2.2.    It is a bit hard to tell from this description and the table exactly what samples were collected
and analyzed.    I take it from the description that 1) subsurface SPM was collected from every green dot
(correct?).    2) deeper water SPM was filtered from 5 sites (perhaps these could be indicated in the table),
3) Sediment samples from 8 cores were collected. I cannot tell from the description what depth in the core
these samples were taken from (10 cm?), nor how 8 cores yielded n = 68.

Perhaps the dots could be color coded to indicate what types of samples exist (surface SPM, subsurface
SPM, these + sediment). It might also be helpful to designate the environment type (river, upstream
estuary, downstream estuary) on the map. It would be particularly helpful to indicate the city of

Poses/location of the dam on this map.

We agree with the reviewer here. We have changed the color of the dots in the map to indicate the different
sample types. Locations where only soils were collected were indicated in black; those where only SPM
were collected were indicated in green; those where both SPM and sediments were collected were
indicated in red. In addition, the location of the dam, as well as information about the environmental type
(river, upstream estuary, downstream estuary), have been added to the map. The revised map and caption
are shown below:

[Figure]

*Figure 1: Geographical locations of sampling sites in the Seine River Basin (KP: kilometric point, the*
*distance in kilometers from the city of Paris (KP 0)). The sampling sites from upstream estuary and*
*downstream estuary are shown in the zoom-in figure. Sub-surface SPM was collected for all sites from*
*site 1 to site 18, while both sub-surface and bottom SPM were collected at sites 4, 6, 10, 13, and 15.*

To maintain the readability of the map and avoid too many colors, additional details have been provided
in the caption as well as in Table 1. Table 1 allows distinguishing 5 categories of sites depending on the
type of samples collected: 1) only soils; 2) only subsurface SPM; 3) subsurface SPM and sediments; 4)
subsurface and bottom SPM as well as sediments; 5) subsurface and bottom SPM. We have differentiated
subsurface and bottom SPM samples in this table.

Regarding the sediment samples, they were collected from 7 cores (and not 8). This typo has been
corrected in the revised manuscript. We have added further details on the sampling strategy as follows
(lines 180-184):
*" Sediments (n=68) from 7 cores (10-cm depth) were collected in the river channel at the same sites as*
*these SPM samples in 2015 and 2016 using a UWITEC corer as described by Thibault et al. (2019) (Table*
*1). These sediments were further sliced (1-cm thickness) and freeze-dried. For each core, ten samples*
*were analyzed for brGDGTs and brGMGTs, except for the one collected at site 17 in April 2016, where*
*no lipids were detected between 4-5 and 5-6 m depth."*

What differentiates "upstream estuary" and "downstream estuary"?    Is this salinity?    Or judgement?
The river and upstream estuary are differentiated by the dam located at Poses. The tide influences the estuary up to Poses, where the dam prevents further tidal propagation. The upstream and downstream estuary are differentiated based on spatiotemporal variations of salinity. The upstream estuary corresponds to the freshwater tidal sector, whereas the downstream estuary is affected by a salinity gradient (e.g. Romero et al., 2016, Environmental Science and Policy; Druine et al., 2018, Marine Geology). This has been clearly specified in the revised manuscript as follows (lines 150-155):

*"The tide influences the estuary up to the city of Poses (site 5, KP 202 in Fig. 1a; KP represents kilometric point and is defined as the distance in kilometers from the city of Paris), where a dam constitutes the boundary between the river and the estuary. Based on spatiotemporal variations of salinity, the estuary can be divided into two major parts. The upstream estuary corresponds to the freshwater tidal sector (KP 202 to KP 298, from site 5 to site 12; Fig. 1a and Table 1) and the downstream estuary is influenced by a salinity gradient (starting at KP 298, from site 12 to the coastal area; Fig. 1a and Table 1) (Romero et al., 2016; Druine et al., 2018)."*

Line 237: "correlations" here should be "relationships". These are not correlations in the statistical sense. This has been corrected.

Line 271?   Should this be "decreased in the downstream estuary" samples (not just "downstream")? Having defined upstream estuary and downstream estuary it is good to stay with these terms. The term "estuarine" has been added in this sentence as well as in other sentences throughout our manuscript.

Line 290. "Negative loadings" is confusing.   On which axis?   Both? I suggest describing the results by axis – first axis 1, then 2. To clarify this point, this sentence has been revised as follows (lines 368-371): *"A Principal Component Analysis (PCA) was performed to statistically compare the fractional abundances of brGDGTs from different location (river, upstream and downstream estuary, based on SPM and sediments collected in the river channel), which explained 54.1% of the variance in the first two dimensions (Fig. 4a). The first axis (PC1) explained 40.9% of the variance, with negative loadings for most of the 6-methyl brGDGTs and positive loadings for the remaining brGDGTs (Fig. 4a)."*

Figure 3 is not particularly helpful to the reader. If the authors wish to retain it, I suggest moving it to supplemental text. As suggested by the reviewer, we have moved this figure to the supplement (Fig. S2).

Results:

The results of the "bulk parameters" describes the elemental and bulk stable isotopic composition of the solid samples.   Nowhere does the paper describe results of other environmental parameters – temperature, etc.   It would be helpful to have at least a table indicating the mean and range of these.   I expect, for instance, that there is a large range of salinities associated with these samples and a very narrow range of temperatures (they are all close to each other). We thank the reviewer for this comment. Our revised manuscript now includes a new supplementary table (shown below) to describe the available environmental parameters (temperature, salinity, water discharge, TOC and TN):

**Table S1. Description of available environmental parameters**

|  | River | Upstream estuary | Downstream estuary | Soil |
|---|---|---|---|---|
| Min temperature (°C) | 20 | 8.49 | 6.4 | n.a. |
| Max temperature (°C) | 23.41 | 24.4 | 23.38 | n.a. |
| Mean temperature (°C) | 21.51 | 20.09 | 18.27 | n.a. |
| Number of samples | 6 | 44 | 62 | n.a. |
|  |  |  |  |  |
| Min salinity | 0 | 0 | 0.1 | n.a. |
| Max salinity | 0.3 | 0.32 | 32.3 | n.a. |
| Mean salinity | 0.2 | 0.22 | 3.77 | n.a. |
| Number of samples | 6 | 43 | 60 | n.a. |
|  |  |  |  |  |
| Min discharge (m³/s) | 99 | 99 | 99 | n.a. |
| Max discharge (m³/s) | 156 | 978 | 978 | n.a. |
| Mean discharge (m³/s) | 129.78 | 183.62 | 218.85 | n.a. |
| Number of samples | 9 | 48 | 62 | n.a. |
|  |  |  |  |  |
| Min TOC (%) | 0.82 | 0.75 | 0.11 | 0.22 |
| Max TOC (%) | 4.22 | 7.71 | 7.35 | 22.28 |
| Mean TOC (%) | 2.88 | 4.64 | 3.3 | 3.03 |
| Number of samples | 9 | 57 | 120 | 51 |
|  |  |  |  |  |
| Min TN (%) | 0.12 | 0.12 | 0.01 | 0.01 |
| Max TN (%) | 0.58 | 0.84 | 0.619 | 1.07 |
| Mean TN (%) | 0.37 | 0.51 | 0.31 | 0.24 |
| Number of samples | 9 | 57 | 120 | 51 |

n.a.= not applicable

The treatment of the soils samples in the analysis and results is difficult to understand. It appears that a
large number of soils (up to 34) was taken from some sampling sites, whereas at others 1 sample was
taken. These data were then analyzed via PCA separately from the aquatic samples, and the PCA was
overlayed onto the PCA of the aquatic samples. The authors conclude that the PCAs show that the GDGT
distribution of soils overlap with the SPM and channel sediments. It the PCAs were done separately, one
cannot simply overlay the biplots and conclude that they overlap – the PCAs may capture different
variance structures such that the PCA axes are not the same.   If the authors wish to compare the soils
and aquatic samples, do a PCA on all the data together. It's always possible to do a second PCA excluding
the soils to evaluate the variance structure of the aquatic samples alone.
This comment was addressed above.

Line 290: "explained 40.9% of the variance in two dimensions".   What is meant by this?   Based on the
plot, axis 1 captures 40.9% of the variance and axis 2 13.2%.

We thank the reviewer 2 for this comment, which was also made by reviewer 1. We have rephrased this paragraph as follows (lines 368-371):

*"A Principal Component Analysis (PCA) was performed to statistically compare the fractional abundances of brGDGTs from different location (river, upstream and downstream estuary, based on SPM and sediments collected in the river channel), which explained 54.1% of the variance in the first two dimensions (Fig. 4a). The first axis (PC1) explained 40.9% of the variance, with negative loadings for most of the 6-methyl brGDGTs and positive loadings for the remaining brGDGTs (Fig. 4a)."*

Line 346: Similar problem.   I think the authors mean that axes 1 and 2 capture 71%.   The PCA will capture more than this on axes 3 - ???

We agree with the reviewer. The first two dimensions explain 70% of the brGMGT variations. The corresponding sentence has been rephrased as follows (lines 436-437):

*"The PCA analysis based on the brGMGT relative abundances (Fig. 4b) explained 70 % of the variance in the first two dimensions, which separates samples from different parts of the basin."*

Similar problems exist in the description of the RDA, Section 3.3

We have specified that 30.2% of the variance was captured from the first two axes in the revised manuscript.

4.1.1.   Why do the authors focus on the 6-methyl brGDGTs here?

We start this section by discussing 6-methyl brGDGTs, as this group of compounds is typically produced in rivers. Nevertheless, this section is also mentioning and discussing the variations of the relative abundances of other types of brGDGTs, especially 7-methyl brGDGTs, across the salinity gradient.

Line 390:   The authors suggest that the higher abundances of 6-methyl brGDGTs in upstream vs. downstream samples may reflect degradation:

"It may reflect the fact that riverine 6-methyl brGDGTs are more easily degraded than soil-derived homologues and only partially transferred downstream."

Why would 6-methyl brGDGTs produced in a river degrade faster than those produced elsewhere? The authors argue that this could reflect attachment to particles – but how do these particles differ in upstream vs. downstream river environments.

It seems likely that production of the 6-methyl compounds is suppressed in downstream environments and the dam traps the upstream sediments (and lipids). Can the authors show that this is not the case?

The decrease in the abundance of 6-methyl brGDGTs from the upstream estuary to the downstream estuary cannot be explained by the dam located at Poses, as the latter is separating the riverine part of the Seine and the upstream part of the estuary. There is no dam between the upstream and downstream parts of the estuary (cf. revised version of the map above). Therefore, we favor other hypotheses discussed in the manuscript to explain the changes in 6-methyl brGDGT abundances along the estuary, including 1) preferential degradation of labile (riverine) 6-methyl brGDGTs, as notaby proposed by De Jonge et al. (2015) and 2) dilution by brGDGTs from other sources during downstream transport.

Regarding the first hypothesis, the higher degradation of 6-methyl brGDGTs upstream could indeed be due to the different attachment to particles upstream *vs.* downstream. The median diameter of the SPM was monitored between February 2015 and June 2016 in the upstream (sites 7 and 10) and downstream (sites 15 and 17) parts of the Seine Estuary (Druine, 2018: https://theses.hal.science/tel-01896520).
Upstream, the size of the particles showed only a slight dispersion (80-110 µm) whatever the hydrological
conditions. The homogeneity of the size of the particles in the upstream estuary likely reflects their
predominant continental origin (i.e. Seine river before the dam of Poses). In contrast, a large variability
in the size of the SPM particles was observed in the downstream estuary (15-20 µm to 80-90 µm), related
to the complex flocculation and defragmentation processes of the particles occurring in this part of the
estuary (Druine, 2018). Therefore, the variability in the size of the SPM particles from upstream to
downstream could have an influence on the brGDGT distribution in the Seine estuary. This point is now
discussed in the revised manuscript using the aforementioned data (lines 500-509):

*"Indeed, the higher degradation of 6-methyl brGDGTs upstream could be attributed to their different
attachment to particles compared to downstream. The median diameter of the SPM was monitored
between February 2015 and June 2016 in both the upstream (sites 7 and 10) and downstream (sites 15
and 17) parts of the Seine Estuary (Druine, 2018). The particle size showed only slight dispersion (80-
µm) under various hydrological conditions in the upstream estuarine section. The homogeneity in
particle size in the upstream estuary likely reflects its predominantly continental origin (i.e. from the
Seine River before the dam at Poses). In contrast, a large variability in the size of SPM particles was
observed in the downstream estuary (15-20 µm to 80-90 µm), attributed to the complex flocculation and
defragmentation processes of particles in this part of the estuary (Druine, 2018). Hence, the variability
in the size of SPM particles from upstream to downstream could influence the distribution of brGDGTs
in the Seine estuary."*

Line 405. Here the authors suggest that the brGDGT distributions in estuarine soils is similar to that of
the downstream samples, based on the PCA (see comment above about the PCA). In the next section
(4.1.2), this is not discussed and instead production of the brGDGTs in saline environments is the primary
factor accounting for compositional differences in upstream vs. downstream samples. Please coordinate
these ideas.

Since PCA alone does not allow distinguishing brGDGT distributions between soils and downstream
estuary samples, we further applied a machine learning approach as suggested by Reviewer #1. This
method supports the *in situ* production of brGDGTs by effectively distinguishing the brGDGT
distributions between downstream estuary and soil samples. As brGDGTs are produced *in situ*, we can
explore the compositional differences of these compounds from upstream to downstream and investigate
the controlling factors. This has been discussed in a revised manuscript, as also detailed in the reply to
reviewer 1.

Line 487, 559: One cannot conclude from concentrations alone that the GMGTs are produced in aquatic
environments. Soils contain abundant coarse clastic material that may be lost in the fine SPM and river
sediment. The distributions (relative abundances) of GMGTs are key to identifying in situ production.

We fully agree with reviewer 2. The relative abundances of brGMGTs are essential to identify *in situ*
production in the estuary, especially through machine learning approach. This comment was addressed
above.

---

## Author Response (AR2)

Response to comments by editor

Dear Zhe-Xuan Zhang and co-authors, we have received two revisions of your resubmitted manuscript. Minor revisions will be needed before we can accept this manuscript for publication. The following changes are requested: i) clarify the copyright of the maps in Fig. 1 (see statement from review file validation), ii) add the range of RIX values in the soil either in the text (fi L 640) or in the Fig. 8F (see comment reviewer report# 2), iii) plotting the BIT index alone does not allow to conclude that there is in-situ production, this needs to be accompanied for instance with concentration changes. To avoid making this part of the discussion too bulky by providing all lines of evidence (which might still be non-conclusive?), the authors can stress the hypothetic nature of their interpretation of the BIT index values, iv) the minor comments from reviewer report #2 will improve the manuscript, I recommend the authors to follow these too.

We would like to thank the editor for her comments. A point-by-point reply to the comments is provided below and is colored blue. The text which has been added into the revised manuscript is shown in orange italics. The line numbers correspond to those of the manuscript with tracked changes.

i) We have added the credit in the caption of Fig. 1 as follows (line 173):
"*The map was generated based on the layer from Agence de l'Eau Seine-Normandie.*"

ii) We have also added the ranges of RIX values for the soils in the text (line 641):
"*The RIX in river (0.51±0.06, SPM) and upstream estuarine (0.40±0.07, SPM and sediments) samples is significantly higher than for soils (0.21±0.13) and downstream estuarine (0.23±0.06, SPM and sediments) samples.*"

iii) We agree with the editor and reviewer that our interpretation of BIT is not conclusive. We thus tone down this statement in the revised manuscript as follows (lines 742-743; lines 754-755).
"*One hypothesis for this distinction could be the sedimentary in situ production of brGDGTs (Peterse et al., 2009).*"

"*One potential hypothesis for the variability in BIT values could be related to the in situ production of brGDGTs within sedimentary environments*"
iv) We have also addressed all the minor comments by reviewer report #2 as shown below.

Response to comments by reviewer

Zhang et al. analyzed brGDGTs and brGMGTs from the Seine basin's land-sea continuum. The authors thoroughly discussed the brGDGT and brGMGT distributions in up/down streams, rivers, and soil, as well as their spatial-temporal variations. Specifically, the authors proposed a new RIX index to evaluate the riverine organic matter inputs and show this index is applicable in two different regions. This is a well-designed research with plenty of valuable data, and shows the potential of the application of brGMGTs which we still know little about, and a limited dataset is available. My focus is on the revised manuscript, and I found it's overall in very good shape, but I still try to put in some thoughts and hope they are helpful.

When the authors try to show their RIX is working perfectly in the region (Fig. 8f), it would be necessary to present the RIX of soils, given the brGMGT distribution in downstream and soils are visually similar (Fig. 6). I understand the FA of H1020c, 1020b, and 1034b are statistically different between downstream and soil, and authors did the evaluation in Fig. S12. It's still meaningful to show that the range of soil RIX has some similarity with the downstream samples. I'm not suggesting moving Fig. S12 into the main text but feel the soil RIX needs to be presented somewhere in the main text, probably in Fig. 8f.

The authors use sedimentary in situ production of brGDGTs to explain the difference between BIT and RIX in Fig. 10 (Lines 740-755). In my opinion, this is sort of unnecessary and the relevant discussion probably needs to be removed because the contribution from sedimentary in situ brGDGTs is not clear yet. As the authors stated in the introduction section, it is already very complex to interpret BIT, and bringing in the sedimentary in situ brGDGTs did not really help clarify anything. For example, the authors speculate the high BIT during and after post-PETM could be attributed to the sedimentary in situ brGDGTs. What if the BIT during that period is 'normal', whereas the other 'good' BIT index is actually disturbed and biased to lower values?

We thank the reviewer for their valuable comments. A point-by-point reply to the comments is provided below and is colored blue. The text which has been added into the revised manuscript is shown in orange italics. The line numbers correspond to those of the manuscript with tracked changes.

We have now indicated the ranges of RIX in soils and tone down the interpretation of BIT for the paleorecord in the revised manuscript. Please kindly refer to our reply to the editor.

Some minor suggestions:
L60, better to just say temperature, rather than 'Mean Annual Air Temperature (MAAT)', because recent studies suggest that the brGDGTs may be used to reconstruct warmer season temperatures, especially for the mid to high-latitude lake and soil settings (e.g., Martínez-Sosa et al., 2021; Raberg et al., 2021, 2024).
Corrected.

L61, Consider cite Raberg et al., (2021) and Zhao et al. (2023) for the brGDGTs in lake sediments.
Thanks for the suggestions. These references have been cited in the revised manuscript.

L62, I suggest removing 'and CBT''. This index is just not that widely applied in paleoclimate research yet.
Corrected.

L73, I feel this statement needs some adjustment. There is a large body of studies showing the in situ brGDGT productions in the aquatic environment. I won't list them here, but this is not new, and I feel the authors should be more confident about this argument.
We agree with the reviewer and have modified this sentence in the revised manuscript as follows (lines 73-75):

*"In addition to terrestrial sources, brGDGTs can also originate from aquatic settings, including rivers*

*(e.g. De Jonge et al., 2015; Freymond et al., 2017; Kim et al., 2015; Zell et al., 2014, 2013), lakes (Tierney and Russell, 2009), and marine settings (Dearing Crampton-Flood et al., 2019; Zeng et al., 2023)."*

L445, is it a 'fact' that the riverine 6-methyl brGDGTs are degraded faster? I could be wrong, but I don't remember any cultural experiment that proves such preferential degradation. It's more like a hypothesis at this stage. The degradation of brGDGTs is generally slow so I feel the difference between homologues is better interpreted as mixed sources or overprinted by the in-situ production.

Thank you for this comment. This is indeed a hypothesis. We have deleted "the fact" in this sentence.

L670, This doesn't make sense to me. The RIX values of soils are lower than that in rivers, but way higher than in marine settings. The soil input, if there is any, won't decrease the RIX in marine sediments, but increase.

Thank you for the comment. To clarify, we have rephrased this sentence in the revised manuscript as follows (lines 666-669):

*"Significant differences in RIX between the soils, SPM and sediment samples from the Godavari River basin are observed (p<0.05, Wilcoxon test; Fig. 9). RIX values in soils (0.49±0.16) around the Godavari River basin are significantly higher than those the marine samples (p<0.05, Wilcoxon test; Fig. 9). Therefore, the potential soil contribution would increase the RIX in marine sediments."*

L425, 463, 532, 943 Dearing Crampton-Flood et al., 2021

Corrected.

References used in the comments:

Martínez-Sosa, P., Tierney, J. E., Stefanescu, I. C., Dearing Crampton-Flood, E., Shuman, B. N. and Routson, C.: A global Bayesian temperature calibration for lacustrine brGDGTs, Geochim. Cosmochim. Acta, 305, 87–105, doi:10.1016/j.gca.2021.04.038, 2021.

Raberg, J. H., Harning, D. J., Crump, S. E., de Wet, G., Blumm, A., Kopf, S., Geirsdóttir, Á., Miller, G. H. and Sepúlveda, J.: Revised fractional abundances and warm-season temperatures substantially improve brGDGT calibrations in lake sediments, Biogeosciences, 18(12), 3579–3603, doi:10.5194/bg-18-3579-2021, 2021.

Raberg, J. H., Crump, S. E., de Wet, G., Harning, D. J., Miller, G. H., Geirsdóttir, Á. and Sepúlveda, J.: BrGDGT lipids in cold regions reflect summer soil temperature and seasonal soil water chemistry, Geochim. Cosmochim. Acta, 369(January), 111–125, doi:10.1016/j.gca.2024.01.034, 2024.

Zhao, B., Russell, J. M., Tsai, V. C., Blaus, A., Parish, M. C., Liang, J., Wilk, A., Du, X. and Bush, M. B.: Evaluating global temperature calibrations for lacustrine branched GDGTs: Seasonal variability, paleoclimate implications, and future directions, Quat. Sci. Rev., 310, 108124, doi:10.1016/j.quascirev.2023.108124, 2023.